# Temper-Then-Tilt: Principled Unlearning for Generative Models through Tempering and Classifier Guidance

**Jacob L. Block** [1] [*]   **Mehryar Mohri** [2]   **Aryan Mokhtari** [1] [2]   **Sanjay Shakkottai** [1]

## Abstract

We study machine unlearning in large generative models by framing the task as density ratio estimation to a target distribution rather than supervised fine-tuning. While classifier guidance is a standard approach for approximating this ratio and can succeed in general, we show it can fail to faithfully unlearn with finite samples when the forget set represents a sharp, concentrated data distribution. To address this, we introduce **Temper-Then-Tilt Unlearning (T3-Unlearning)**, which freezes the base model and applies a two-step inference procedure: (i) *tempering* the base distribution to flatten high-confidence spikes, and (ii) *tilting* the tempered distribution using a lightweight classifier trained to distinguish retain from forget samples. Our theoretical analysis provides finite-sample guarantees linking the surrogate classifier's risk to unlearning error, proving that tempering is necessary to successfully unlearn for concentrated distributions. Empirical evaluations on the TOFU benchmark show that T3-Unlearning improves forget quality and generative utility over existing baselines, while training only a fraction of the parameters with a minimal runtime.

## 1. Introduction

Modern generative models are often trained on large datasets comprising dynamic information, such as private or copyrighted data with revocable permissions (Cooper et al., 2025). As a result, efficient mechanisms are needed to remove the influence of specific samples from trained models, both to comply with privacy regulations (European Union, 2016; California State Legislature, 2018) and to prevent the

extraction of unwanted information (Carlini et al., 2021).

*Machine unlearning* addresses this by modifying a model trained on a dataset $\mathcal{D}$ to approximate the outcome of retraining from scratch on a "retain set" $\mathcal{D}_r \subset \mathcal{D}$, removing the influence of the "forget set" $\mathcal{D}_f = \mathcal{D} \setminus \mathcal{D}_r$. While there are many ways to formalize and address this problem, we focus on generative models and frame unlearning as a distribution correction task rather than a parameter update task.

Specifically, we assume the original model has learned the ground truth distribution $p$ over the full dataset $\mathcal{D}$. Since $\mathcal{D}$ is a disjoint union of the retain and forget sets, $\mathcal{D} = \mathcal{D}_r \sqcup \mathcal{D}_f$, we model $p$ as a mixture of two component distributions: the target retain distribution $p_r$ and the forget distribution $p_f$, corresponding to $\mathcal{D}_r$ and $\mathcal{D}_f$, respectively. This formulation resembles the Huber contamination model (Huber, 1964), where an underlying distribution is corrupted by an adversarial contamination component. In the context of unlearning, our aim is to learn to sample from $p_r$ given access to the mixture $p$ and samples from $p_r$ and $p_f$ respectively.

A principled approach to recover the target distribution $p_r$ is to *tilt* the original distribution $p$, shifting probability via multiplicative reweighting. This arises in KL-constrained reinforcement learning (Schulman et al., 2015; 2017; Rafailov et al., 2023), importance sampling (Sugiyama et al., 2008; Gretton et al., 2009), and posterior sampling for generative models (Dhariwal & Nichol, 2021; Chung et al., 2023; Yang & Klein, 2021; Mudgal et al., 2024; Rashid et al., 2025). Rather than directly estimating the optimal tilt, given by the density ratio $p_r/p$, as in importance weight estimation methods (Sugiyama et al., 2008; Gretton et al., 2009; Kanamori et al., 2009), we adopt an implicit classifier-guided approach, leveraging the fact that this ratio is proportional to the optimal probabilistic classifier distinguishing retain from forget samples (Bickel et al., 2007; Rizvi et al., 2024).

**Contributions.** This paper develops a principled framework for unlearning in generative models based on distributional tilting. We characterize the statistical limits of classifier-guided unlearning, propose a method that overcomes these limitations, and validate its effectiveness both theoretically and empirically. We detail these contributions below.

We first study the connection between unlearning and classi-

---

[*]Work partially done while a Student Researcher at Google Research. Authors listed in alphabetical order. [1]The University of Texas at Austin. [2]Google Research. Correspondence to: Jacob L. Block <jblock@utexas.edu>.

*Proceedings of the 43$^{rd}$ International Conference on Machine Learning*, Seoul, South Korea. PMLR 306, 2026. Copyright 2026 by the author(s).

fication in the finite-sample case. We consider unconditional generative models over a continuous domain and analyze how the error of a classifier-guided estimator of the retain distribution $p_r$ depends on the excess risk $\delta$ of the learned classifier on the surrogate classification task. For an estimator $\hat{p}_r$, we distinguish two notions of unlearning error: the *Retain Error* $\mathcal{E}_r$, measured by the $p_r$-weighted discrepancy $\mathcal{E}_r(\hat{p}_r) := \text{KL}\left(p_r \,\|\, \hat{p}_r\right)$, and the *Forget Error* $\mathcal{E}_f$, measured by the $p_f$-weighted $\ell_1$-norm $\mathcal{E}_f(\hat{p}_r) := \|p_r - \hat{p}_r\|_{1, p_f}$.

Under standard classifier-guided tilting, the estimator is $\hat{p}_r \propto p \cdot \hat{f}$, where the learned classifier $\hat{f}$ approximates $p_r/p$. We show that while this approach reliably recovers the modes of $p_r$, with Retain Error scaling as $\mathcal{O}(\delta)$ (Theorem 3.3), it can leak forget set information. In particular, the Forget Error scales as $\mathcal{O}(\|p_f\|_\infty \sqrt{\delta})$ (Theorem 3.4), which becomes vacuous when $p_f$ is highly concentrated. We further show the dependence on the sharpness of $p_f$ is unavoidable. There exists a problem instance and a classifier with excess risk at most $\delta$ such that the estimator incurs Forget Error on the order of $\Omega\left(\|p_f\|_\infty \cdot \delta\right)$ (Theorem 3.5). Thus, strong classification alone does not prevent information leakage when the forget distribution is sharply peaked.

This limitation parallels known hardness results in importance sampling, reinforcement learning, and inference time alignment, which show that it is statistically and computationally difficult to add a sharp mode in a region assigned negligible probability by a base distribution (Cortes et al., 2010; Vehtari et al., 2024; Xie et al., 2025; Foster et al., 2025; Huang et al., 2025; Rohatgi et al., 2025). Crucially, our dual task of *removing* sharp modes is central to unlearning, as it captures settings in which the forget set consists of highly specific or atypical information relative to the retain set that is memorized by the model with high confidence.

To address this issue, we propose **Temper-Then-Tilt Unlearning (T3-Unlearning)**. The method first tempers the base distribution, flattening sharp probability spikes, and then applies classifier-guided tilting to the tempered distribution. Concretely, for a learned classifier $\hat{f}$ and temperature $T \geq 1$, the estimator takes the form $\hat{p}_r^{(T)} \propto p^{1/T} \cdot \hat{f}$. Tempering amplifies the influence of the classifier and enables stronger suppression of concentrated forget-set modes.

We prove that the Forget Error of $\hat{p}_r^{(T)}$ is bounded by $\mathcal{O}(\|p_f\|_\infty^{1/T} \cdot \delta^{1/2k})$, up to a tempering-induced bias term that vanishes when $T = 1$, where $k \geq T$ reflects a mild integrability condition (Theorem 3.7). This improved scaling with $\|p_f\|_\infty$ comes at the cost of slower dependence on the classifier excess risk. Finally, we bound the additional Retain Error induced by tempering, showing that this bias worsens as the entropy $H(p_r)$ decreases (Theorem 3.8). Together, these results establish a principled tradeoff between tempering-induced bias and robustness to sharply concentrated forget distributions.

**LLM Experiments.** We apply T3-Unlearning to empirical unlearning tasks for large language models (LLMs). For a given context $\boldsymbol{x}$, the classifier scores each candidate completion $(\boldsymbol{x}, y)$ and is used to tilt the tempered base distribution $p^{(T)}(y \,|\, \boldsymbol{x}) \propto p(y \,|\, \boldsymbol{x})^{1/T}$ prior to sampling (Figure 1). We implement the T3-Unlearning classifier as a small linear head on top of the fixed hidden states of the base model, efficiently unlearning while avoiding costly updates to the underlying pretrained network. Experiments on the TOFU benchmark (Maini et al., 2024) show that T3-Unlearning consistently outperforms existing baselines while incurring only a fraction of their computational cost.

## 1.1. Related Work

Theoretical studies of machine unlearning have primarily focused on recovering model parameters that match retraining from scratch on $\mathcal{D}_r$ or achieve approximate statistical indistinguishability, analyzing convex losses (Guo et al., 2020; Sekhari et al., 2021; Neel et al., 2021), linear classification (Lu et al., 2025), and overparameterized models (Block et al., 2025). In contrast to these works, which formulate unlearning as a parameter recovery problem, we treat unlearning as a distribution correction problem and operate directly on the learned data distribution. This avoids reliance on the original parameter space, which is prohibitively large for modern generative models such as LLMs.

Other works proposed data partitioning approaches which achieve exact unlearning through model checkpointing (Bourtoule et al., 2021; Ghazi et al., 2023). While effective in their intended settings, they rely on specialized training pipelines and are not applicable to arbitrarily trained models.

A separate line of work addresses unlearning in LLMs through supervised fine-tuning objectives that explicitly degrade performance on the forget set while preserving performance on the retain set (Maini et al., 2024; Wang et al., 2025; Yang et al., 2025; Dong et al., 2025; Li et al., 2024; Zhang et al., 2024; Fan et al., 2025). These schemes require fine-tuning the base model, which is computationally costly and can induce unintended degradation on behavior unrelated to the unlearning task. In contrast, our method freezes the base model and instead learns a distributional reweighting via a lightweight parameterization, yielding substantially lower computational cost. Further, unlike fine-tuning-based methods that suppress forget-set generations and whose ascent-style objectives can provably fail to unlearn (Mavrothalassitis et al., 2026), our method admits theoretical guarantees for minimizing specific Retain and Forget Errors that capture key notions of discrepancy with respect to the ground-truth retain distribution.

Recent work, closer in spirit to our framework, has proposed an alternative unlearning paradigm for LLMs where the base model is kept frozen and unlearning is performed by learn-

ing a perturbation of the model's output distribution (Eldan & Russinovich, 2023; Ji et al., 2024; Suriyakumar et al., 2025). For a pretrained model $p_{\boldsymbol{\theta}^*}(y \mid \boldsymbol{x})$ parameterized by $\boldsymbol{\theta}^* \in \boldsymbol{\Theta}$, these methods introduce a tilt function $f_{\boldsymbol{\phi}}(\boldsymbol{x}, y)$, parameterized separately by $\boldsymbol{\phi}$, and define the updated next-token distribution as $\hat{p}(y \mid \boldsymbol{x}) \propto p_{\boldsymbol{\theta}^*}(y \mid \boldsymbol{x}) \cdot f_{\boldsymbol{\phi}}(\boldsymbol{x}, y)$. This formulation preserves the original model $p_{\boldsymbol{\theta}^*}$ and decouples unlearning from $\boldsymbol{\Theta}$, enabling updates over a lower-dimensional auxiliary parameter space. However, these approaches parameterize $f_{\boldsymbol{\phi}}$ using an auxiliary LLM or a LoRA-style adaptation (Hu et al., 2022), requiring forward and backward passes through deep networks and large amounts of data to train a highly expressive tilting function. While these methods aim to up-weight retain set sequences and suppress forget set sequences, they incur substantial computational cost and lack theoretical guarantees for recovering the target retain distribution. In contrast, building on the same tilting scheme, we propose a computationally efficient unlearning method with provable finite-sample guarantees for recovering the target distribution $p_r$.

**Notation.** Bold symbols denote vectors and multivariate quantities. We abuse notation by using the symbols $p$, $p_r$, and $p_f$ interchangeably to denote a measure over samples (e.g., $\boldsymbol{Z} \sim p$), its corresponding density, or probability mass (e.g., $p(\boldsymbol{z})$ or $p(y \mid \boldsymbol{x})$). $H(p) = \mathbb{E}\left[-\ln p\right]$ denotes entropy, and $\mathrm{Std}_p[\cdot]$ denotes standard deviation under $p$. For a function $g$ and $k \geq 1$, $\|g\|_{k,p} := \mathbb{E}_{\boldsymbol{Z} \sim p}\left[|g(\boldsymbol{Z})|^k\right]^{1/k}$.

## 2. Proposed Method: T3-Unlearning

We first formalize the problem of generative model unlearning. For generality, we present the formulation in terms of a generic random variable $\boldsymbol{Z}$ and use $p(\boldsymbol{z})$ to denote either a density or probability mass, referring to both simply as densities for brevity. We specialize to the setting of conditional generation for LLMs when needed.

We consider a generative model parameterized by $\boldsymbol{\theta}$, with query access to its density $p_{\boldsymbol{\theta}}(\boldsymbol{z})$. For LLMs, we write $\boldsymbol{Z} = (\boldsymbol{X}, Y)$ where $\boldsymbol{X}$ denotes the context and $Y$ the next token, with the conditional distribution $p_{\boldsymbol{\theta}}(y \mid \boldsymbol{x})$. As mentioned above, we are given the original pretrained model $p_{\boldsymbol{\theta}^*}$, which we assume approximates the data distribution $p$ over the full dataset $\mathcal{D} = \mathcal{D}_r \sqcup \mathcal{D}_f$, together with access to the samples in both $\mathcal{D}_r$ and $\mathcal{D}_f$. The goal is then to produce a model that generates samples from the ground-truth retain distribution $p_r$, corresponding only to the samples in $\mathcal{D}_r$.

### 2.1. Probabilistic Formulation

Under the distribution tilting framework, where the updated distribution takes the form $\hat{p}_r \propto p \cdot \hat{f}$, the optimal tilt $f^*$ that exactly recovers $p_r$ is proportional to the *density ratio* of the target $p_r$ and original distribution $p$. For autoregressive

models with $\boldsymbol{z} = (\boldsymbol{x}, y)$, this reduces to $f^*(\boldsymbol{x}, y) \propto p_r(y \mid \boldsymbol{x})/p(y \mid \boldsymbol{x})$, and more generally to $f^*(\boldsymbol{z}) \propto p_r(\boldsymbol{z})/p(\boldsymbol{z})$.

We aim to recover the unknown target distribution $p_r$ by estimating this ratio. Formally, we label each sample in the full dataset $\mathcal{D} = \mathcal{D}_r \sqcup \mathcal{D}_f$ by its membership in the retain set, yielding the labeled dataset $\mathcal{S} = \{(\boldsymbol{z}, s) \mid \boldsymbol{z} \in \mathcal{D}\}$, where $s = \mathbb{1}\{\boldsymbol{z} \in \mathcal{D}_r\}$. We model $(\boldsymbol{z}, s)$ as realizations of the random variables $(\boldsymbol{Z}, S) \in \mathcal{Z} \times \{0, 1\}$ drawn from measure $\mathbb{P}$, where the event $\{S = 1\}$ indicates that $\boldsymbol{Z}$ is a valid sample from the true unlearned model. Let $p(\boldsymbol{z})$ denote the density over $\boldsymbol{Z}$. We then define the retain and forget set component densities $p_r$ and $p_f$ as the conditional densities of $\boldsymbol{Z}$ given $S = 1$ and $S = 0$ respectively:

$$p_r(\boldsymbol{z}) := p(\boldsymbol{z} \mid S = 1), \quad p_f(\boldsymbol{z}) := p(\boldsymbol{z} \mid S = 0).$$

This generic formulation extends the retain–forget partition to the population level. Note that $p_r$ and $p_f$ may have overlapping support, meaning unlearning may require modifying the probability density or mass assigned to a point $\boldsymbol{z}$ rather than only enforcing hard exclusion constraints.

We define the population forget set proportion $\gamma$ such that $S \sim \mathrm{Bernoulli}(1 - \gamma)$. The unconditional density $p(\boldsymbol{z})$ is then the $\gamma$-weighted mixture

$$p(\boldsymbol{z}) = (1 - \gamma)p_r(\boldsymbol{z}) + \gamma p_f(\boldsymbol{z}). \tag{1}$$

By Bayes' rule, we can express the target retain distribution $p_r$ as a tilt of the original model $p$,

$$p_r(\boldsymbol{z}) \propto p(\boldsymbol{z}) \cdot \mathbb{P}\left(S = 1 \mid \boldsymbol{Z} = \boldsymbol{z}\right).$$

For the conditional LLM setting, this reduces to

$$p_r(y \mid \boldsymbol{x}) \propto p(y \mid \boldsymbol{x}) \cdot \mathbb{P}\left(S = 1 \mid (\boldsymbol{X}, Y) = (\boldsymbol{x}, y)\right).$$

This recovers the standard classifier-guidance formulation: the target distribution $p_r$ can be obtained by reweighting $p$ with the optimal tilt function $f^*$, which can be interpreted as the *posterior probability*

$$f^*(\boldsymbol{z}) := \mathbb{P}\left(S = 1 \mid \boldsymbol{Z} = \boldsymbol{z}\right). \tag{2}$$

Thus, we can unlearn in practice by estimating $f^*$ from finite samples via the surrogate task of training a probabilistic classifier to predict $S$ from $\boldsymbol{Z}$. This reframes the unlearning problem, which is often characterized by ill-posed objectives, into a well-defined supervised learning problem.

### 2.2. T3-Unlearning Procedure

Standard classifier guidance implicitly assumes access to an accurate estimate of the optimal classifier $f^*$ which yields an accurate estimate of the desired posterior. However, as detailed in Section 3, this assumption breaks down in

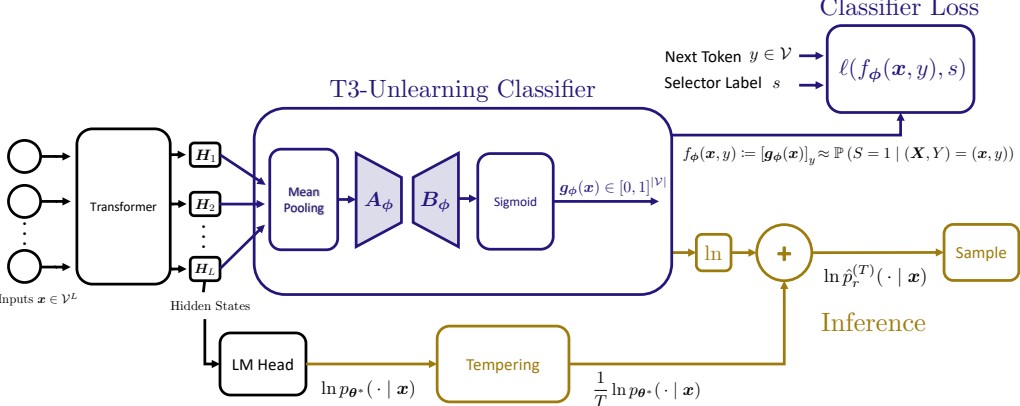

*Figure 1.* T3-Unlearning for LLMs. We freeze the base model and train a linear head (shaded) on pooled hidden states to predict the vector $\boldsymbol{g}_\phi(\boldsymbol{x})$ of class posteriors for all possible next tokens. In training, we apply the loss to the entry $[\boldsymbol{g}_\phi(\boldsymbol{x})]_y$ corresponding to the estimator of the class posterior $\mathbb{P}\left(S = 1 \mid (\boldsymbol{X}, Y) = (\boldsymbol{x}, y)\right)$, while the entire vector $\boldsymbol{g}_\phi(\boldsymbol{x})$ tilts the base model's tempered logits for inference.

the finite-sample regime when the forget distribution $p_f$ is highly concentrated. In such settings, even small classification errors can be amplified by sharp modes in $p_f$, leading to information leakage. To address this, we propose the T3-Unlearning algorithm which applies a two-stage process:

**Classifier Training.** For a dataset $\mathcal{S}_n = \{(\boldsymbol{z}_i, s_i)\}_{i=1}^n$ drawn from the data model in Section 2.1, we learn a classifier $\hat{f}(\boldsymbol{z}) \approx \mathbb{P}(S = 1 \mid \boldsymbol{Z} = \boldsymbol{z})$ within some hypothesis class $\mathcal{F}$. Given a specific parameterization $f_\phi(\boldsymbol{z})$, we minimize the regularized cross-entropy loss

$$L_n^\lambda(\boldsymbol{\phi}) = \frac{1}{n} \sum_{i=1}^n \ell(f_\phi(\boldsymbol{z}_i), s_i) + \lambda \|\boldsymbol{\phi}\|_2^2, \quad (3)$$

where $\ell(f(\boldsymbol{z}), s) = -s \ln f(\boldsymbol{z}) - (1 - s) \ln(1 - f(\boldsymbol{z}))$ and $\lambda \geq 0$ is the regularization coefficient.

**Tempered Inference.** We apply *base model tempering*, first smoothing the base distribution $p$ via a temperature $T \geq 1$ and then tilting using the learned classifier $\hat{f}$, yielding:

$$\hat{p}_r^{(T)}(\boldsymbol{z}) \propto p(\boldsymbol{z})^{1/T} \hat{f}(\boldsymbol{z}). \quad (4)$$

For LLMs, where $\hat{\boldsymbol{\phi}}$ are the learned parameters which minimize (3) and $p_{\boldsymbol{\theta}^*}$ is the frozen original model, this translates to the autoregressive update rule:

$$\hat{p}_r^{(T)}(y \mid \boldsymbol{x}) \propto p_{\boldsymbol{\theta}^*}(y \mid \boldsymbol{x})^{1/T} f_{\hat{\boldsymbol{\phi}}}(\boldsymbol{x}, y). \quad (5)$$

While standard classifier guidance sharpens the classifier to control guidance strength (Dhariwal & Nichol, 2021), we instead temper the base model, reducing the magnitude of the correction needed by the classifier. See Remark 3.6 for the limitations of classifier sharpening for unlearning.

### 2.3. LLM Implementation

We propose the following procedure for applying T3-Unlearning to LLMs (Figure 1). We define the classifier $f_\phi$

as a low-rank linear head over pre-computed features from the frozen base model $\boldsymbol{\Psi}$. For vocabulary $\mathcal{V}$ and an $L$-length context $\boldsymbol{x} \in \mathcal{V}^L$, we obtain a fixed-size representation by mean-pooling over the sequence dimension of the final hidden states $(\boldsymbol{H}_1, \ldots, \boldsymbol{H}_L) = \boldsymbol{\Psi}(\boldsymbol{x}) \in \mathbb{R}^{d \times L}$. This is passed through the low-rank classifier factorized by $\boldsymbol{A}_\phi \in \mathbb{R}^{h \times d}$, $\boldsymbol{B}_\phi \in \mathbb{R}^{|\mathcal{V}| \times h}$ ($h \ll d$) to produce per-token probabilities:

$$\boldsymbol{g}_\phi(\boldsymbol{x}) = \sigma\left(\boldsymbol{B}_\phi \boldsymbol{A}_\phi \operatorname{MeanPool}(\boldsymbol{\Psi}(\boldsymbol{x}))\right) \in [0, 1]^{|\mathcal{V}|}.$$

We then define $f_\phi(\boldsymbol{x}, y) \approx \mathbb{P}\left(S = 1 \mid (\boldsymbol{X}, Y) = (\boldsymbol{x}, y)\right)$ as the $y^{\text{th}}$ entry of $\boldsymbol{g}_\phi(\boldsymbol{x})$:

$$f_\phi(\boldsymbol{x}, y) \coloneqq [\boldsymbol{g}_\phi(\boldsymbol{x})]_y .$$

During training (Figure 1, Top), we compute the loss $\ell(f_\phi(\boldsymbol{x}, y), s)$ for each pair $(\boldsymbol{x}, y)$ and label $s$ using only the classifier output. For inference (Figure 1, Bottom), we apply the classifier guidance to the tempered base model distribution, sampling from $\hat{p}_r^{(T)}(y \mid \boldsymbol{x})$:

$$\hat{p}_r^{(T)}(y \mid \boldsymbol{x}) = \operatorname*{softmax}_y\left(\frac{\ln p_{\boldsymbol{\theta}^*}(y \mid \boldsymbol{x})}{T} + \ln[\boldsymbol{g}_\phi(\boldsymbol{x})]_y\right).$$

Our design offers two key advantages: hidden states in training can be cached, eliminating repeated forward passes through the base model, and the number of trainable parameters is minimal, enabling fast, memory-efficient unlearning.

## 3. Theoretical Guarantees

### 3.1. Unlearning Metrics

Before establishing our theoretical guarantees, we first formalize the criteria used to evaluate an unlearning procedure. Recall from (1) that we observe samples from the mixture $p(\boldsymbol{z}) = (1 - \gamma) p_r(\boldsymbol{z}) + \gamma p_f(\boldsymbol{z})$, and our goal is to generate samples according to the retain distribution $p_r(\boldsymbol{z})$. For any

candidate estimate $\hat{p}_r$ of $p_r$, we evaluate unlearning along two axes: *Retain Error*, which measures how accurately the estimate recovers the target retain distribution, and *Forget Error*, which measures how effectively it avoids leaking information in regions associated with the forget distribution.

The first notion of unlearning error, termed the *Retain Error*, evaluates the accuracy of $\hat{p}_r$ in regions where the target distribution itself places probability mass. We formally quantify this via the forward KL divergence

$$\mathcal{E}_r(\hat{p}_r) := \mathrm{KL}\left(p_r \parallel \hat{p}_r\right) = \mathbb{E}_{\boldsymbol{Z} \sim p_r}\left[\ln \frac{p_r(\boldsymbol{Z})}{\hat{p}_r(\boldsymbol{Z})}\right] \quad (6)$$

As this is an expectation under $p_r$, it is insensitive to discrepancies on sets where $p_r$ assigns negligible probability. While such insensitivity is often benign in standard estimation settings, it is problematic in unlearning: an estimate can achieve small $\mathrm{KL}\left(p_r \parallel \hat{p}_r\right)$ while assigning nontrivial probability to regions where the forget distribution $p_f$ concentrates, thereby leaking information about the forget set.

To capture this failure mode, we define a complementary notion of error, the *Forget Error*, that focuses explicitly on the behavior of $\hat{p}_r$ over regions emphasized by $p_f$. Concretely, we measure this error as the $p_f$-weighted $\ell_1$-norm

$$\mathcal{E}_f(\hat{p}_r) := \|p_r - \hat{p}_r\|_{1, p_f} = \mathbb{E}_{\boldsymbol{Z} \sim p_f}\left[|p_r(\boldsymbol{Z}) - \hat{p}_r(\boldsymbol{Z})|\right]. \tag{7}$$

Taken together, the Retain and Forget Errors provide complementary control: the former ensures faithful recovery of $p_r$ on its typical set, while the latter detects residual mismatch in regions where $p_f$ places weight, which is precisely where unlearning failures manifest.

*Remark* 3.1. While direct control of distributional discrepancies ($\mathcal{E}_r$ and $\mathcal{E}_f$) provides a definitive notion of unlearning, these quantities are intractable to compute for LLMs, as they require expectations over a combinatorial input space. Consequently, empirical evaluations rely on hypothesis tests (Maini et al., 2024) and membership inference attacks (Shi et al., 2025) to assess distinguishability from a retrained reference. Our theory targets these fundamental discrepancies directly, since minimizing them implies indistinguishability under any test function, and in Section 4 we show strong performance under the empirical proxies used in practice.

### 3.2. Surrogate Analysis Framework

We work in a general nonparametric setting where the retain and forget distributions admit densities $p_r$ and $p_f$ over a continuous domain $\mathcal{Z} \subseteq \mathbb{R}^d$. We aim to approximate $p_r$ via the T3-Unlearning estimator $\hat{p}_r^{(T)}$ defined as

$$\hat{p}_r^{(T)}(\boldsymbol{z}) = \frac{p(\boldsymbol{z})^{1/T} \hat{f}(\boldsymbol{z})}{\int_{\boldsymbol{u}} p(\boldsymbol{u})^{1/T} \hat{f}(\boldsymbol{u})}, \tag{8}$$

where $\hat{f} \in \mathcal{F}$ is a learned surrogate classifier intended to approximate the Bayes posterior $f^*$.

*Remark* 3.2. We assume $T$ is chosen so that $\int p^{1/\tau} < \infty$ for all $1 \leq \tau \leq T$, ensuring $\hat{p}_r^{(T)}$ is well-defined. This always holds for sub-exponential distributions; for power-law tails $p \propto \|\boldsymbol{z}\|_2^{-\alpha}$, it requires $\alpha > dT$, for ambient dimension $d$.

Our analysis directly relates the unlearning error of $\hat{p}_r^{(T)}$ to the statistical quality of $\hat{f}$. Define the population risk under the cross-entropy loss $\ell$ as

$$L(f) := \mathbb{E}_{(\boldsymbol{Z}, S) \sim \mathbb{P}}\left[\ell(f(\boldsymbol{Z}), S)\right].$$

We assume $\hat{f}$ achieves excess risk at most $\delta$, i.e., $L(\hat{f}) - L(f^*) \leq \delta$, and derive bounds on the Retain and Forget Errors of $\hat{p}_r^{(T)}$ as functions of $\delta$ and the mixture parameters.

While particular function classes admit standard excess risk rates when the true posterior lies in the hypothesis space (e.g., logistic regression achieving $\mathcal{O}(n^{-1/2})$ rates; see Appendix C), our theory is stated in terms of the surrogate excess risk $\delta$. This abstraction allows it to apply across learning settings with favorable risk guarantees, agnostic to any function class. Consequently, regardless of how the surrogate classifier $\hat{f}$ is obtained, we characterize unlearning performance solely through its surrogate risk.

### 3.3. Unlearning Guarantees: Untempered Estimator

To motivate the role of tempering, we first analyze the baseline that *directly* tilts the observed mixture using standard classifier-guidance. Concretely, for a classifier $\hat{f}$, we define the *untempered* estimator as $\hat{p}_r^{(1)}$, the special case of the T3-Unlearning estimator (8) with fixed temperature $T = 1$.

We first establish in Theorem 3.3 that this standard approach strictly controls Retain Error, and then subsequently show that it can fail to bound Forget Error when the forget distribution $p_f$ is sharply concentrated.

**Theorem 3.3.** *Let* $\hat{f} \in \mathcal{F}$ *satisfy the excess risk bound* $L(\hat{f}) - L(f^*) \leq \delta$. *Then the Retain Error* (6) *of the untempered estimate* $\hat{p}_r^{(1)}$ *in* (8) *satisfies*

$$\mathcal{E}_r(\hat{p}_r^{(1)}) \leq \frac{\delta}{1 - \gamma}.$$

Theorem 3.3 shows that the Retain Error of the untempered estimator is controlled by the classifier's excess risk, up to the factor $(1 - \gamma)^{-1}$. The bound scales linearly in $\delta$, as improved classification performance leads to a more accurate reconstruction of the retain distribution. The factor $(1-\gamma)^{-1}$ reflects the inverse proportion of the retain component in the original mixture. As $\gamma \to 1$, the signal from $p_r$ vanishes, rendering recovery ill-conditioned. However, in standard unlearning scenarios where the forget set is small ($\gamma \ll 1$),

this factor is negligible. In this regime, the bound in Theorem 3.3 remains sharp, effectively equating the Retain Error with the classifier's excess risk. While Theorem 3.3 shows the untempered estimator attains small Retain Error when the classifier has low excess risk (and $\gamma$ is not too large), this alone does not ensure effective unlearning. As discussed in Section 3.1, controlling the Forget Error (7) is essential to prevent information leakage from the forget set. The next theorem makes this explicit by upper-bounding the Forget Error of the untempered estimator, showing that it may degrade even when the Retain Error is small.

**Theorem 3.4.** *Let $\hat{f} \in \mathcal{F}$ satisfy the excess risk bound $L(\hat{f}) - L(f^*) \leq \delta$. Then the Forget Error (7) of the untempered estimate $\hat{p}_r^{(1)}$ in (8) satisfies*

$$\mathcal{E}_f(\hat{p}_r^{(1)}) \leq \|p_f\|_\infty \sqrt{\frac{2\delta}{1-\gamma}}.$$

The bound in Theorem 3.4 depends on the surrogate excess risk $\delta$ and the mixture weight $\gamma$ in the expected way: it improves as $\delta$ decreases and deteriorates as $\gamma$ increases, with the same $(1-\gamma)^{-1}$ amplification as in Theorem 3.3. Unlike the Retain Error guarantee, this bound scales with $\sqrt{\delta}$ and $\sqrt{(1-\gamma)^{-1}}$. This reflects that Retain Error and excess risk are both KL-based metrics, while Forget Error is $\ell_1$-based; applying Pinsker's inequality relates the two quantities, degrading the bound from linear to square-root.

Crucially, the bound also carries a multiplicative dependence on $\|p_f\|_\infty$. This captures the sharpness of the forget distribution and can be very large when $p_f$ is highly concentrated on a small region, in which case the bound may become uninformative even for small $\delta$. In the next theorem, we show that this sensitivity to the concentration of $p_f$ is not an artifact of the analysis by constructing an instance where the Forget Error is lower bounded in terms of $\|p_f\|_\infty$.

**Theorem 3.5.** *Let $p_r$ and $p_f$ have disjoint supports, with $p_f$ uniform on its support. Then for any mixture weight $\gamma \in (0,1)$ and any $\delta > 0$, there exists a classifier $\hat{f}$ achieving excess risk $L(\hat{f}) - L(f^*) \leq \delta$ such that the Forget Error (7) of the untempered estimate $\hat{p}_r^{(1)}$ in (8) satisfies*

$$\mathcal{E}_f(\hat{p}_r^{(1)}) \geq \|p_f\|_\infty \cdot \frac{\gamma \left(1 - \exp\left(-\frac{\delta}{\gamma}\right)\right)}{1 - \gamma \exp\left(-\frac{\delta}{\gamma}\right)}.$$

*In particular, as $\delta \to 0$, we have $\mathcal{E}_f(\hat{p}_r^{(1)}) = \Omega\left(\|p_f\|_\infty \cdot \delta\right)$.*

Theorem 3.5 shows that for a broad class of component distributions $p_r$ and $p_f$, the Forget Error of the untempered estimator necessarily scales with the peak density. This confirms that the dependence on $\|p_f\|_\infty$ in Theorem 3.4 is intrinsic rather than an artifact of the analysis. The proof

constructs a worst-case classifier $\hat{f}$ which incurs error only on the forget set. In this case, the Forget Error is lower bounded by the sharpness of $p_f$, quantified by its squared $\ell_2$-norm $\|p_f\|_2^2$. When $p_f$ is uniform, this coincides with $\|p_f\|_\infty$ yielding the stated bound. While the lower bound scales linearly in $\delta$, compared to the $\sqrt{\delta}$ rate in the upper bound in Theorem 3.4, it establishes the unavoidable dependence on the peak density of the forget distribution.

*Remark* 3.6 (Limitations of Classifier Sharpening). Prior work (Dhariwal & Nichol, 2021; Ji et al., 2024) controls guidance strength by sharpening the classifier $\hat{f}^w$ for $w > 0$. While this can be interpreted as a strategy for improving the surrogate task excess risk to some $\delta_w$, Theorems 3.4 and 3.5 show that such sharpening cannot eliminate the fundamental dependence of the Forget Error on the sharpness of $p_f$.

### 3.4. Unlearning Guarantees: The Tempered Estimator

Theorems 3.4 and 3.5 show that the untempered estimator is highly sensitive to the sharpness of $p_f$: its Forget Error scales linearly with $\|p_f\|_\infty$, and the corresponding lower bound confirms this dependence is intrinsic in general. We show that tempering overcomes this limitation. Under a mild integrability condition on the mixture density $p$, the $T$-tempered estimator $\hat{p}_r^{(T)}$ admits a Forget Error bound which depends on $\|p_f\|_\infty$ sublinearly as $\|p_f\|_\infty^{1/T}$, substantially improving robustness when $p_f$ is sharply concentrated.

**Theorem 3.7.** *Let $\hat{f} \in \mathcal{F}$ satisfy the excess risk bound $L(\hat{f}) - L(f^*) \leq \delta$. Define the $\tau$-tempered oracle estimate $p_r^{(\tau)} \propto p^{1/\tau} \cdot f^*$ for $\tau \in [1, T]$. Consider $k \geq T$ such that $\int p^{\frac{k-T}{T(k-1)}} < \infty$. Then for some $\tau \in [1, T]$, the Forget Error (7) of the $T$-tempered estimate $\hat{p}_r^{(T)}$ in (8) satisfies*

$$\mathcal{E}_f(\hat{p}_r^{(T)}) \leq \left(1 - \tfrac{1}{T}\right) \|p_f\|_{2, p_r^{(\tau)}} \cdot \mathrm{Std}_{p_r^{(\tau)}}[\ln p]$$
$$+ \mathcal{O}(\|p_f\|_\infty^{1/T} \cdot \delta^{1/2k}).$$

Theorem 3.7 bounds the Forget Error of the $T$-tempered estimator by a sum of two terms: a *tempering bias* term and a $\delta$-dependent *estimation error* term. The second term exhibits the key benefit of tempering: the dependence on the sharpness of the forget distribution is sublinear, scaling as $\|p_f\|_\infty^{1/T}$ rather than $\|p_f\|_\infty$ as in the untempered bound of Theorem 3.4. While the dependence on the classifier excess risk worsens from $\delta^{1/2}$ to $\delta^{1/2k}$, for fixed $\delta$ and large $\|p_f\|_\infty$ tempering can greatly improve the Forget Error guarantee.

The first term quantifies the price of tempering. It depends on the intermediate "oracle" tempered distribution $p_r^{(\tau)}$, for some $\tau \in [1, T]$, and measures the distortion caused by raising $p$ to the power $1/T$. When $p_r$ and $p_f$ are well separated, this bias is small, and it vanishes entirely in the idealized case of disjoint supports, since then $\|p_f\|_{2, p_r^{(\tau)}} = 0$. In this regime, the overall error is dominated by the estimation

term, directly addressing the failure mode of the untempered estimator, whose Forget Error provably exhibits linear dependence on $\|p_f\|_\infty$ in the disjoint-support setting (cf. Theorem 3.5). However, taking $T$ too large causes the bias term to dominate, as overaggressive tempering flattens $p$ and induces an unavoidable mismatch even when using the Bayes-optimal classifier. This indicates that minimizing Forget Error requires balancing these competing effects.

While tempering is crucial for controlling Forget Error when $p_f$ is sharply concentrated, it globally distorts the base distribution. We next quantify the induced Retain Error bias.

**Theorem 3.8.** *Let $\hat{f} \in \mathcal{F}$ satisfy the excess risk bound $L(\hat{f}) - L(f^*) \leq \delta$. Define the $\tau$-tempered density $p^{(\tau)} \propto p^{1/\tau}$ for $\tau \in [1, T]$. Then for some $\tau \in [1, T]$, the Retain Error (6) of the $T$-tempered estimate $\hat{p}_r^{(T)}$ in (8) satisfies*

$$
\mathcal{E}_r(\hat{p}_r^{(T)}) \leq \frac{\delta}{1-\gamma} +
$$

$$
\left(1 - \frac{1}{T}\right)\left(\frac{\left(\int p^{1/\tau}\right) \mathbb{E}_{\boldsymbol{Z} \sim p^{(\tau)}}[|\ln p(\boldsymbol{Z})|]}{(1-\gamma)^{\frac{\tau+1}{\tau}} \exp\left(\frac{\tau-1}{\tau} H(p_r) - \frac{\delta - \gamma \ln \gamma}{1-\gamma}\right)} - H(p_r)\right)
$$

Theorem 3.8 decomposes the Retain Error of the tempered estimator into two terms. The first, $\delta/(1-\gamma)$, matches the untempered guarantee (Theorem 3.3) and reflects error from imperfect classification. The second is a *tempering-induced bias* that arises even with the Bayes-optimal classifier, as tempering $p$ to $p^{(T)} \propto p^{1/T}$ introduces a global distortion. As expected, the bias vanishes when $T = 1$.

This clarifies when tempering inflates Retain Error: the bias is amplified when the retain distribution $p_r$ is concentrated (small $H(p_r)$), as tempering flattens high-density regions and increases mismatch. Combined with Theorem 3.7, this forms a fundamental tradeoff: $T$ should be large enough to tame $\|p_f\|_\infty$, yet not so large that tempering distortion overwhelms the gains. We empirically observe this tradeoff both in continuous-domain synthetic experiments (Appendix E) and in LLM benchmarks (Appendix F.3.3). We next present the LLM experiments in detail.

# 4. Experiments

We evaluate our method in the practical LLM setting using the Task of Fictitious Unlearning (TOFU) benchmark (Maini et al., 2024). TOFU consists of a fine-tuned LLM and a dataset of question-answer pairs regarding 200 fictitious authors generated by GPT-4 (OpenAI, 2023). The objective requires forgetting all question-answer pairs from a subset of authors (the forget set) while preserving knowledge of the remaining authors (retain set) as well as heldout real-world authors (RA) and general world facts (WF) datasets.

## 4.1. Unlearning Metrics

TOFU evaluates unlearning using several metrics built around the *Truth Ratio* ($R_{\text{truth}}$), an intermediate statistic comparing the model's relative probability of incorrect "perturbed" answers $\mathcal{A}_{\text{pert}}$ to that of the paraphrased true answer $\boldsymbol{a}^\dagger$ for a given question $\boldsymbol{q}$:

$$
R_{\text{truth}}(p_{\boldsymbol{\theta}}, \boldsymbol{q}) = \frac{\frac{1}{|\mathcal{A}_{\text{pert}}|} \sum_{\tilde{\boldsymbol{a}} \in \mathcal{A}_{\text{pert}}} p_{\boldsymbol{\theta}}(\tilde{\boldsymbol{a}} \mid \boldsymbol{q})^{1/|\tilde{\boldsymbol{a}}|}}{p_{\boldsymbol{\theta}}(\boldsymbol{a}^\dagger \mid \boldsymbol{q})^{1/|\boldsymbol{a}^\dagger|}}.
$$

From this statistic, TOFU defines two summary metrics.

**Forget Quality (FQ)** quantifies statistical indistinguishability from a retrained reference model. It is computed as the p-value of a two-sample Kolmogorov–Smirnov test comparing the Truth Ratio distributions generated by the unlearned model and the reference model over the forget set.

**Model Utility (MU)** assesses retained performance, defined as the harmonic mean, over the retain, RA, and WF datasets, of three components: (i) *Probability*, the length-normalized probability of the true answer; (ii) *ROUGE*, the ROUGE-L recall (Lin, 2004); and (iii) *TR+*, an inverted confidence score given by TR+$(p_{\boldsymbol{\theta}}, \boldsymbol{q}) = \max\{0, 1 - R_{\text{truth}}(p_{\boldsymbol{\theta}}, \boldsymbol{q})\}$. While prior work often conflates $R_{\text{truth}}$ and TR+ under the term "Truth Ratio," we distinguish them explicitly for clarity. See Appendix F.2 for detailed definitions for all metrics.

**MU-ROUGE (Generative Utility).** While MU provides a broad summary of retained performance, it aggregates heterogeneous quantities with distinct semantic meanings (Probability, ROUGE, and TR+) into a single harmonic mean. In particular, increases in token-level probability or confidence can raise the overall score even when the quality of generated text does not improve. To isolate generative performance, we propose *MU-ROUGE*, defined as the harmonic mean of ROUGE-L scores across the retain, RA, and WF datasets. By focusing on generation quality, MU-ROUGE provides a more direct and interpretable measure of retained utility. We report both MU and MU-ROUGE but emphasize MU-ROUGE as the primary utility metric.

## 4.2. Empirical Results

We use the OpenUnlearning (Dorna et al., 2025) implementation of TOFU along with the baseline methods GradAscent, GradDiff, IdkDPO (Maini et al., 2024); WGA (Wang et al., 2025); SatImp (Yang et al., 2025); UnDIAL (Dong et al., 2025); RMU (Li et al., 2024); ULD (Ji et al., 2024); NPO (Zhang et al., 2024); and SimNPO (Fan et al., 2025). We use the Llama 3.1 8B model (Llama Team, 2024) and evaluate the tasks of forgetting 5% and 10% of the authors.

To evaluate each method, we tune hyperparameters using random seeds 1 and 2 and select the best-performing configuration. In this multi-objective setting, we prioritize Forget

*Table 1.* Average Forget Quality (FQ), MU-ROUGE, and Model Utility (MU) for each unlearning method on the TOFU benchmark using Llama 3.1 8B. Each row corresponds to a mean over 5 trials. Larger values are better for all metrics. The top row reports the performance of the original model before unlearning, while the bottom shaded row corresponds to our method T3-Unlearning.

| | | | | | Utility Metrics | | | | | | | | |
| | | | | | Retain | | | WF | | | RA | | |
| Split | Method | FQ | MU-ROUGE | MU | Prob. | ROUGE | TR+ | Prob. | ROUGE | TR+ | Prob. | ROUGE | TR+ |
|---|---|---|---|---|---|---|---|---|---|---|---|---|---|
| - | Original Model | 0.000 | 0.948 | 0.637 | 0.991 | 0.995 | 0.531 | 0.479 | 0.905 | 0.625 | 0.409 | 0.949 | 0.514 |
| 5% | GradAscent | 0.000 | 0.875 | 0.576 | 0.925 | 0.852 | 0.525 | 0.420 | 0.888 | 0.543 | 0.352 | 0.885 | 0.463 |
| | GradDiff | 0.003 | 0.281 | 0.363 | 0.536 | 0.529 | 0.452 | 0.359 | 0.499 | 0.477 | 0.359 | 0.153 | 0.480 |
| | WGA | 0.252 | 0.424 | 0.418 | 0.405 | 0.436 | 0.408 | 0.374 | 0.617 | 0.502 | 0.371 | 0.327 | 0.478 |
| | SatImp | 0.445 | 0.419 | 0.412 | 0.407 | 0.436 | 0.413 | 0.368 | 0.612 | 0.489 | 0.351 | 0.313 | 0.465 |
| | UnDIAL | 0.016 | 0.653 | 0.567 | 0.601 | 0.470 | 0.460 | 0.479 | 0.832 | 0.645 | 0.470 | 0.795 | 0.596 |
| | RMU | 0.532 | 0.000 | 0.000 | 0.000 | 0.038 | 0.275 | 0.185 | 0.005 | 0.449 | 0.202 | 0.000 | 0.516 |
| | ULD | 0.169 | 0.944 | 0.644 | 0.988 | 0.974 | 0.534 | 0.498 | 0.910 | 0.641 | 0.413 | 0.950 | 0.518 |
| | IdkDPO | 0.535 | 0.669 | 0.545 | 0.678 | 0.534 | 0.481 | 0.429 | 0.833 | 0.585 | 0.402 | 0.708 | 0.518 |
| | NPO | 0.793 | 0.713 | 0.631 | 0.779 | 0.590 | 0.507 | 0.547 | 0.846 | 0.719 | 0.504 | 0.751 | 0.638 |
| | SimNPO | 0.745 | 0.786 | 0.653 | 0.830 | 0.668 | 0.508 | 0.549 | 0.868 | 0.720 | 0.505 | 0.856 | 0.634 |
| | T3-Unlearning (ours) | 0.914 | 0.900 | 0.612 | 0.415 | 0.983 | 0.461 | 0.545 | 0.874 | 0.688 | 0.511 | 0.853 | 0.648 |
| 10% | GradAscent | 0.000 | 0.950 | 0.638 | 0.991 | 0.995 | 0.531 | 0.481 | 0.910 | 0.630 | 0.409 | 0.949 | 0.512 |
| | GradDiff | 0.065 | 0.000 | 0.000 | 0.018 | 0.035 | 0.364 | 0.239 | 0.001 | 0.335 | 0.216 | 0.000 | 0.359 |
| | WGA | 0.020 | 0.834 | 0.629 | 0.873 | 0.748 | 0.502 | 0.511 | 0.898 | 0.691 | 0.430 | 0.875 | 0.545 |
| | SatImp | 0.219 | 0.301 | 0.375 | 0.524 | 0.531 | 0.444 | 0.368 | 0.517 | 0.494 | 0.353 | 0.165 | 0.462 |
| | UnDIAL | 0.000 | 0.927 | 0.697 | 0.971 | 0.961 | 0.517 | 0.561 | 0.908 | 0.729 | 0.504 | 0.914 | 0.633 |
| | RMU | 0.001 | 0.000 | 0.000 | 0.000 | 0.072 | 0.299 | 0.229 | 0.023 | 0.588 | 0.298 | 0.000 | 0.477 |
| | ULD | 0.012 | 0.944 | 0.642 | 0.988 | 0.978 | 0.533 | 0.494 | 0.904 | 0.638 | 0.412 | 0.952 | 0.517 |
| | IdkDPO | 0.020 | 0.413 | 0.465 | 0.656 | 0.509 | 0.486 | 0.431 | 0.655 | 0.586 | 0.402 | 0.271 | 0.522 |
| | NPO | 0.185 | 0.521 | 0.493 | 0.602 | 0.542 | 0.468 | 0.422 | 0.664 | 0.569 | 0.393 | 0.416 | 0.506 |
| | SimNPO | 0.536 | 0.840 | 0.662 | 0.870 | 0.745 | 0.529 | 0.506 | 0.904 | 0.685 | 0.502 | 0.890 | 0.649 |
| | T3-Unlearning (ours) | 0.671 | 0.899 | 0.612 | 0.423 | 0.992 | 0.462 | 0.543 | 0.863 | 0.686 | 0.506 | 0.856 | 0.641 |

Quality, since a utility-focused method could simply return the original model. See Appendix F.3 for details.

Average results over 5 trials are reported in Table 1 (see Appendix F.3.2 for per-trial results). The summary metrics FQ and MU-ROUGE are shaded, with T3-Unlearning highlighted in the bottom row. T3-Unlearning consistently outperforms all baselines in FQ, and achieves highest MU-ROUGE among methods with competitive Forget Quality, demonstrating a favorable utility–forgetting tradeoff. Baselines like SimNPO can achieve higher MU, as the tempering of T3-Unlearning can lower absolute probabilities without degrading the generation quality, indicated by the strong MU-ROUGE scores. We thus find MU to be an unreliable utility metric, as it can even exceed the utility of the original model before unlearning without corresponding improvements in its generations. In contrast, MU-ROUGE provides a more faithful measure of utility, as it does not meaningfully exceed the original model's performance after unlearning.

### 4.3. Unlearning Efficiency

We evaluate computational efficiency, since unlearning is fundamentally motivated by avoiding the expensive process of retraining from scratch. Beyond its strong performance, we show below that T3-Unlearning is highly efficient in both memory and computation.

*Table 2.* Trainable parameters for each unlearning method on Llama 3.1 8B. ULD uses LoRA rank 32 on the first 16 layers, while T3-Unlearning uses classifier hidden dimension $h = 20$.

| Method | # Parameters |
|---|---|
| Full Fine-Tuning | $8.03 \times 10^9$ |
| ULD (LoRA rank 32, first 16 layers) | $4.19 \times 10^7$ |
| T3-Unlearning ($h = 20$) | $2.65 \times 10^6$ |

**Parameter Efficiency.** Table 2 reports the number of trainable parameters for each method. For parameter-efficient methods (T3-Unlearning and ULD), we report the configurations that achieve the best performance in Table 1; specifically, T3-Unlearning uses a classifier hidden dimension $h = 20$, while ULD applies LoRA with rank 32 to the first 16 transformer layers. All other baselines are implemented as full fine-tuning. Under this setting, T3-Unlearning trains only 0.03% of the original model parameters.

**Runtime.** Since T3-Unlearning trains the classifier head on pooled hidden states from the frozen base model, each sample's feature representation needs to be computed only once. These fixed vectors then act as inputs for the surrogate classification task, eliminating repeated forward and backward passes through the full network required by fine-tuning.

*Table 3.* Average runtime (seconds) and epochs for each unlearning method on the 5% and 10% TOFU splits using Llama 3.1 8B.

| Method | 5% Split | | 10% Split | |
|---|---|---|---|---|
| | **Epochs** | **Runtime (s)** | **Epochs** | **Runtime (s)** |
| GradAscent | 10 | 236 | 10 | 467 |
| GradDiff | 20 | 801 | 10 | 823 |
| WGA | 10 | 529 | 5 | 524 |
| SatImp | 10 | 443 | 10 | 1040 |
| UnDIAL | 5 | 241 | 10 | 966 |
| RMU | 10 | 566 | 10 | 1100 |
| ULD | 20 | 63.7 | 20 | 121 |
| IdkDPO | 10 | 837 | 10 | 1680 |
| NPO | 20 | 1270 | 10 | 1140 |
| SimNPO | 10 | 456 | 10 | 904 |
| T3-Unlearning (naïve) | 100 | 216 | 100 | 431 |
| T3-Unlearning (preprocessed) | 100 | **5.12** | 100 | **7.39** |

Table 3 shows runtimes for the numbers of epochs yielding the results in Table 1 for each method. For fair comparison, we implemented T3-Unlearning within the OpenUnlearning codebase using the baseline data-loading logic, performing forward passes through the base model for each batch; we refer to this as *T3-Unlearning (naïve)*. To illustrate the full efficiency potential, we also preprocess a full epoch of data, storing pooled hidden states as input vectors, and record the classifier training time on these saved representations. We report these runtimes as *T3-Unlearning (preprocessed)*, showing that T3-Unlearning can unlearn in a small fraction of the time required by full fine-tuning baselines.

## 5. Conclusion

We introduced the *Temper-Then-Tilt Unlearning* framework (T3-Unlearning), which framed unlearning as density-ratio estimation via probabilistic classification. Our theoretical analysis established finite-sample guarantees for recovering the ground truth unlearned distribution in terms of the Retain and Forget Error metrics. Importantly, we showed that base model tempering was essential for unlearning highly concentrated forget-set distributions. Empirically, T3-Unlearning outperformed existing baselines on the TOFU benchmark; by training only a small linear classifier over frozen representations, it achieved superior forget quality and generative utility with minimal computational cost.

## Acknowledgments

This work was supported in part by NSF Grants 2019844, 2107037, 2112471, and 2505865, ONR Grant N00014-19-1-2566, the Machine Learning Lab (MLL) at UT Austin, the NSF AI Institute for Foundations of Machine Learning (IFML), and the Wireless Networking and Communications Group (WNCG) Industrial Affiliates Program. We are grateful for computing support on the Vista GPU Cluster through the Center for Generative AI (CGAI) and the Texas Advanced Computing Center (TACC) at the University of Texas at Austin.

## Impact Statement

This paper presents work whose goal is to advance the field of Machine Learning. There are many potential societal consequences of our work, none of which we feel must be specifically highlighted here.

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

# A. Proofs

## A.1. Proof of Theorem 3.3

**Theorem.** *Let $\hat{f} \in \mathcal{F}$ satisfy the excess risk bound $L(\hat{f}) - L(f^*) \leq \delta$. Then the Retain Error (6) of the untempered estimate $\hat{p}_r^{(1)}$ in (8) satisfies*

$$\mathcal{E}_r(\hat{p}_r^{(1)}) \leq \frac{\delta}{1-\gamma}.$$

*Proof of Theorem 3.3.* Although we focus on estimating $p_r$ as $\hat{p}_r^{(1)} \propto \hat{f} \cdot p$, we can similarly consider estimating $p_f$ as $\hat{p}_f^{(1)} \propto (1 - \hat{f}) \cdot p$ as a tool for our analysis. Thus, we define the following estimators for $p_r$ and $p_f$ respectively:

$$\hat{p}_r^{(1)}(z) = \frac{p(z)\hat{f}(z)}{\int_u p(u)\hat{f}(u)} \quad \text{and} \quad \hat{p}_f^{(1)}(z) = \frac{p(z)(1 - \hat{f}(z))}{\int_u p(u)(1 - \hat{f}(u))}$$

By construction, $\hat{f}(z)$ estimates the class posterior $f^*(z) = \mathbb{P}\left(S = 1 \mid Z = z\right) = (1 - \gamma)\frac{p_r(z)}{p(z)}$. Define the Bernoulli PMF over the class label $s$ induced by $\hat{f}$ as

$$\hat{\pi}(s \mid z) = \begin{cases} \hat{f}(z) & \text{if } s = 1 \\ 1 - \hat{f}(z) & \text{else,} \end{cases}$$

and let $\pi^*(s \mid z)$ denote the distribution induced by the true class posterior $f^*$. Recall that $\mathbb{P}$ is the base measure over pairs $(z, s)$ defined in Section 2.1. We can then translate the excess risk of $\hat{f}$ directly into a bound on the KL-divergence between the target distribution $p_r$ and our estimator $\hat{p}_r$.

$$
\begin{aligned}
L(\hat{f}) - L(f^*) &= \mathbb{E}_{(Z,S)\sim\mathbb{P}}\left[\ln\frac{\pi^*(S \mid Z)}{\hat{\pi}(S \mid Z)}\right] \\
&= \mathbb{E}_{Z\mid S=1}\left[\ln\frac{\pi^*(S \mid Z)}{\hat{\pi}(S \mid Z)} \mid S = 1\right]\mathbb{P}\left(S = 1\right) + \mathbb{E}_{Z\mid S=0}\left[\ln\frac{\pi^*(S \mid Z)}{\hat{\pi}(S \mid Z)} \mid S = 0\right]\mathbb{P}\left(S = 0\right) \\
&= (1 - \gamma)\mathbb{E}_{Z\sim p_r}\left[\ln\frac{\pi^*(1 \mid Z)}{\hat{\pi}(1 \mid Z)}\right] + \gamma\,\mathbb{E}_{Z\sim p_f}\left[\ln\frac{\pi^*(0 \mid Z)}{\hat{\pi}(0 \mid Z)}\right] \\
&= (1 - \gamma)\mathbb{E}_{Z\sim p_r}\left[\ln\left(\frac{f^*(Z)}{\hat{f}(Z)}\right)\right] + \gamma\,\mathbb{E}_{Z\sim p_f}\left[\ln\left(\frac{1 - f^*(Z)}{1 - \hat{f}(Z)}\right)\right] \\
&= (1 - \gamma)\mathbb{E}_{Z\sim p_r}\left[\ln\left(\frac{(1-\gamma)\frac{p_r(Z)}{p(Z)}}{\frac{\hat{p}_r^{(1)}(Z)}{p(Z)}\int_u p(u)\hat{f}(u)}\right)\right] + \gamma\,\mathbb{E}_{Z\sim p_f}\left[\ln\left(\frac{\gamma\frac{p_f(Z)}{p(Z)}}{\frac{\hat{p}_f^{(1)}(Z)}{p(Z)}\int_u p(u)(1 - \hat{f}(u))}\right)\right] \\
&= (1 - \gamma)\mathrm{KL}\left(p_r \,\|\, \hat{p}_r^{(1)}\right) + \gamma\mathrm{KL}\left(p_f \,\|\, \hat{p}_f^{(1)}\right) - (1 - \gamma)\ln\int_u \frac{p(u)\hat{f}(u)}{1-\gamma} - \gamma\ln\int_u \frac{p(u)(1 - \hat{f}(u))}{\gamma} \\
&\geq (1 - \gamma)\mathrm{KL}\left(p_r \,\|\, \hat{p}_r^{(1)}\right) + \gamma\mathrm{KL}\left(p_f \,\|\, \hat{p}_f^{(1)}\right) - \ln\left(\int_u p(u)\hat{f}(u) + p(u)(1 - \hat{f}(u))\right) \\
&= (1 - \gamma)\mathrm{KL}\left(p_r \,\|\, \hat{p}_r^{(1)}\right) + \gamma\mathrm{KL}\left(p_f \,\|\, \hat{p}_f^{(1)}\right) - \ln\int_u p(u) \\
&= (1 - \gamma)\mathrm{KL}\left(p_r \,\|\, \hat{p}_r^{(1)}\right) + \gamma\mathrm{KL}\left(p_f \,\|\, \hat{p}_f^{(1)}\right)
\end{aligned}
$$

Note that the sole inequality above follows from the convexity of $-\ln(\cdot)$. Thus,

$$(1 - \gamma)\mathrm{KL}\left(p_r \,\|\, \hat{p}_r^{(1)}\right) + \gamma\mathrm{KL}\left(p_f \,\|\, \hat{p}_f^{(1)}\right) \leq L(\hat{f}) - L(f^*) \leq \delta$$

Since KL divergence is non-negative, we have that

$$\mathcal{E}_r(\hat{p}_r^{(1)}) := \mathrm{KL}\left(p_r \,\|\, \hat{p}_r^{(1)}\right) \leq \frac{\delta}{1-\gamma}.$$

$\square$

## A.2. Proof of Theorem 3.4

**Theorem.** *Let $\hat{f} \in \mathcal{F}$ satisfy the excess risk bound $L(\hat{f}) - L(f^*) \leq \delta$. Then the Forget Error (7) of the untempered estimate $\hat{p}_r^{(1)}$ in (8) satisfies*

$$\mathcal{E}_f(\hat{p}_r^{(1)}) \leq \|p_f\|_\infty \sqrt{\frac{2\delta}{1-\gamma}}.$$

*Proof of Theorem 3.4.* We can immediately relate the $\ell_1$ error to the KL divergence bound in Theorem 3.3 using Pinsker's inequality.

$$
\begin{aligned}
\mathcal{E}_f(\hat{p}_r^{(1)}) := \mathbb{E}_{\boldsymbol{Z} \sim p_f}\left[\left|p_r(\boldsymbol{Z}) - \hat{p}_r^{(1)}(\boldsymbol{Z})\right|\right] = \int_{\boldsymbol{z}} p_f(\boldsymbol{z}) \left|p_r(\boldsymbol{z}) - \hat{p}_r^{(1)}(\boldsymbol{z})\right| & \\
\leq \|p_f\|_\infty \left\|p_r - \hat{p}_r^{(1)}\right\|_1 & \\
\leq \|p_f\|_\infty \sqrt{2 \cdot \mathrm{KL}\left(p_r \,\|\, \hat{p}_r^{(1)}\right)} \qquad & \text{(Pinsker's)} \\
\leq \|p_f\|_\infty \sqrt{\frac{2\delta}{(1-\gamma)}} \qquad & \text{(Theorem 3.3).}
\end{aligned}
$$

$\square$

## A.3. Proof of Theorem 3.5

**Theorem.** *Let $p_r$ and $p_f$ have disjoint supports, with $p_f$ uniform on its support. Then for any mixture weight $\gamma \in (0,1)$ and any $\delta > 0$, there exists a classifier $\hat{f}$ achieving excess risk $L(\hat{f}) - L(f^*) \leq \delta$ such that the Forget Error (7) of the untempered estimate $\hat{p}_r^{(1)}$ in (8) satisfies*

$$\mathcal{E}_f(\hat{p}_r^{(1)}) \geq \|p_f\|_\infty \cdot \frac{\gamma\left(1 - \exp\left(-\frac{\delta}{\gamma}\right)\right)}{1 - \gamma\exp\left(-\frac{\delta}{\gamma}\right)}.$$

*In particular, as $\delta \to 0$, we have $\mathcal{E}_f(\hat{p}_r^{(1)}) = \Omega\left(\|p_f\|_\infty \cdot \delta\right)$.*

*Proof of Theorem 3.5.* Let $\mathcal{Z}_r = \mathrm{supp}(p_r)$ and $\mathcal{Z}_f = \mathrm{supp}(p_f)$ where $\mathcal{Z}_r \cap \mathcal{Z}_f = \emptyset$. Define $p_f$ as the uniform distribution over $\mathcal{Z}_f$, while $p_r$ can be any arbitrary density supported on $\mathcal{Z}_r$.

We first construct a witness classifier $\hat{f}$ that satisfies the risk bound $L(\hat{f}) - L(f^*) \leq \delta$ but maximizes the unlearning error. Define $\hat{f}$ as:

$$\hat{f}(\boldsymbol{z}) = \begin{cases} 1 & \text{if } \boldsymbol{z} \in \mathcal{Z}_r \\ \epsilon & \text{if } \boldsymbol{z} \in \mathcal{Z}_f \end{cases} \tag{9}$$

where $\epsilon \in (0,1)$ is a constant scalar. Thus, $\hat{f}$ achieves perfect performance over the retain component support $\mathcal{Z}_r$ but has $\epsilon$ error over $\mathcal{Z}_f$. Note that the optimal classifier $f^*(\boldsymbol{z})$ is the indicator function:

$$f^*(\boldsymbol{z}) = \mathbb{1}\{\boldsymbol{z} \in \mathcal{Z}_r\}.$$

Thus, we compute the excess risk of $\hat{f}$ using the decomposition in the proof of Theorem 3.3 in Appendix A.1:

$$
\begin{aligned}
L(\hat{f}) - L(f^*) &= (1-\gamma)\mathbb{E}_{\boldsymbol{Z} \sim p_r}\left[\ln\left(\frac{f^*(\boldsymbol{Z})}{\hat{f}(\boldsymbol{Z})}\right)\right] + \gamma\,\mathbb{E}_{\boldsymbol{Z} \sim p_f}\left[\ln\left(\frac{1-f^*(\boldsymbol{Z})}{1-\hat{f}(\boldsymbol{Z})}\right)\right] \\
&= \gamma\mathbb{E}_{\boldsymbol{Z} \sim p_f}[-\ln(1-\hat{f}(\boldsymbol{Z}))] \\
&= -\gamma\ln(1-\epsilon).
\end{aligned}
$$

We saturate the risk bound by setting $-\gamma \ln(1 - \epsilon) = \delta$, implying that $\epsilon = 1 - \exp\left(-\frac{\delta}{\gamma}\right)$.

For our estimator $\hat{p}_r$, let $N$ denote the partition function:

$$N = \int_{\boldsymbol{u}} p(\boldsymbol{u})\, \hat{f}(\boldsymbol{u}).$$

Note that for any $\boldsymbol{z}_f \in \mathcal{Z}_f$ we have that $p(\boldsymbol{z}_f) = \gamma p_f(\boldsymbol{z}_f)$ and $p_r(\boldsymbol{z}_f) = 0$. The Forget Error is then

$$\mathcal{E}_f(\hat{p}_r^{(1)}) := \mathbb{E}_{\boldsymbol{Z} \sim p_f}\left[\left|p_r(\boldsymbol{Z}) - \hat{p}_r^{(1)}(\boldsymbol{Z})\right|\right] = \int_{\boldsymbol{z} \in \mathcal{Z}_f} p_f(\boldsymbol{z})\left|0 - \frac{\gamma p_f(\boldsymbol{z})\epsilon}{N}\right| = \|p_f\|_2^2 \cdot \frac{\gamma\epsilon}{N}.$$

Since $p_f$ is uniform over $\mathcal{Z}_f$, its squared $\ell_2$-norm coincides with its peak value, which is the inverse of the support volume:

$$\|p_f\|_2^2 = \|p_f\|_\infty = \mathrm{Vol}(\mathcal{Z}_f)^{-1}.$$

Thus,

$$\mathcal{E}_f(\hat{p}_r) = \|p_f\|_\infty \cdot \frac{\gamma\epsilon}{N}. \tag{10}$$

We lastly compute $N$ exactly using the definition of $\hat{f}$ in (9):

$$
\begin{aligned}
N &= \int_{\boldsymbol{z}} p(\boldsymbol{z})\hat{f}(\boldsymbol{z}) \\
&= \int_{\boldsymbol{z} \in \mathcal{Z}_r} (1 - \gamma)p_r(\boldsymbol{z})\hat{f}(\boldsymbol{z}) + \int_{\boldsymbol{z} \in \mathcal{Z}_f} \gamma p_f(\boldsymbol{z})\hat{f}(\boldsymbol{z}) \\
&= \int_{\boldsymbol{z} \in \mathcal{Z}_r} (1 - \gamma)p_r(\boldsymbol{z}) + \int_{\boldsymbol{z} \in \mathcal{Z}_f} \gamma p_f(\boldsymbol{z})\epsilon \\
&= (1 - \gamma) + \gamma\epsilon
\end{aligned}
$$

Substituting $N = (1 - \gamma) + \gamma\epsilon$ and $\epsilon = 1 - \exp\left(-\frac{\delta}{\gamma}\right)$ into the expression in (10):

$$
\begin{aligned}
\mathcal{E}_f(\hat{p}_r^{(1)}) &\geq \|p_f\|_\infty \cdot \frac{\gamma\left(1 - \exp\left(-\frac{\delta}{\gamma}\right)\right)}{(1 - \gamma) + \gamma\left(1 - \exp\left(-\frac{\delta}{\gamma}\right)\right)} \\
&= \|p_f\|_\infty \cdot \frac{\gamma\left(1 - \exp\left(-\frac{\delta}{\gamma}\right)\right)}{1 - \gamma\exp\left(-\frac{\delta}{\gamma}\right)}
\end{aligned}
$$

This establishes the main theorem statement. We now analyze how this lower bound scales as $\delta \to 0$. Using the Taylor expansion $e^x = 1 + x + \mathcal{O}(x^2)$ and substituting $x = -\delta/\gamma$, we get:

$$\exp\left(-\frac{\delta}{\gamma}\right) = 1 - \frac{\delta}{\gamma} + \mathcal{O}(\delta^2).$$

Substituting this expansion:

$$
\begin{aligned}
\|p_f\|_\infty \cdot \frac{\gamma\left(1 - \exp\left(-\frac{\delta}{\gamma}\right)\right)}{1 - \gamma\exp\left(-\frac{\delta}{\gamma}\right)} &= \|p_f\|_\infty \cdot \frac{\gamma\left(1 - \left(1 - \frac{\delta}{\gamma} + \mathcal{O}(\delta^2)\right)\right)}{1 - \gamma\left(1 - \frac{\delta}{\gamma} + \mathcal{O}(\delta^2)\right)} \\
&= \|p_f\|_\infty \cdot \frac{\gamma\left(\frac{\delta}{\gamma} - \mathcal{O}(\delta^2)\right)}{1 - \gamma + \delta - \mathcal{O}(\delta^2)} \\
&= \|p_f\|_\infty \cdot \frac{\delta + \mathcal{O}(\delta^2)}{(1 - \gamma) + \delta + \mathcal{O}(\delta^2)}.
\end{aligned}
$$

The term $(1 - \gamma)$ is a non-zero constant, so as $\delta \to 0$, the higher-order terms and the $\delta$ in the denominator become negligible compared to $(1 - \gamma)$. Further since $(1 - \gamma) > 0$ is fixed and independent of $\delta$, we obtain:

$$\mathcal{E}_f(\hat{p}_r^{(1)}) = \Omega\left(\|p_f\|_\infty \cdot \delta\right).$$

$\square$

## A.4. Supporting Lemmas

Before proving the remaining theorem statements, we prove two key lemmas.

**Lemma A.1** (Classifier $\ell_1$ Error Bound). *Let $\hat{f} \in \mathcal{F}$ satisfy the population risk bound $L(\hat{f}) - L(f^*) \leq \delta$. Then,*

$$\mathbb{E}_{\boldsymbol{z} \sim p}\left[\left|f^*(\boldsymbol{z}) - \hat{f}(\boldsymbol{z})\right|\right] \leq \sqrt{\frac{\delta}{2}}.$$

*Proof of Lemma A.1.* We identify $\left|\hat{f}(\boldsymbol{z}) - f^*(\boldsymbol{z})\right|$ as the total variation distance between the Bernoulli random variables over the conditional class label $s \mid \boldsymbol{z}$ induced by $\hat{f}$ and $f^*$. Define the Bernoulli PMF over the class label $s$ induced by $\hat{f}$ as

$$\hat{\pi}(s \mid \boldsymbol{z}) = \begin{cases} \hat{f}(\boldsymbol{z}) & \text{if } s = 1 \\ 1 - \hat{f}(\boldsymbol{z}) & \text{else,} \end{cases}$$

and let $\pi^*(s \mid \boldsymbol{z})$ denote the distribution induced by the true class posterior $f^*$. Then,

$$\begin{aligned}
\mathbb{E}_{\boldsymbol{Z} \sim p}\left[\left|\hat{f}(\boldsymbol{Z}) - f^*(\boldsymbol{Z})\right|\right] &= \mathbb{E}_{\boldsymbol{Z} \sim p}\left[D_{\text{TV}}\left(\pi^*(\cdot \mid \boldsymbol{Z}), \hat{\pi}(\cdot \mid \boldsymbol{Z})\right)\right] \\
&\leq \mathbb{E}_{\boldsymbol{Z} \sim p}\left[\sqrt{\frac{1}{2}\text{KL}\left(\pi^*(\cdot \mid \boldsymbol{Z}) \,\|\, \hat{\pi}(\cdot \mid \boldsymbol{Z})\right)}\right] && \text{(Pinsker's)} \\
&\leq \sqrt{\mathbb{E}_{\boldsymbol{Z} \sim p}\left[\frac{1}{2}\text{KL}\left(\pi^*(\cdot \mid \boldsymbol{Z}) \,\|\, \hat{\pi}(\cdot \mid \boldsymbol{Z})\right)\right]} && \text{(Jensen's)} \\
&= \sqrt{\frac{1}{2}\left(L(\hat{f}) - L(f^*)\right)} \\
&\leq \sqrt{\frac{\delta}{2}}
\end{aligned}$$

$\square$

**Lemma A.2** (Partition Function Lower Bound). *Let $\hat{f} \in \mathcal{F}$ satisfy the population risk bound $L(\hat{f}) - L(f^*) \leq \delta$. Then,*

$$\int_{\boldsymbol{z}} p(\boldsymbol{z})^{1/T}\hat{f}(\boldsymbol{z}) \geq (1 - \gamma)^{\frac{T+1}{T}} \exp\left(\frac{T-1}{T}H(p_r) - \frac{\delta - \gamma \ln \gamma}{1 - \gamma}\right)$$

*Proof of Lemma A.2.* We first lower bound the mixture density $p = (1 - \gamma)p_r + \gamma p_f$ by just the retain component $(1 - \gamma)p_r$:

$$\begin{aligned}
\int_{\boldsymbol{z}} p(\boldsymbol{z})^{1/T}\hat{f}(\boldsymbol{z}) &\geq \int_{\boldsymbol{z}}\left((1 - \gamma)p_r(\boldsymbol{z})\right)^{1/T}\hat{f}(\boldsymbol{z}) \\
&= (1 - \gamma)^{1/T}\int_{\boldsymbol{z}} p_r(\boldsymbol{z})^{1/T}\hat{f}(z) \\
&= (1 - \gamma)^{1/T}\mathbb{E}_{\boldsymbol{z} \sim p_r}\left[p_r(\boldsymbol{z})^{\frac{1-T}{T}}\hat{f}(\boldsymbol{z})\right]. && (11)
\end{aligned}$$

Since the logarithm is concave, by Jensen's Inequality:

$$
\begin{aligned}
\mathbb{E}_{\mathbf{Z}\sim p_r}\left[p_r(\mathbf{Z})^{\frac{1-T}{T}}\hat{f}(\mathbf{Z})\right] &= \exp\left(\ln\mathbb{E}_{\mathbf{Z}\sim p_r}\left[p_r(\mathbf{Z})^{\frac{1-T}{T}}\hat{f}(\mathbf{z})\right]\right) \\
&\geq \exp\left(\mathbb{E}_{\mathbf{Z}\sim p_r}\left[\ln\left(p_r(\mathbf{Z})^{\frac{1-T}{T}}\hat{f}(\mathbf{Z})\right)\right]\right) \\
&= \exp\left(\mathbb{E}_{\mathbf{Z}\sim p_r}\left[\frac{1-T}{T}\ln p_r(\mathbf{Z}) + \ln\hat{f}(\mathbf{Z})\right]\right) \\
&= \exp\left(\frac{T-1}{T}H(p_r) + \mathbb{E}_{\mathbf{Z}\sim p_r}\left[\ln\hat{f}(\mathbf{z})\right]\right).
\end{aligned}
\tag{12}
$$

Then we can control the final term $\mathbb{E}_{\mathbf{Z}\sim p_r}\left[\ln\hat{f}(\mathbf{Z})\right]$ using the excess risk bound $L(\hat{f}) - L(f^*) \leq \delta$. From the total risk bound, we have the intermediate bound (See Appendix A.1):

$$
(1-\gamma)\mathbb{E}_{\mathbf{Z}\sim p_r}\left[\ln\left(\frac{f^*(\mathbf{Z})}{\hat{f}(\mathbf{Z})}\right)\right] + \gamma\,\mathbb{E}_{\mathbf{Z}\sim p_f}\left[\ln\left(\frac{1-f^*(\mathbf{Z})}{1-\hat{f}(\mathbf{Z})}\right)\right] \leq \delta
$$

We lower bound the second term as follows:

$$
\begin{aligned}
\gamma\,\mathbb{E}_{\mathbf{Z}\sim p_f}\left[\ln\left(\frac{1-f^*(\mathbf{Z})}{1-\hat{f}(\mathbf{Z})}\right)\right] &= \gamma\,\mathbb{E}_{\mathbf{Z}\sim p_f}\left[\ln\left(\frac{\gamma\frac{p_f(\mathbf{Z})}{p(\mathbf{Z})}}{\frac{\hat{p}_f(\mathbf{Z})}{p(\mathbf{Z})}\int_{\mathbf{u}}p(\mathbf{u})(1-\hat{f}(\mathbf{u}))}\right)\right] \\
&= \gamma\left(\mathrm{KL}\left(p_f \,\|\, \hat{p}_f\right) - \ln\int_{\mathbf{u}}\frac{p(\mathbf{u})(1-\hat{f}(\mathbf{u}))}{\gamma}\right) \\
&\geq -\gamma\ln\int_{\mathbf{u}}\frac{p(\mathbf{u})(1-\hat{f}(\mathbf{u}))}{\gamma} \\
&\geq -\gamma\ln\int_{\mathbf{u}}\frac{p(\mathbf{u})}{\gamma} \\
&= \gamma\ln\gamma.
\end{aligned}
$$

The first inequality follows from the non-negativity of KL divergence, and the second comes from the fact that $\hat{f}(\mathbf{u}) \in [0,1]$. Thus,

$$
(1-\gamma)\mathbb{E}_{\mathbf{Z}\sim p_r}\left[\ln\frac{f^*(\mathbf{Z})}{\hat{f}(\mathbf{Z})}\right] \leq \delta - \gamma\ln\gamma
$$

Rearranging terms,

$$
\begin{aligned}
\mathbb{E}_{\mathbf{Z}\sim p_r}\left[\ln\hat{f}(\mathbf{Z})\right] &\geq -\frac{\delta - \gamma\ln\gamma}{1-\gamma} + \mathbb{E}_{\mathbf{Z}\sim p_r}\left[\ln f^*(\mathbf{Z})\right] \\
&= -\frac{\delta - \gamma\ln\gamma}{1-\gamma} + \mathbb{E}_{\mathbf{Z}\sim p_r}\left[\ln\left((1-\gamma)\frac{p_r(\mathbf{Z})}{p(\mathbf{Z})}\right)\right] \\
&= -\frac{\delta - \gamma\ln\gamma}{1-\gamma} + \ln(1-\gamma) + \mathrm{KL}\left(p_r \,\|\, p\right) \\
&\geq -\frac{\delta - \gamma\ln\gamma}{1-\gamma} + \ln(1-\gamma).
\end{aligned}
\tag{13}
$$

Substituting the bound in (13) for (12) gives that

$$
\mathbb{E}_{\mathbf{Z}\sim p_r}\left[p_r(\mathbf{Z})^{\frac{1-T}{T}}\hat{f}(\mathbf{Z})\right] \geq (1-\gamma)\exp\left(\frac{T-1}{T}H(p_r) - \frac{\delta - \gamma\ln\gamma}{1-\gamma}\right)
$$

Then by (11),

$$\int_{\boldsymbol{z}} p(\boldsymbol{z})^{1/T} \hat{f}(\boldsymbol{z}) \geq (1-\gamma)^{\frac{T+1}{T}} \exp\left(\frac{T-1}{T} H(p_r) - \frac{\delta - \gamma \ln \gamma}{1-\gamma}\right)$$

$\square$

### A.5. Proof of Theorem 3.7

**Theorem.** *Let $\hat{f} \in \mathcal{F}$ satisfy the excess risk bound $L(\hat{f}) - L(f^*) \leq \delta$. Define the $\tau$-tempered oracle estimate $p_r^{(\tau)} \propto p^{1/\tau} \cdot f^*$ for $\tau \in [1, T]$. Consider $k \geq T$ such that*

$$\int p^{\frac{k-T}{T(k-1)}} < \infty.$$

*Then for some $\tau \in [1, T]$, the Forget Error (7) of the $T$-tempered estimate $\hat{p}_r^{(T)}$ in (8) satisfies*

$\mathcal{E}_f(\hat{p}_r^{(T)}) \leq$

$$\left(1 - \tfrac{1}{T}\right) \|p_f\|_{2, p_r^{(\tau)}} \cdot \mathrm{Std}_{p_r^{(\tau)}}[\ln p] + \frac{\|p_f\|_\infty^{1/T} (\delta/2)^{\frac{1}{2T}}}{(1-\gamma)^{\frac{T+1}{T}} \exp\left(\frac{T-1}{T} H(p_r) + \frac{\gamma \ln \gamma}{1-\gamma}\right)} + \frac{\|p_f\|_\infty^{1/T} \left(\int_{\boldsymbol{z}} p(\boldsymbol{z})^{\frac{k-T}{T(k-1)}}\right)^{\frac{k-1}{k}} (\delta/2)^{\frac{1}{2k}}}{(1-\gamma)^{\frac{2T+2}{T}} \exp\left(\frac{2T-2}{T} H(p_r) - \frac{\delta - 2\gamma \ln \gamma}{1-\gamma}\right)}$$

*Proof of Theorem 3.7.* Define the ground truth $T$-tempered density $p_r^{(T)}(\boldsymbol{z}) \propto p(\boldsymbol{z})^{1/T} \cdot f^*(\boldsymbol{z})$. We then decompose the Forget Error into a tempering bias and estimation error.

$$\mathcal{E}_f(\hat{p}_r^{(T)}) := \mathbb{E}_{\boldsymbol{Z} \sim p_f}\left[\left|p_r(\boldsymbol{Z}) - \hat{p}_r^{(T)}(\boldsymbol{Z})\right|\right] = \mathbb{E}_{\boldsymbol{Z} \sim p_f}\left[\left|p_r(\boldsymbol{Z}) - p_r^{(T)}(\boldsymbol{Z}) + p_r^{(T)}(\boldsymbol{Z}) - \hat{p}_r^{(T)}(\boldsymbol{Z})\right|\right]$$

$$\leq \underbrace{\mathbb{E}_{\boldsymbol{Z} \sim p_f}\left[\left|p_r(\boldsymbol{Z}) - p_r^{(T)}(\boldsymbol{Z})\right|\right]}_{\text{Tempering Bias}} + \underbrace{\mathbb{E}_{\boldsymbol{Z} \sim p_f}\left[\left|p_r^{(T)}(\boldsymbol{Z}) - \hat{p}_r^{(T)}(\boldsymbol{Z})\right|\right]}_{\text{Estimation Error}} \quad (14)$$

We first bound the tempering bias term. Using the fact that $f^* \propto p_r/p$, we have that

$$p_r^{(T)}(\boldsymbol{z}) \propto p_r(\boldsymbol{z}) p(\boldsymbol{z})^{\frac{1}{T}-1}.$$

Rather than working with the temperature indexed family $p_r^{(\cdot)}$ directly, we define a family of distributions $q_\alpha(\boldsymbol{z})$ indexed by the inverse temperature $\alpha \in [1/T, 1]$, which interpolates between the tempered estimator and the target distribution:

$$q_\alpha(\boldsymbol{z}) := p_r^{(1/\alpha)} = \frac{p_r(\boldsymbol{z}) p(\boldsymbol{z})^{\alpha-1}}{N(\alpha)}, \quad \text{where } N(\alpha) = \int_{\boldsymbol{u}} p_r(\boldsymbol{u}) p(\boldsymbol{u})^{\alpha-1}.$$

Note that at the endpoints of our interval, we recover our distributions of interest:

$$q_{1/T}(\boldsymbol{z}) = p_r^{(T)}(\boldsymbol{z}) \quad \text{and} \quad q_1(\boldsymbol{z}) = p_r(\boldsymbol{z}).$$

We first compute the derivative of the density $q_\alpha(\boldsymbol{z})$ with respect to $\alpha$:

$$\frac{\partial q_\alpha(\boldsymbol{z})}{\partial \alpha} = q_\alpha(\boldsymbol{z}) \frac{\partial \ln q_\alpha(\boldsymbol{z})}{\partial \alpha} = q_\alpha(\boldsymbol{z}) \left(\ln p(\boldsymbol{z}) - \mathbb{E}_{\boldsymbol{u} \sim q_\alpha}[\ln p(\boldsymbol{u})]\right).$$

By the Fundamental Theorem of Calculus, we can express the difference between the target and the estimator as the integral of this derivative:

$$p_r(\boldsymbol{z}) - p_r^{(T)}(\boldsymbol{z}) = \int_{1/T}^1 \frac{\partial q_\alpha(\boldsymbol{z})}{\partial \alpha} d\alpha.$$

We substitute this into our error metric $\mathcal{E}_f(\hat{p}_r^{(T)})$:

$$\mathcal{E}_f(\hat{p}_r^{(T)}) := \mathbb{E}_{\boldsymbol{Z} \sim p_f}[|p_r(\boldsymbol{Z}) - p_r^{(T)}(\boldsymbol{Z})|] = \int_{\boldsymbol{z}} \left(p_f(\boldsymbol{z}) \left|\int_{1/T}^1 \frac{\partial q_\alpha(\boldsymbol{z})}{\partial \alpha} d\alpha\right|\right)$$

$$\leq \int_{\boldsymbol{z}} \left(p_f(\boldsymbol{z}) \int_{1/T}^1 \left|\frac{\partial q_\alpha(\boldsymbol{z})}{\partial \alpha}\right| d\alpha\right).$$

We apply Fubini's Theorem to swap the order of integration:

$$\mathcal{E}_f(\hat{p}_r^{(T)}) \le \int_{1/T}^1 \underbrace{\left( \int_{\boldsymbol{z}} p_f(\boldsymbol{z}) \left| \frac{\partial q_\alpha(\boldsymbol{z})}{\partial \alpha} \right| \right)}_{M(\alpha)} d\alpha.$$

Here, $M(\alpha)$ is a scalar function representing the instantaneous error rate at temperature $\alpha$. By the Mean Value Theorem for Integrals, there exists a fixed $\alpha^* \in [1/T, 1]$ such that:

$$\int_{1/T}^1 M(\alpha) d\alpha = \left( 1 - \frac{1}{T} \right) M(\alpha^*).$$

Substituting the definition of the derivative back into $M(\alpha^*)$, we obtain:

$$\mathcal{E}_f(\hat{p}_r^{(T)}) \le \left( 1 - \frac{1}{T} \right) \int_{\boldsymbol{z}} p_f(\boldsymbol{z}) q_{\alpha^*}(\boldsymbol{z}) \left| \ln p(\boldsymbol{z}) - \mathbb{E}_{q_{\alpha^*}}[\ln p] \right|.$$

Let $\tau = 1/\alpha^*$ be the corresponding temperature. To interpret the resulting bound, we apply Hölder's inequality:

$$\begin{aligned}
\mathcal{E}_f(\hat{p}_r^{(T)}) &\le \left( 1 - \frac{1}{T} \right) \int_{\boldsymbol{z}} p_f(\boldsymbol{z}) p_r^{(\tau)}(\boldsymbol{z}) \left| \ln p(\boldsymbol{z}) - \mathbb{E}_{\boldsymbol{U} \sim p_r^{(\tau)}}[\ln p(\boldsymbol{U})] \right| \\
&= \left( 1 - \frac{1}{T} \right) \int_{\boldsymbol{z}} \left( p_f(\boldsymbol{z}) \sqrt{p_r^{(\tau)}(\boldsymbol{z})} \right) \left( \sqrt{p_r^{(\tau)}(\boldsymbol{z})} \left| \ln p(\boldsymbol{z}) - \mathbb{E}_{\boldsymbol{U} \sim p_r^{(\tau)}}[\ln p(\boldsymbol{U})] \right| \right) \\
&\le \left( 1 - \frac{1}{T} \right) \|p_f\|_{2, p_r^{(\tau)}} \cdot \mathrm{Std}_{\boldsymbol{Z} \sim p_r^{(\tau)}}[\ln p(\boldsymbol{Z})],
\end{aligned}$$

where $\|p_f\|_{2, p_r^{(\tau)}} = (\int p_r^{(\tau)} p_f^2)^{1/2}$.

We now bound the estimation error. Define the partition functions:

$$N_T = \int_{\boldsymbol{z}} p(\boldsymbol{z})^{1/T} \hat{f}(\boldsymbol{z}) \qquad \text{and} \qquad N_T^* = \int_{\boldsymbol{z}} p(\boldsymbol{z})^{1/T} f^*(\boldsymbol{z})$$

We then expand the estimation error term using the triangle inequality:

$$\begin{aligned}
\mathbb{E}_{\boldsymbol{Z} \sim p_f(\boldsymbol{z})} &\left[ \left| p_r^{(T)}(\boldsymbol{Z}) - \hat{p}_r^{(T)}(\boldsymbol{Z}) \right| \right] \\
&= \int_{\boldsymbol{z}} p_f(\boldsymbol{z}) \left| p_r^{(T)}(\boldsymbol{z}) - \hat{p}_r^{(T)}(\boldsymbol{z}) \right| \\
&= \int_{\boldsymbol{z}} p_f(\boldsymbol{z}) \left| \frac{p(\boldsymbol{z})^{1/T} f^*(\boldsymbol{z})}{N_T^*} - \frac{p(\boldsymbol{z})^{1/T} \hat{f}(\boldsymbol{z})}{N_T} \right| \\
&= \int_{\boldsymbol{z}} p_f(\boldsymbol{z}) \left| \frac{p(\boldsymbol{z})^{1/T} f^*(\boldsymbol{z})}{N_T^*} - \frac{p(\boldsymbol{z})^{1/T} \hat{f}(\boldsymbol{z})}{N_T^*} + \frac{p(\boldsymbol{z})^{1/T} \hat{f}(\boldsymbol{z})}{N_T^*} - \frac{p(\boldsymbol{z})^{1/T} \hat{f}(\boldsymbol{z})}{N_T} \right| \\
&\le \frac{1}{N_T^*} \int_{\boldsymbol{z}} p_f(\boldsymbol{z}) \left| p(\boldsymbol{z})^{1/T} f^*(\boldsymbol{z}) - p(\boldsymbol{z})^{1/T} \hat{f}(\boldsymbol{z}) \right| + \int_{\boldsymbol{z}} p_f(\boldsymbol{z}) p(\boldsymbol{z})^{1/T} \hat{f}(\boldsymbol{z}) \left| \frac{1}{N_T^*} - \frac{1}{N_T} \right| \\
&= \frac{1}{N_T^*} \int_{\boldsymbol{z}} p_f(\boldsymbol{z}) \left| p(\boldsymbol{z})^{1/T} f^*(\boldsymbol{z}) - p(\boldsymbol{z})^{1/T} \hat{f}(\boldsymbol{z}) \right| + \int_{\boldsymbol{z}} p_f(\boldsymbol{z}) p(\boldsymbol{z})^{1/T} \hat{f}(\boldsymbol{z}) \frac{|N_T^* - N_T|}{N_T^* N_T}. \quad (15)
\end{aligned}$$

Using Lemma A.2, we can immediately lower bound the partition functions:

$$N_T^* \ge (1 - \gamma)^{\frac{T+1}{T}} \exp\left( \frac{T-1}{T} H(p_r) + \frac{\gamma \ln \gamma}{1 - \gamma} \right) \quad (16)$$

$$N_T \ge (1 - \gamma)^{\frac{T+1}{T}} \exp\left( \frac{T-1}{T} H(p_r) - \frac{\delta - \gamma \ln \gamma}{1 - \gamma} \right). \quad (17)$$

Thus, we must control the remaining two integrals as well as the partition function difference $N_T^* - N_T$. Define the integrals:

$$I_1 = \int_z p_f(z) \left| p(z)^{1/T} f^*(z) - p(z)^{1/T} \hat{f}(z) \right|$$

$$I_2 = \int_z p_f(z) p(z)^{1/T} \hat{f}(z).$$

We first upper bound $I_1$ by applying Hölder's inequality using conjugate pairs $\frac{T}{T-1}$ and $T$:

$$I_1 = \int_z p_f(z) p(z)^{1/T} \left| f^*(z) - \hat{f}(z) \right| \leq \|p_f(z)\|_{\frac{T}{T-1}} \cdot \left\| p^{1/T} \cdot |f^* - \hat{f}| \right\|_T$$

$$= \left( \int_z p_f(z)^{\frac{T}{T-1}} \right)^{\frac{T-1}{T}} \left( \int_z p(z) \left| f^*(z) - \hat{f}(z) \right|^T \right)^{\frac{1}{T}}$$

$$= \left( \int_z p_f(z) \cdot p_f(z)^{\frac{1}{T-1}} \right)^{\frac{T-1}{T}} \left( \int_z p(z) \left| f^*(z) - \hat{f}(z) \right|^T \right)^{\frac{1}{T}}$$

$$\leq \left( \|p_f\|_\infty^{\frac{1}{T-1}} \int_z p_f(z) \right)^{\frac{T-1}{T}} \left( \int_z p(z) \left| f^*(z) - \hat{f}(z) \right| \right)^{\frac{1}{T}}$$

$$= \|p_f\|_\infty^{1/T} \mathbb{E}_{z \sim p} \left[ \left| f^*(z) - \hat{f}(z) \right| \right]^{\frac{1}{T}}$$

$$\leq \|p_f\|_\infty^{1/T} \left( \frac{\delta}{2} \right)^{\frac{1}{2T}} \qquad \text{(Lemma A.1)}$$

The second inequality follows from the fact that $f^*(z), \hat{f}(z) \in [0,1]$, so $|f^*(z) - \hat{f}(z)| \in [0,1]$. We bound $I_2$ using a similar technique:

$$I_2 = \int_z p_f(z) p(z)^{1/T} \hat{f}(z) \leq \int_z p_f(z) p(z)^{1/T}$$

$$\leq \|p_f\|_{\frac{T}{T-1}} \cdot \|p^{1/T}\|_T$$

$$= \left( \int_z p_f^{\frac{T}{T-1}}(z) \right)^{\frac{T-1}{T}} \left( \int_z p(z) \right)^{\frac{1}{T}}$$

$$= \left( \int_z p_f(z) \, p_f(z)^{\frac{1}{T-1}} \right)^{\frac{T-1}{T}}$$

$$\leq \left( \|p_f\|_\infty^{\frac{1}{T-1}} \int_z p_f(z) \right)^{\frac{T-1}{T}}$$

$$= \|p_f\|_\infty^{1/T}$$

We lastly bound the difference in partition functions $|N_T^* - N_T|$. Expanding their definitions:

$$|N_T^* - N_T| = \left| \int_z p(z)^{1/T} f^*(z) - \int_z p(z)^{1/T} \hat{f}(z) \right|$$

$$\leq \int_z p(z)^{1/T} \left| \hat{f}(z) - f^*(z) \right|$$

Applying Hölder's inequality with conjugate exponents $q = \frac{k}{k-1}$ and $q' = k$ for some $k \geq T$:

$$
\begin{aligned}
\int_{\mathbf{z}} p(\mathbf{z})^{1/T} \left| \hat{f}(\mathbf{z}) - f^*(\mathbf{z}) \right| &= \int_{\mathbf{z}} p(\mathbf{z})^{\frac{k-T}{kT}} \left( p(\mathbf{z})^{\frac{1}{k}} \left| \hat{f}(\mathbf{z}) - f^*(\mathbf{z}) \right| \right) \\
&\leq \left\| p^{\frac{k-T}{kT}} \right\|_{\frac{k}{k-1}} \cdot \left\| p^{1/k} \cdot |\hat{f} - f^*| \right\|_k \\
&= \left( \int_{\mathbf{z}} p(\mathbf{z})^{\frac{k-T}{kT} \cdot \frac{k}{k-1}} \right)^{\frac{k-1}{k}} \left( \int_{\mathbf{z}} p(\mathbf{z}) \left| \hat{f}(\mathbf{z}) - f^*(\mathbf{z}) \right|^k \right)^{1/k} \\
&\leq \left( \int_{\mathbf{z}} p(\mathbf{z})^{\frac{k-T}{T(k-1)}} \right)^{\frac{k-1}{k}} \cdot \mathbb{E}_{\mathbf{z} \sim p} \left[ \left| \hat{f}(\mathbf{z}) - f^*(\mathbf{z}) \right| \right]^{1/k},
\end{aligned}
$$

where the last inequality again follows from the fact that $|f^*(\mathbf{z}) - \hat{f}(\mathbf{z})| \in [0, 1]$.

By Lemma A.1, we have that $\mathbb{E}_{\mathbf{z} \sim p} \left[ \left| \hat{f}(\mathbf{z}) - f^*(\mathbf{z}) \right| \right]^{1/k} \leq \left( \frac{\delta}{2} \right)^{\frac{1}{2k}}$. Combining the above relationships gives that

$$
|N_T^* - N_T| \leq \left( \int_{\mathbf{z}} p(\mathbf{z})^{\frac{k-T}{T(k-1)}} \right)^{\frac{k-1}{k}} \left( \frac{\delta}{2} \right)^{\frac{1}{2k}}
$$

Combining all results:

$$
\mathcal{E}_f(\hat{p}_r^{(T)}) \leq
$$

$$
\left( 1 - \frac{1}{T} \right) \| p_f \|_{2, p_r^{(\tau)}} \cdot \mathrm{Std}_{p_r^{(\tau)}}[\ln p] + \frac{\| p_f \|_\infty^{1/T} (\delta/2)^{\frac{1}{2T}}}{(1-\gamma)^{\frac{T+1}{T}} \exp\left( \frac{T-1}{T} H(p_r) + \frac{\gamma \ln \gamma}{1-\gamma} \right)} + \frac{\| p_f \|_\infty^{1/T} \left( \int_{\mathbf{z}} p(\mathbf{z})^{\frac{k-T}{T(k-1)}} \right)^{\frac{k-1}{k}} (\delta/2)^{\frac{1}{2k}}}{(1-\gamma)^{\frac{2T+2}{T}} \exp\left( \frac{2T-2}{T} H(p_r) - \frac{\delta - 2\gamma \ln \gamma}{1-\gamma} \right)}
$$

$\square$

## A.6. Proof of Theorem 3.8

**Theorem.** *Let $\hat{f} \in \mathcal{F}$ satisfy the excess risk bound $L(\hat{f}) - L(f^*) \leq \delta$. Define the $\tau$-tempered density $p^{(\tau)} \propto p^{1/\tau}$ for $\tau \in [1, T]$. Then for some $\tau \in [1, T]$, the Retain Error (6) of the $T$-tempered estimate $\hat{p}_r^{(T)}$ in (8) satisfies*

$$
\mathcal{E}_r(\hat{p}_r^{(T)}) := \mathrm{KL}\left( p_r \,\|\, \hat{p}_r^{(T)} \right) \leq \frac{\delta}{1-\gamma} + \left( 1 - \frac{1}{T} \right) \left( \frac{\left( \int_{\mathbf{z}} p^{1/\tau}(\mathbf{z}) \right) \mathbb{E}_{\mathbf{Z} \sim p^{(\tau)}} \left[ |\ln p(\mathbf{Z})| \right]}{(1-\gamma)^{\frac{\tau+1}{\tau}} \exp\left( \frac{\tau-1}{\tau} H(p_r) - \frac{\delta - \gamma \ln \gamma}{1-\gamma} \right)} - H(p_r) \right)
$$

*Proof of Theorem 3.8.* We first decompose the error $\mathcal{E}_r(\hat{p}_r^{(T)}) := \mathrm{KL}\left( p_r \,\|\, \hat{p}_r^{(T)} \right)$ into the KL between $p_r$ and the untempered estimate $\mathrm{KL}\left( p_r \,\|\, \hat{p}_r^{(1)} \right)$ as well as the discrepancy between the tempered and untempered estimates:

$$
\begin{aligned}
\mathrm{KL}\left( p_r \,\|\, \hat{p}_r^{(T)} \right) &= \mathbb{E}_{\mathbf{Z} \sim p_r} \left[ \ln \frac{p_r(\mathbf{Z})}{\hat{p}_r^{(T)}(\mathbf{Z})} \right] = \mathbb{E}_{\mathbf{Z} \sim p_r} \left[ \ln \frac{p_r(\mathbf{Z})}{\hat{p}_r^{(1)}(\mathbf{Z})} + \ln \frac{\hat{p}_r^{(1)}(\mathbf{Z})}{\hat{p}_r^{(T)}(\mathbf{Z})} \right] \\
&\leq \frac{\delta}{1-\gamma} + \mathbb{E}_{\mathbf{Z} \sim p_r} \left[ \ln \frac{\hat{p}_r^{(1)}(\mathbf{Z})}{\hat{p}_r^{(T)}(\mathbf{Z})} \right],
\end{aligned} \tag{18}
$$

where the inequality follows from Theorem 3.3. To bound the remaining term, we define the partition functions

$$
N_T = \int_{\mathbf{z}} p(\mathbf{z})^{1/T} \hat{f}(\mathbf{z}) \quad \text{and} \quad N_1 = \int_{\mathbf{z}} p(\mathbf{z}) \hat{f}(\mathbf{z}).
$$

Then,

$$\mathbb{E}_{\boldsymbol{Z}\sim p_r}\left[\ln\frac{\hat{p}_r^{(1)}(\boldsymbol{Z})}{\hat{p}_r^{(T)}(\boldsymbol{Z})}\right] = \mathbb{E}_{\boldsymbol{Z}\sim p_r}\left[\ln\left(\frac{N_T}{N_1}\frac{p(\boldsymbol{Z})\hat{f}(\boldsymbol{Z})}{p(\boldsymbol{Z})^{1/T}\hat{f}(\boldsymbol{Z})}\right)\right]$$

$$= \mathbb{E}_{\boldsymbol{Z}\sim p_r}\left[\ln\left(\frac{N_T}{N_1}p^{1-1/T}(\boldsymbol{Z})\right)\right]$$

$$= \left(1-\frac{1}{T}\right)\mathbb{E}_{\boldsymbol{Z}\sim p_r}\left[\ln p(\boldsymbol{Z})\right] + \ln N_T - \ln N_1 \tag{19}$$

We now focus on bounding the difference in log-partition functions $\ln N_T - \ln N_1$. Define the function $\psi : [\frac{1}{T}, 1] \to \mathbb{R}$ as

$$\psi(\alpha) = \ln\int_{\boldsymbol{z}} p^\alpha(\boldsymbol{z})\hat{f}(\boldsymbol{z}).$$

Then, we can express the log-partition difference in terms of $\psi$:

$$\ln N_T - \ln N_1 = \ln\left(\int_{\boldsymbol{z}} p(\boldsymbol{z})^{1/T}\hat{f}(\boldsymbol{z})\right) - \ln\left(\int_{\boldsymbol{z}} p(\boldsymbol{z})\hat{f}(\boldsymbol{z})\right) = \psi(1/T) - \psi(1). \tag{20}$$

By the Mean Value Theorem, there exists an intermediate temperature $\tau \in [1, T]$ such that:

$$\psi(1) - \psi(1/T) = \psi'(1/\tau)\left(1 - \frac{1}{T}\right)$$

We compute the derivative $\psi'(1/\tau)$:

$$\psi'(1/\tau) = \left(\frac{d}{d\alpha}\ln\int_{\boldsymbol{z}} p(\boldsymbol{z})^\alpha \hat{f}(\boldsymbol{z})\right)\Big|_{\alpha=1/\tau} = \frac{\int_{\boldsymbol{z}} p(\boldsymbol{z})^{1/\tau}\ln p(\boldsymbol{z})\hat{f}(\boldsymbol{z})}{\int_{\boldsymbol{z}} p(\boldsymbol{z})^{1/\tau}\hat{f}(\boldsymbol{z})} = \mathbb{E}_{\boldsymbol{Z}\sim\hat{p}_r^{(\tau)}}\left[\ln p(\boldsymbol{Z})\right]$$

Thus, $\ln N_T - \ln N_1 = -\psi'(1/\tau)\left(1 - \frac{1}{T}\right) = -(1 - \frac{1}{T})\mathbb{E}_{\boldsymbol{Z}\sim\hat{p}_r^{(\tau)}}\left[\ln p(\boldsymbol{Z})\right]$. Substituting this into (19) gives that

$$\mathbb{E}_{\boldsymbol{Z}\sim p_r}\left[\ln\frac{\hat{p}_r^{(1)}(\boldsymbol{Z})}{\hat{p}_r^{(T)}(\boldsymbol{Z})}\right] = \left(1 - \frac{1}{T}\right)\left(\mathbb{E}_{\boldsymbol{Z}\sim p_r}\left[\ln p(\boldsymbol{Z})\right] - \mathbb{E}_{\boldsymbol{Z}\sim\hat{p}_r^{(\tau)}}\left[\ln p(\boldsymbol{Z})\right]\right). \tag{21}$$

This represents the additional error incurred by tempering via (18). We upper bound the first term on the RHS as

$$\mathbb{E}_{\boldsymbol{Z}\sim p_r}\left[\ln p(\boldsymbol{Z})\right] \leq \mathbb{E}_{\boldsymbol{Z}\sim p_r}\left[\ln p_r(\boldsymbol{Z})\right] = -H(p_r) \tag{22}$$

We bound the second term as

$$-\mathbb{E}_{\boldsymbol{Z}\sim\hat{p}_r^{(\tau)}}\left[\ln p(\boldsymbol{Z})\right] = -\int_{\boldsymbol{z}}\left(\frac{p^{1/\tau}(\boldsymbol{z})\hat{f}(\boldsymbol{z})}{\int_{\boldsymbol{u}} p^{1/\tau}(\boldsymbol{u})\hat{f}(\boldsymbol{u})}\right)\ln p(\boldsymbol{z})$$

$$\leq \int_{\boldsymbol{z}}\left(\frac{p^{1/\tau}(\boldsymbol{z})}{\int_{\boldsymbol{u}} p^{1/\tau}(\boldsymbol{u})\hat{f}(\boldsymbol{u})}\right)|\ln p(\boldsymbol{z})|$$

Then by Lemma A.2, we can lower bound the $\tau$-tempered partition function as

$$\int_{\boldsymbol{u}} p^{1/\tau}(\boldsymbol{u})\hat{f}(\boldsymbol{u}) \geq (1-\gamma)^{\frac{\tau+1}{\tau}}\exp\left(\frac{\tau-1}{\tau}H(p_r) - \frac{\delta-\gamma\ln\gamma}{1-\gamma}\right).$$

Define the $\tau$-tempered distribution $p^{(\tau)}(\boldsymbol{z}) \propto p^{1/\tau}(\boldsymbol{z})$. Then, the second term in (21) is bounded as

$$-\mathbb{E}_{\boldsymbol{Z} \sim \hat{p}_r^{(\tau)}} \left[\ln p(\boldsymbol{Z})\right] \leq \frac{\left(\int_{\boldsymbol{z}} p^{1/\tau}(\boldsymbol{z})\right) \mathbb{E}_{\boldsymbol{Z} \sim p^{(\tau)}} \left[|\ln p(\boldsymbol{Z})|\right]}{(1-\gamma)^{\frac{\tau+1}{\tau}} \exp\left(\frac{\tau-1}{\tau} H(p_r) - \frac{\delta - \gamma \ln \gamma}{1-\gamma}\right)} \tag{23}$$

Substituting the bounds in (22) and (23) for (21) gives an upper bound on the cost of tempering for the Retain Error:

$$\mathbb{E}_{\boldsymbol{Z} \sim p_r} \left[\ln \frac{\hat{p}_r^{(1)}(\boldsymbol{Z})}{\hat{p}_r^{(T)}(\boldsymbol{Z})}\right] \leq \left(1 - \frac{1}{T}\right)\left(-H(p_r) + \frac{\left(\int_{\boldsymbol{z}} p^{1/\tau}(\boldsymbol{z})\right) \mathbb{E}_{\boldsymbol{Z} \sim p^{(\tau)}} \left[|\ln p(\boldsymbol{Z})|\right]}{(1-\gamma)^{\frac{\tau+1}{\tau}} \exp\left(\frac{\tau-1}{\tau} H(p_r) - \frac{\delta - \gamma \ln \gamma}{1-\gamma}\right)}\right)$$

Applying this to (18) gives the desired result:

$$\mathcal{E}_r(\hat{p}_r^{(T)}) := \mathrm{KL}\left(p_r \,\|\, \hat{p}_r^{(T)}\right) \leq \frac{\delta}{1-\gamma} + \left(1 - \frac{1}{T}\right)\left(\frac{\left(\int_{\boldsymbol{z}} p^{1/\tau}(\boldsymbol{z})\right) \mathbb{E}_{\boldsymbol{Z} \sim p^{(\tau)}} \left[|\ln p(\boldsymbol{Z})|\right]}{(1-\gamma)^{\frac{\tau+1}{\tau}} \exp\left(\frac{\tau-1}{\tau} H(p_r) - \frac{\delta - \gamma \ln \gamma}{1-\gamma}\right)} - H(p_r)\right)$$

$\square$

## B. Tempering as Smoothing under a KL Constraint

We briefly describe how tempering a distribution can be interpreted as maximizing entropy under a KL divergence constraint. Consider the following optimization problem, where $p$ is some fixed distribution and $C > 0$ is a constant:

$$\mathring{p} = \underset{\mu}{\arg\max}\, H(\mu) \quad \text{s.t.} \quad \mathrm{KL}\left(\mu \,\|\, p\right) \leq C. \tag{24}$$

This is a convex program in $\mu$. Solving the Lagrangian of (24) shows that

$$\mathring{p} \propto p^{1/T(C)},$$

where $T(C) \geq 1$ is a function of the KL constraint parameter $C$. Thus, tempering can be seen as finding the distribution with maximum entropy within a KL ball of radius $C$ around $p$, i.e., the "smoothest" distribution that stays close to $p$ in KL divergence.

## C. Logistic Regression Excess Risk

**Proposition C.1.** *Assume $\mathcal{S}_n$ comprises $n$ i.i.d. samples of pairs $(\boldsymbol{z}, s)$. Let $\hat{\boldsymbol{\phi}}_\lambda = \arg\min_{\boldsymbol{\phi}} L_n^\lambda(\boldsymbol{\phi})$ minimize the regularized finite-sample loss (3). Then*

$$\mathbb{E}_{\mathcal{S}_n}\left[L(\hat{\boldsymbol{\phi}}_\lambda) - L(\boldsymbol{\phi}^*)\right] \leq \lambda \|\boldsymbol{\phi}^*\|_2^2 + \frac{2\mathbb{E}\left[\|\boldsymbol{z}\|_2^2\right]}{n\lambda}.$$

*Moreover, setting $\lambda = \frac{1}{\|\boldsymbol{\phi}^*\|_2} \sqrt{\frac{2\mathbb{E}\left[\|\boldsymbol{z}\|_2^2\right]}{n}}$ gives that*

$$\mathbb{E}_{\mathcal{S}_n}\left[L(\hat{\boldsymbol{\phi}}_\lambda) - L(\boldsymbol{\phi}^*)\right] \leq 2 \|\boldsymbol{\phi}^*\|_2 \sqrt{\frac{2\mathbb{E}\left[\|\boldsymbol{z}\|_2^2\right]}{n}}. \tag{25}$$

*Remark* C.2 (Class Imbalance). While this analysis assumes samples in $\mathcal{S}_n$ are drawn i.i.d. from $\mathbb{P}$ in Section 2.1, datasets in practice are often formed by subsampling $\mathcal{D}_r$ against the full $\mathcal{D}_f$, leading to a higher forget-set proportion than the population value $\gamma$. Appendix D describes a correction that restores infinite-sample consistency under such class imbalance.

*Proof of Proposition C.1.* The proof follows the derivation in Chapter 13 of (Shalev-Shwartz & Ben-David, 2014).

Denote the samples in $\mathcal{S}_n$ as $\mathcal{S}_n = \{(z_i, s_i)\}_{i=1}^n$. We assume each $(z_i, s_i)$ are i.i.d. samples from $\mathbb{P}$, and we denote the marginal distribution over the inputs as $p(z)$. For clarity, we first define the notation for the finite-sample, population, and unregularized losses along with their minimizers.

We consider functions $\mathcal{F} = \{f_\phi \mid f_\phi(z) = \sigma\left(\phi^\top z\right)\}$, where $\sigma$ denotes the sigmoid function. For a single sample $(z_i, s_i)$, we denote the cross-entropy loss as $\ell(f_\phi(z_i), s_i)$.

$L$ denotes the population risk which is minimized by $\phi^*$:

$$\phi^* = \operatorname*{argmin}_\phi L(\phi) \qquad L(\phi) := \mathbb{E}_{(\mathbf{Z},S)\sim\mathbb{P}}\left[\ell(f_\phi(\mathbf{Z}), S)\right].$$

$L_n(\cdot\,; \mathcal{S}_n)$ denotes the finite-sample risk over the dataset $\mathcal{S}_n$:

$$L_n(\phi\,; \mathcal{S}_n) := \frac{1}{n} \sum_{i=1}^n \ell(f_\phi(z_i), s_i)$$

$L_n^\lambda(\cdot\,; \mathcal{S}_n)$ denotes the regularized finite-sample risk over $\mathcal{S}_n$, which is minimized by $\hat{\phi}_\lambda$:

$$\hat{\phi}_\lambda = \operatorname*{argmin}_\phi L_n^\lambda(\phi\,; \mathcal{S}_n) \qquad L_n^\lambda(\phi\,; \mathcal{S}_n) := \frac{1}{n} \sum_{i=1}^n \ell(f_\phi(z_i), s_i) + \lambda \|\phi\|_2^2.$$

Although the main text assumes losses are evaluated on $\mathcal{S}_n$, we make the dataset explicit here for clarity, as the proof relies on a stability argument involving a swapped dataset. We fix an index $j \in \{1, \ldots, n\}$, and let $\mathcal{S}_n^{(j)}$ be the dataset obtained by replacing the $j^{\text{th}}$ sample $(z_j, s_j)$ with an i.i.d. copy $(z_j', s_j') \sim \mathbb{P}$. Let $\hat{\phi}_\lambda^{(j)}$ denote the corresponding minimizer of $L_n^\lambda$ computed on $\mathcal{S}_n^{(j)}$

$$\hat{\phi}_\lambda^{(j)} = \operatorname*{argmin}_\phi L_n^\lambda(\phi\,; \mathcal{S}_n^{(j)}) \tag{26}$$

Since the logistic loss is convex and the regularization term is $2\lambda$-strongly convex, the objective $L_n^\lambda$ is $2\lambda$-strongly convex. Consequently, for any $\phi$:

$$L_n^\lambda(\hat{\phi}_\lambda^{(j)}\,; \mathcal{S}_n) - L_n^\lambda(\hat{\phi}_\lambda\,; \mathcal{S}_n) \geq \lambda \left\|\hat{\phi}_\lambda - \hat{\phi}_\lambda^{(j)}\right\|_2^2. \tag{27}$$

The two empirical objectives computed on $\mathcal{S}_n$ and $\mathcal{S}_n^{(j)}$ differ in exactly one sample, which implies that

$$\begin{aligned}
L_n^\lambda(\hat{\phi}_\lambda^{(j)}\,; \mathcal{S}_n) - L_n^\lambda(\hat{\phi}_\lambda\,; \mathcal{S}_n) &= L_n^\lambda(\hat{\phi}_\lambda^{(j)}\,; \mathcal{S}_n^{(j)}) - L_n^\lambda(\hat{\phi}_\lambda\,; \mathcal{S}_n^{(j)}) \\
&\quad + \frac{1}{n}\left(\ell(f_{\hat{\phi}_\lambda^{(j)}}(z_j), s_j) - \ell(f_{\hat{\phi}_\lambda^{(j)}}(z_j'), s_j') - \ell(f_{\hat{\phi}_\lambda}(z_j), s_j) + \ell(f_{\hat{\phi}_\lambda}(z_j'), s_j')\right) \\
&\leq \frac{1}{n}\left(\ell(f_{\hat{\phi}_\lambda^{(j)}}(z_j), s_j) - \ell(f_{\hat{\phi}_\lambda^{(j)}}(z_j'), s_j') - \ell(f_{\hat{\phi}_\lambda}(z_j), s_j) + \ell(f_{\hat{\phi}_\lambda}(z_j'), s_j')\right), \quad (28)
\end{aligned}$$

where the inequality follows from the optimality of $\hat{\phi}_\lambda^{(j)}$ over $L_n^\lambda(\phi\,; \mathcal{S}_n^{(j)})$ from (26).

The sample-wise loss $\ell(f_\phi(z), s)$ is $\|z\|_2$-Lipschitz continuous with respect to $\phi$, so for any $\phi, \phi'$:

$$|\ell(f_\phi(z), s) - \ell(f_{\phi'}(z), s)| \leq \|\phi - \phi'\|_2 \|z\|_2.$$

Applying this inequality to (28) gives that

$$L_n^\lambda(\hat{\phi}_\lambda^{(j)}\,; \mathcal{S}_n) - L_n^\lambda(\hat{\phi}_\lambda\,; \mathcal{S}_n) \leq \frac{1}{n}\left\|\hat{\phi}_\lambda^{(j)} - \hat{\phi}_\lambda\right\|_2 \left(\|z_j\|_2 + \|z_j'\|_2\right). \tag{29}$$

Combining the bounds in (27) and (29) yields a bound on the distance to the "swapped" estimator:

$$\left\|\hat{\phi}_\lambda^{(j)} - \hat{\phi}_\lambda\right\|_2 \leq \frac{1}{n\lambda}\left(\|z_j\|_2 + \|z_j'\|_2\right). \tag{30}$$

Thus, on the sample $(z_j, s_j)$, we can bound the additional loss incurred by the swapped estimator again using Lipschitzness:

$$\ell\left(f_{\hat{\phi}_\lambda^{(j)}}(z_j), s_j\right) - \ell\left(f_{\hat{\phi}_\lambda}(z_j), s_j\right) \leq \left\|\hat{\phi}_\lambda^{(j)} - \hat{\phi}_\lambda\right\|_2 \|z_j\|_2$$

$$\leq \frac{1}{n\lambda}\left(\|z_j\|_2 + \|z_j'\|_2\right)\|z_j\|_2 \tag{31}$$

We now use the fact that the expected generalization gap when learning from the finite sample dataset $\mathcal{S}_n$ is equal to the expected additional loss incurred by the swapped estimator on the replaced sample (Theorem 13.2 of (Shalev-Shwartz & Ben-David, 2014)). Formally,

$$\mathbb{E}_{\mathcal{S}_n}\left[L(\hat{\phi}_\lambda) - L_n(\hat{\phi}_\lambda)\right] = \mathbb{E}_{(\mathcal{S}_n, (z_j', s_j'))}\left[\ell\left(f_{\hat{\phi}_\lambda^{(j)}}(z_j), s_j\right) - \ell\left(f_{\hat{\phi}_\lambda}(z_j), s_j\right)\right] \tag{32}$$

Applying (32) to the bound in (31):

$$\mathbb{E}_{\mathcal{S}_n}\left[L(\hat{\phi}_\lambda) - L_n(\hat{\phi}_\lambda)\right] \leq \mathbb{E}\left[\frac{1}{n\lambda}\left(\|z_j\|_2 + \|z_j'\|_2\right)\|z_j\|_2\right]$$

$$= \frac{1}{n\lambda}\left(\mathbb{E}\left[\|z_j\|_2^2\right] + \mathbb{E}^2\left[\|z_j\|_2\right]\right) \quad \text{(since } z_j \text{ and } z_j' \text{ are i.i.d.)}$$

$$\leq \frac{2}{n\lambda}\mathbb{E}\left[\|z\|_2^2\right], \tag{33}$$

where $z \sim p$ is a generic input variable drawn from the marginal $p$. Since the finite-sample regularized loss is larger than the unregularized loss and minimized by $\hat{\phi}_\lambda$, we have that for any $\phi$:

$$L_n(\hat{\phi}_\lambda) \leq L_n^\lambda(\hat{\phi}_\lambda) \leq L_n^\lambda(\phi).$$

Taking the expectation over $\mathcal{S}_n$:

$$\mathbb{E}_{\mathcal{S}_n}\left[L_n(\hat{\phi}_\lambda)\right] \leq L(\phi) + \lambda\|\phi\|_2^2 \tag{34}$$

Applying (34) to $\phi = \phi^*$ and using the bound in (33):

$$\mathbb{E}_{\mathcal{S}_n}\left[L(\hat{\phi}_\lambda) - L(\phi^*)\right] - \lambda\|\phi^*\|_2^2 \leq \mathbb{E}_{\mathcal{S}_n}\left[L(\hat{\phi}_\lambda) - L_n(\hat{\phi}_\lambda)\right] \leq \frac{2}{n\lambda}\mathbb{E}\left[\|z\|_2^2\right] \tag{35}$$

Rearranging the outer inequality of (35) gives the desired expression. $\qquad\square$

## D. Density Ratio Estimation under Class Imbalance

The logistic regression excess risk bound assumes the samples in the unlearning dataset $\mathcal{S}_n = \{(z_i, s_i)\}_{i=1}^n$ reflect the true population proportion of forget set samples $\gamma$, meaning that

$$\mathbb{E}_{\mathcal{S}_n}\left[\frac{1}{n}\sum_{i=1}^n \mathbb{1}\{s_i = 0\}\right] = \gamma$$

However, as mentioned in Remark C.2, practical unlearning datasets are often constructed by subsampling $\mathcal{D}_r$ along the full $\mathcal{D}_f$, resulting in an observed forget set proportion which is larger than the population value $\gamma$. In this section, we show how to modify our estimator in this case to account for this mismatch and maintain infinite-sample consistency.

Consider an $n$-sample unlearning dataset $\mathcal{S}_n^\mu = \{(z_i^\mu, s_i^\mu)\}_{i=1}^n$ with forget set proportion $\mu$:

$$\mu = \frac{1}{|\mathcal{S}_n^\mu|}\sum_{i=1}^n \mathbb{1}\{s_i^\mu = 0\}$$

We still apply the probabilistic classification subproblem and recover an estimator $\hat{f}_\mu$ for the true conditional distribution $f_\mu^*(z)$ defined as

$$\hat{f}_\mu(\boldsymbol{z}) \approx f_\mu^*(\boldsymbol{z}) = \mathbb{P}_\mu(S = 1 \mid \boldsymbol{Z} = \boldsymbol{z}) = \frac{(1-\mu)p_r(\boldsymbol{z})}{(1-\mu)p_r(\boldsymbol{z}) + \mu p_f(\boldsymbol{z})},$$

where $\mathbb{P}_\mu$ denotes the base measure for the analogous generative process to the one defined in Section 2.1, except when $\gamma = \mu$.

Define $p_\mu(\boldsymbol{z})$ as the marginal over $\boldsymbol{z}$ induced by $\mathbb{P}_\mu$:

$$p_\mu(\boldsymbol{z}) = (1-\mu)p_r(\boldsymbol{z}) + \mu p_f(\boldsymbol{z}).$$

If we could access $p_\mu$, then we could employ the same estimator $\hat{p}_r$ for $p_r$ from our previous theoretical analysis as $\hat{p}_r(\boldsymbol{z}) \propto p_\mu(\boldsymbol{z})\hat{f}_\mu(\boldsymbol{z})$. However, we can only access $p(\boldsymbol{z})$ which includes mixture weight $\gamma \neq \mu$.

However, we can still recover an asymptotically consistent estimator of $p_r$ by modifying our transformation of $\hat{f}_\mu$ to extract the density ratio $\frac{p_r(\boldsymbol{z})}{p(\boldsymbol{z})}$ using the following relationship:

$$\frac{p_r(\boldsymbol{z})}{p(\boldsymbol{z})} = \frac{\mu f_\mu^*(\boldsymbol{z})}{(\mu - \gamma)f_\mu^*(\boldsymbol{z}) + \gamma(1-\mu)}$$

Thus, we can apply a different algebraic transformation of $p$ and $\hat{f}_\mu$ to recover an asymptotically consistent estimator $\bar{p}_r$:

$$\bar{p}_r(\boldsymbol{z}) \propto \frac{\mu \hat{f}_\mu(\boldsymbol{z})p(\boldsymbol{z})}{(\mu - \gamma)\hat{f}_\mu(\boldsymbol{z}) + \gamma(1-\mu)}$$

When $\gamma = \mu$, we recover the same estimator $\bar{p}_r = \hat{p}_r \propto p \cdot \hat{f}_\mu$ as in the case where $\mathcal{S}_n$ contains i.i.d. samples from the model in Section 2.1. While this new estimator requires knowledge of $\gamma$, we assume this is known from the original training process. Finally, we note that the original (incomputable) estimator $\hat{p}_r \propto \hat{f}_\mu \cdot p_\mu$ is equal to our new estimator $\bar{p}_r$ if and only if $\hat{f}_\mu = f_\mu^*$, i.e., the two estimators are only equivalent in the limit where they achieve optimality.

# E. Synthetic Data Experiments

We evaluate the proposed method on a synthetic benchmark generated from Gaussian component distributions to verify our theoretical insights. Specifically, we consider the setting where the retain and forget distributions follow univariate Gaussian distributions, denoted as $p_r = \mathcal{N}(\mu_r, v_r)$ and $p_f = \mathcal{N}(\mu_f, v_f)$, with means $\mu_r, \mu_f$ and variances $v_r, v_f$, respectively.

**Data Generation and Classification.** We construct the mixture distribution $p(z) = \gamma p_f(z) + (1-\gamma)p_r(z)$. We generate an $n$-sample dataset $\mathcal{S}_n = \{(z_i, s_i)\}_{i=1}^n$ by sampling the component label $s_i \sim \text{Bernoulli}(1 - \gamma)$, where $s_i = 1$ indicates $z_i \sim p_r$ and $s_i = 0$ indicates $z_i \sim p_f$. Throughout these experiments, we set $\gamma = 0.1$, fix the means as $\mu_r = 1.0$ and $\mu_f = 0.0$, and fix the retain variance $v_r = 1.0$.

In this experimental setup, we consider the full T3-Unlearning pipeline: we train a probabilistic classifier $\hat{f}$ using the generated data and construct the retain set distribution estimator $\hat{p}_r^{(T)} \propto p^{1/T} \cdot \hat{f}$. In the univariate Gaussian setting, the Bayes optimal classifier $f^*(z) = \mathbb{P}(s = 1 \mid z)$ takes the form of a sigmoid applied to a quadratic function of the input. Consequently, we train our surrogate classifier using a specific parameterization $\phi$ by computing a quadratic feature map of the inputs $z \in \mathbb{R}$.

Let $\varphi(z) = [1, z, z^2]^\top \in \mathbb{R}^3$ denote the fixed quadratic feature map, and define the family of classifiers $f_\phi(\varphi(z)) = \sigma\left(\phi^\top \varphi(z)\right)$ parameterized by $\phi \in \mathbb{R}^3$, where $\sigma$ denotes the sigmoid function. For a regularization coefficient $\lambda \geq 0$, we minimize the regularized logistic regression objective $L_n^\lambda$ over the $n$-sample dataset $\mathcal{S}_n$ defined as

$$L_n^\lambda(\phi\,;\mathcal{S}_n) := \frac{1}{n}\sum_{i=1}^n \ell(f_\phi(\varphi(z_i)), s_i) + \lambda \|\phi\|_2^2,$$

where $\ell$ denotes the binary cross-entropy loss. Let $\hat{\phi}_\lambda$ minimize $L_n^\lambda(\phi; \mathcal{S}_n)$, so our recovered classifier is $f_{\hat{\phi}_\lambda}$ and the corresponding density estimate is $\hat{p}_r^{(T)}(z) \propto p^{1/T}(z) \cdot f_{\hat{\phi}_\lambda}(\varphi(z))$.

**Partition Function Estimation.** When applying non-trivial tempering ($T > 1$), the normalized density estimate for recovered classifier parameters $\hat{\phi}_\lambda$ is given by:

$$\hat{p}_r^{(T)}(z) = \frac{p^{1/T}(z) \cdot f_{\hat{\phi}_\lambda}(\varphi(z))}{\int_u p^{1/T}(u) \cdot f_{\hat{\phi}_\lambda}(\varphi(u))}. \tag{36}$$

Computing the partition function (the denominator) requires numerical integration. Since tempering a Gaussian distribution $\mathcal{N}(\mu, \sigma^2)$ by temperature $T$ effectively scales the variance to $T\sigma^2$, we employ an approximation that assumes negligible overlap between the tempered components. We approximate the base density term as $p(z)^{1/T} \approx (1-\gamma)^{1/T} p_r(z)^{1/T} + \gamma^{1/T} p_f(z)^{1/T}$ and then estimate the integral via Monte Carlo sampling.

We similarly evaluate the Retain Error (6) and Forget Error (7) metrics via Monte Carlo sampling, as they both represent expectations under the respective ground truth densities $p_r$ and $p_f$.

**Connection to Theory.** The following experiments aim to validate the theoretical analysis in Section 3, which relates Retain and Forget Errors (measures of unlearning quality) to the excess risk of the recovered classifier $f_{\hat{\phi}_\lambda}$ and underlying distribution parameters such as the peak forget density $\|p_f\|_\infty$ and forget proportion $\gamma$. While the theory abstracts classifier performance using the excess risk $\delta$, here we provide a more concrete perspective, linking $\delta$ to practical factors that govern its achievable value: the distribution parameters and the number of available samples $n$.

We report results in terms of the underlying quantities which directly control the key theoretical parameters $\|p_f\|_\infty$ (peak forget component density) and $\delta$ (classifier excess risk). To probe the effect of $\|p_f\|_\infty$, we vary the forget variance $v_f$, noting that for a Gaussian, the peak density is $\|p_f\|_\infty = (2\pi v_f)^{-1/2}$. Thus, sending $v_f \to 0$ allows us to study the impact of highly concentrated $p_f$ on unlearning. Regarding classifier performance, the primary practical factor controlling excess risk is the number of samples $n$. In the notation of this specific parameterization, the excess risk associated with a candidate set of parameters $\phi$ is defined

$$\delta = \mathbb{E}_{(Z,S)}\left[\ell(f_\phi(\varphi(Z)), S)\right] - \mathbb{E}_{(Z,S)}\left[\ell(f^*(Z), S)\right],$$

where $f^*(z) := \mathbb{P}\left(S = 1 \mid Z = z\right)$ is the Bayes optimal classifier.

As shown in Appendix C, for regularized logistic regression with properly tuned $\lambda$, the expected excess risk decays as $\mathcal{O}(n^{-1/2})$. While this rate also depends on the data distribution variance, in Experiment 2 we fix distribution parameters and vary $n$ to isolate its effect.

**Experiment 1: Robustness to Forget Distribution Sharpness.** We analyze how tempering affects unlearning error for varying levels of forget variance $v_f$. As $v_f \to 0$, the forget component's peak density grows unbounded, $\|p_f\|_\infty \to \infty$. This experiment tests our theoretical predictions that (i) tempering reduces Forget Error when $\|p_f\|_\infty$ is large (small $v_f$), and (ii) as $\|p_f\|_\infty$ increases ($v_f$ decreases), larger base model temperatures $T$ are required to minimize Forget Error. In particular, we aim to illustrate the tradeoff predicted by Theorem 3.7: increasing $T$ initially lowers Forget Error, but eventually the tempering-induced bias dominates, especially when $p_r$ and $p_f$ have significant overlap.

For each distinct value of $v_f$, we first perform a hyperparameter search for the regularization coefficient $\lambda$ that minimizes the average population risk over 10 trials. Using the selected $\lambda$, we train classifiers on fresh $n$-sample datasets and evaluate unlearning performance across a range of temperatures $T \in [1.0, 3.0]$. We report the average Retain and Forget Errors as a function of base model temperature $T$ over 200 trials in Figure 2 for three levels of $v_f$, with all other data distribution parameters fixed.

For the smallest forget variance $v_f = 1 \times 10^{-6}$, corresponding to the largest peak density $\|p_f\|_\infty$ (purple curve), the Forget Error decreases monotonically with $T$. This matches our theory: tempering is essential to control the large peak, and since $p_r$ and $p_f$ are effectively disjoint in this case, the tempering-induced bias in Theorem 3.7 is negligible. As a result, increasing $T$ continues to reduce Forget Error with minimal cost. For the intermediate variance $v_f = 1 \times 10^{-3}$ (blue curve), we observe a clear tradeoff: Forget Error initially decreases with $T$, but then rises as tempering-induced bias begins to dominate. For the largest variance $v_f = 1 \times 10^0$, where the forget distribution is flattest (yellow curve), Forget Error is minimized at $T = 1.0$, i.e., the untempered estimator. This aligns with theory, which predicts that tempering primarily benefits sharply concentrated forget components.

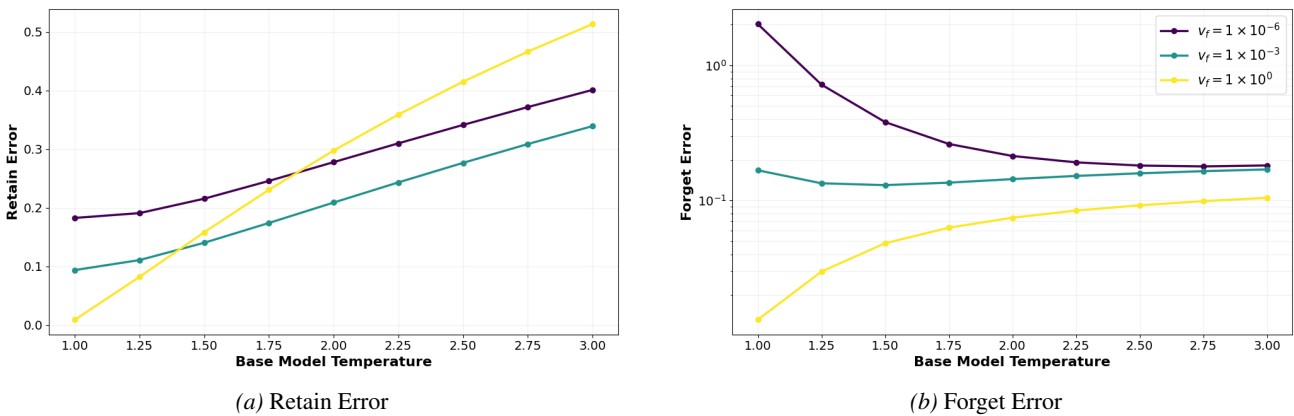

*(a)* Retain Error            *(b)* Forget Error

*Figure 2.* Retain and Forget Errors as a function of forget set variance $v_f$ and base model temperature $T$.

In terms of Retain Error, increasing $T$ consistently worsens performance across all settings. We observe that the largest forget variance setting $v_f = 1 \times 10^0$ is most sensitive to tempering, as the Retain Error increases at a faster rate with respect to $T$ compared to when $v_f \in \{1 \times 10^{-3}, 1 \times 10^{-6}\}$. This is consistent with Theorem 3.8, which quantifies the tempering-induced bias on the Retain Error $\mathcal{E}_r(\hat{p}_r^{(T)})$. The effects of the forget variance are captured in the numerator of the bias term which depends on the entire mixture $p$:

$$\mathcal{E}_r(\hat{p}_r^{(T)}) := \mathrm{KL}\left(p_r \,\|\, \hat{p}_r^{(T)}\right) \leq \frac{\delta}{1-\gamma} + \underbrace{\left(1 - \frac{1}{T}\right)\left(\frac{\left(\int_z p^{1/\tau}(z)\right)\mathbb{E}_{\boldsymbol{Z}\sim p^{(\tau)}}\left[|\ln p(\boldsymbol{Z})|\right]}{(1-\gamma)^{\frac{\tau+1}{\tau}}\exp\left(\frac{\tau-1}{\tau}H(p_r) - \frac{\delta-\gamma\ln\gamma}{1-\gamma}\right)} - H(p_r)\right)}_{\text{Tempering-Induced Bias}}.$$

Specifically, the rate at which Retain Error increases as a function of $T$ depends on the forget variance through the mixture functional

$$\left(\int_z p^{1/\tau}(\boldsymbol{z})\right)\mathbb{E}_{\boldsymbol{Z}\sim p^{(\tau)}}\left[|\ln p(\boldsymbol{Z})|\right] := \int_z p^{1/\tau}(\boldsymbol{z})\left|\ln p(\boldsymbol{z})\right|.$$

Intuitively, we expect this term to grow as $p$ becomes flatter, aligning with our observation that Retain Error is more sensitive to increases in $T$ when the variance of one of its modes $v_f$ increases. To see this, consider the simplified setting where $p$ is a single univariate Gaussian, centered at 0 with variance $v$, denoted $p_v$. We compute this integral explicitly, aiming to show it is increasing in $v$. To simplify the calculation, consider the regime where $v \geq \frac{1}{2\pi}$, so that $\ln p_v(z) \leq 0$ for all $z$. Then,

$$\int_z p_v^{1/\tau}(z)\left|\ln p_v(z)\right| = \int_z \sqrt{\tau}(2\pi v)^{\frac{\tau-1}{2\tau}}p_{v\tau}(z)\left(\frac{z^2}{2v} + \frac{1}{2}\ln(2\pi v)\right)$$

$$= \sqrt{\tau}(2\pi v)^{\frac{\tau-1}{2\tau}}\left(\frac{\tau}{2} + \frac{1}{2}\ln(2\pi v)\right)$$

Since $\tau \geq 1$, the result is increasing in $v$. While this calculation is carried out for a single Gaussian density, it captures the dominant tail-driven behavior of the mixture functional $\int_z p^{1/\tau}(z)\left|\ln p(z)\right|$. In particular, increasing the variance of one mixture component spreads probability mass into regions where $|\ln p(z)|$ is large, and the tempered weighting $p^{1/\tau}(z)$ places increased emphasis on these regions. As a result, the mixture functional becomes more sensitive to variance inflation of an individual component. This aligns with the empirically observed trend that the Retain Error is more sensitive in practice to increases in $T$ when the forget set variance is set to the largest experimental value $v_f = 1 \times 10^0$.

These results highlight the practical role of tempering: for data distributions with sharply concentrated forget components (small $v_f$), tempering can substantially reduce Forget Error relative to the untempered baseline ($T = 1.0$) without requiring additional data, i.e., for the same classifier quality and the same excess risk $\delta$.

**Experiment 2: Impact of Sample Size.** In this experiment, we fix the forget component variance at $v_f = 10^{-3}$ and vary the dataset size $n$. Following the same procedure as above, we select the optimal regularization coefficient and then plot

the unlearning errors as functions of the base model temperature $T$. This setup tests the regimes in which tempering is beneficial: as $n$ increases, the expected classifier excess risk $\delta$ decays at an $\mathcal{O}(n^{-1/2})$ rate. According to Theorem 3.7, tempering is most useful when the classifier exhibits higher excess risk, which occurs with smaller sample sizes, relative to the sharpness of $p_f$. Consequently, we expect that as $n$ grows, the temperature $T$ that minimizes Forget Error approaches the untempered limit $T = 1.0$. We plot results in Figure 3.

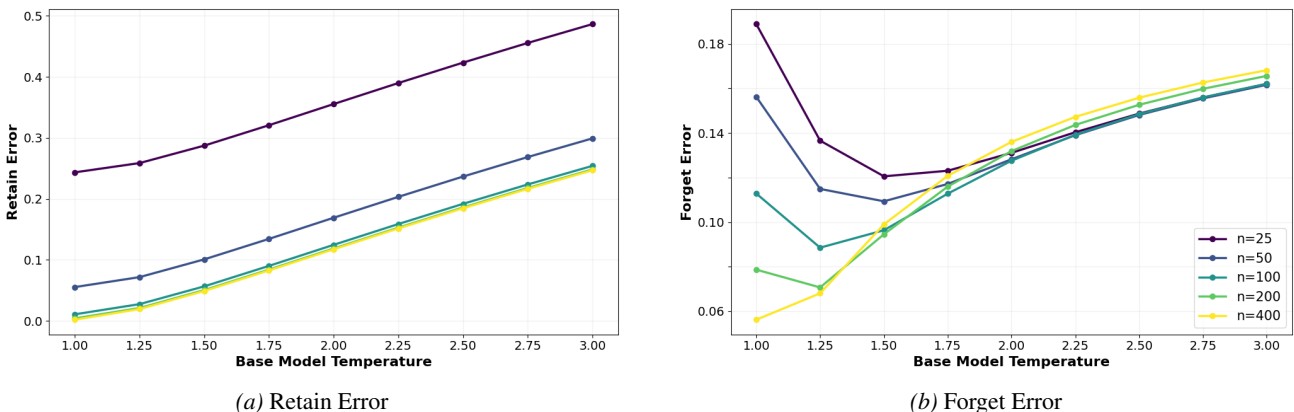

*(a)* Retain Error            *(b)* Forget Error

*Figure 3.* Retain and Forget Errors as a function of sample size $n$ and temperature $T$.

We observe the behavior predicted by our theory: as the number of samples $n$ increases (corresponding to the brighter curves) the temperature $T$ that minimizes Forget Error decreases. In the limiting case of $n = 400$ (yellow curve), Forget Error is minimized at $T = 1.0$. This confirms the theoretical tradeoff: tempering reduces the dependence of Forget Error on the sharpness $\|p_f\|_\infty$, but slows convergence with respect to classifier risk. When $n$ is small, the classifier is weaker, and tempering provides a robust mechanism to control Forget Error. However, as $n$ grows, we learn stronger classifiers which achieve smaller values of excess risk, so the need for tempering lessens.

Thus, for a fixed sample size $n$, tempering is an effective tool in reducing Forget Error when the forget component sharpness dominates the achievable excess risk. As expected, the Retain Error increases monotonically with $T$ across all settings.

For the setting $n = 25$, we visualize an example learned classifier and the resulting density estimates $\hat{p}_r^{(T)}$ for various values of $T$ in Figure 4 below, showing how tempering reduces the information leakage from the forget set (decreasing Forget Error) while increasing mismatch to $p_r$ in regions where the retain set samples are more likely (increasing Retain Error).

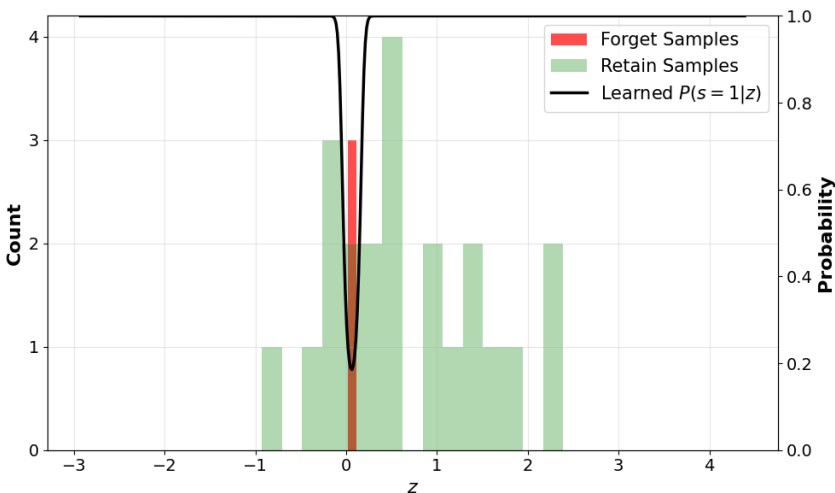

*Figure 4.* Example generated data and learned classifier for the setting of Experiment 2 with $n = 25$ samples. The y-axis tracks both the sample counts for the generated samples (left) and the probabilities assigned by the classifier (right).

In this example, the classifier learns to predict the correct labels for the narrow region in which the forget set samples are generated, indicated by the classifier probability (black curve) which follows the y-axis ticks on the right side of the plot. For this classifier, we then plot the approximated retain set density $\hat{p}_r^{(T)}$ for $T \in \{1.0, 1.5, 2.0\}$ in Figure 5 below, visually depicting the tradeoffs associated with tempering.

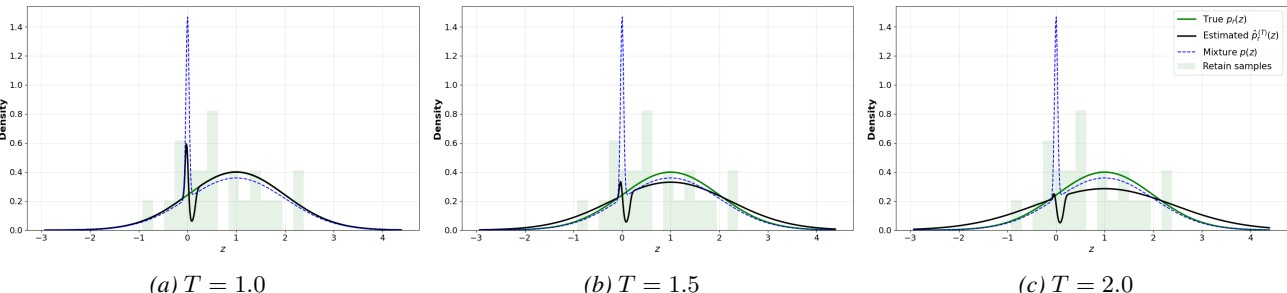

*(a) $T = 1.0$*  *(b) $T = 1.5$*  *(c) $T = 2.0$*

*Figure 5.* Estimated retain set densities $\hat{p}_r^{(T)}$ in the setting of Experiment 2 for $n = 25$ samples and temperature $T \in \{1.0, 1.5, 2.0\}$.

We observe that as $T$ increases, the estimated density $\hat{p}_r^{(T)}$ (black curve) becomes progressively less influenced by the forget set, with the spike at $z = 0$ increasingly suppressed. The classifier learns a smooth prediction function that overestimates the region where samples are likely from the forget component, causing the estimated retain density to dip just beyond the spike. For larger $T$, this effect produces a more noticeable mismatch between $\hat{p}_r^{(T)}$ and the true retain density $p_r$ in regions away from the forget mode, illustrating how larger temperatures can increase Retain Error.

## F. LLM Experiment Details

### F.1. Baseline Methods

We formally define each of the baseline methods and the notation for their hyperparameters. Let $z_r = (q_r, a_r) \in \mathcal{D}_r$ and $z_f = (q_f, a_f) \in \mathcal{D}_f$ denote representative samples from the retain and forget sets, respectively. We use the notation $q, a$ since benchmarks like TOFU consist of question-answer pairs, although the methods apply to general text sequences.

We denote the unlearned model by $p_\theta$, where $\theta$ are the parameters optimized during unlearning, and the original pretrained parameters by $\theta^*$. We denote by $\mathrm{StopGrad}[\cdot]$ the stop-gradient operator (equivalent to .detach() in automatic differentiation frameworks). Let $\mathrm{Unif}(\cdot)$ denote the uniform distribution over its argument. For distributions $p_1$ and $p_2$, the cross entropy is defined as

$$\mathrm{CrossEnt}(p_1, p_2) := -\mathbb{E}_{p_1}[\ln p_2].$$

We now write the loss function $\mathcal{L}(\theta; z_r, z_f)$ for each method.

**Gradient Ascent (GradAscent)** (Dorna et al., 2025) maximizes the negative log-likelihood (NLL) loss on the forget set:

$$\mathcal{L}_{\mathrm{GradAscent}}(\theta; z_r, z_f) = \ln p_\theta(a_f \mid q_f)$$

**Gradient Difference (GradDiff)** (Dorna et al., 2025) minimizes the NLL on the retain set while maximizing the NLL on the forget set. For retain and forget loss weights $\alpha_r, \alpha_f \geq 0$:

$$\mathcal{L}_{\mathrm{GradDiff}}(\theta; z_r, z_f) = \alpha_f \ln p_\theta(a_f \mid q_f) - \alpha_r \ln p_\theta(a_r \mid q_r)$$

**Weighted Gradient Ascent (WGA)** (Wang et al., 2025) modulates the forget-set unlearning signal using a confidence-based weight $w_{\mathrm{WGA}}$ defined as

$$w_{\mathrm{WGA}}(z_f) = \mathrm{StopGrad}[p_\theta(a_f \mid q_f)^\beta].$$

The resulting objective then maximizes the weighted NLL on the forget set sample while minimizing the NLL on the retain set sample. For a retain and forget loss weight $\alpha_r, \alpha_f \geq 0$:

$$\mathcal{L}_{\mathrm{WGA}}(\theta; z_r, z_f) = \alpha_f \, w_{\mathrm{WGA}}(z_f) \ln p_\theta(a_f \mid q_f) - \alpha_r \ln p_\theta(a_r \mid q_r).$$

**Saturation and Importance (SatImp)** (Yang et al., 2025) performs a similar procedure to WGA, maximizing a weighted NLL over the forget set while minimizing the NLL over the retain set. The SatImp weight $w_{\text{SatImp}}$ is defined as

$$w_{\text{SatImp}}(\boldsymbol{z}_f) = \text{StopGrad}\Big[p_{\boldsymbol{\theta}}(\boldsymbol{a}_f \mid \boldsymbol{q}_f)^{\beta_1}\big(1 - p_{\boldsymbol{\theta}}(\boldsymbol{a}_f \mid \boldsymbol{q}_f)\big)^{\beta_2}\Big],$$

where $\beta_1, \beta_2 \geq 0$ control the strength of the two weights. For retain and forget loss weights $\alpha_r, \alpha_f \geq 0$, the resulting objective is then

$$\mathcal{L}_{\text{SatImp}}(\boldsymbol{\theta}\,;\,\boldsymbol{z}_r, \boldsymbol{z}_f) = \alpha_f\, w_{\text{SatImp}}(\boldsymbol{z}_f) \ln p_{\boldsymbol{\theta}}(\boldsymbol{a}_f \mid \boldsymbol{q}_f) - \alpha_r \ln p_{\boldsymbol{\theta}}(\boldsymbol{a}_r \mid \boldsymbol{q}_r).$$

**Unlearning via Self-Distillation on Adjusted Logits (UnDIAL)** (Dong et al., 2025) perturbs the original model's distribution and distills this into the unlearned model. Consider a single next-token $y_f$ which completes the context $\boldsymbol{x}_f$ from the forget set, meaning that the sequence $(\boldsymbol{x}_f, y_f)$ is a subsequence of the question-answer pair $(\boldsymbol{q}_f, \boldsymbol{a}_f)$. We denote the perturbed distribution conditioned on the forget set context $\boldsymbol{x}_f$ as $\tilde{p}_{\boldsymbol{\theta}^*}(\cdot \mid \boldsymbol{x}_f)$, defined by

$$\ln \tilde{p}_{\boldsymbol{\theta}^*}(y \mid \boldsymbol{x}_f) = \mathbb{1}\{y = y_f\}\left(\ln p_{\boldsymbol{\theta}^*}(y \mid \boldsymbol{x}_f) - \beta\right) + \mathbb{1}\{y \neq y_f\} \ln p_{\boldsymbol{\theta}^*}(y \mid \boldsymbol{x}_f) + C(\boldsymbol{x}_f),$$

where $\beta \geq 0$ subtracts from the original model's logit assigned the true response and $C(\boldsymbol{x}_f)$ is a normalization constant. For retain and forget loss weights $\alpha_r, \alpha_f \geq 0$, the resulting UnDIAL objective is then

$$\mathcal{L}_{\text{UnDIAL}}(\boldsymbol{\theta}\,;\,\boldsymbol{z}_r, \boldsymbol{z}_f) = \alpha_f \text{CrossEnt}\left(\tilde{p}_{\boldsymbol{\theta}^*}, p_{\boldsymbol{\theta}}\right) - \alpha_r \ln p_{\boldsymbol{\theta}}(\boldsymbol{a}_r \mid \boldsymbol{q}_r).$$

**Representation Misdirection for Unlearning (RMU)** (Li et al., 2024) performs unlearning by steering internal representations of forget set inputs away from their original values, while preserving representations on the retain set.

Fix a layer index $k$ and let $M_{\boldsymbol{\theta}}^k(\boldsymbol{q}) \in \mathbb{R}^{d \times L}$ denote the hidden representations at layer $k$ of model $p_{\boldsymbol{\theta}}$ for input question $\boldsymbol{q}$, where $d$ is the hidden dimension and $L$ is the sequence length. RMU samples a random "control vector" $\boldsymbol{u} \sim \text{Unif}([0, 1]^d)$ once at the start of unlearning and rescales it to have $\ell_2$ norm equal to some $c > 0$. Let $\mathbf{1}_L \in \mathbb{R}^L$ denote the vector of all ones.

The RMU objective encourages forget set representations to align with this fixed random target, while constraining retain set representations to remain close to those of the original pretrained model. For a norm constraint $c > 0$ and loss weights $\alpha_r, \alpha_f \geq 0$, the resulting RMU objective is then

$$\mathcal{L}_{\text{RMU}}(\boldsymbol{\theta}\,;\,\boldsymbol{z}_r, \boldsymbol{z}_f) = \alpha_f \left\|M_{\boldsymbol{\theta}}^k(\boldsymbol{q}_f) - \frac{c}{\|\boldsymbol{u}\|_2}\boldsymbol{u}\mathbf{1}_L^\top\right\|_F^2 + \alpha_r \left\|M_{\boldsymbol{\theta}}^k(\boldsymbol{q}_r) - M_{\boldsymbol{\theta}^*}^k(\boldsymbol{q}_r)\right\|_F^2.$$

**Unlearning from Logit Difference (ULD)** (Ji et al., 2024) introduces an auxiliary assistant LLM $f_{\boldsymbol{\phi}}(\boldsymbol{q}, \boldsymbol{a})$ that outputs a distribution over next tokens, so $f_{\boldsymbol{\phi}}(\boldsymbol{q}, \boldsymbol{a})$ assigns a probability to $\boldsymbol{a} \mid \boldsymbol{q}$. Formally, $f_{\boldsymbol{\phi}}(\boldsymbol{x}, \cdot) \in \mathcal{P}(\mathcal{V})$, where $\mathcal{P}(\mathcal{V})$ denotes the probability simplex over the vocabulary. This assistant is used to tilt the frozen base model distribution $p_{\boldsymbol{\theta}^*}(\cdot \mid \boldsymbol{q})$, yielding the updated model

$$\hat{p}_{\boldsymbol{\theta}^*, \boldsymbol{\phi}}(\boldsymbol{a} \mid \boldsymbol{q}) \propto \frac{p_{\boldsymbol{\theta}^*}(\boldsymbol{a} \mid \boldsymbol{q})}{f_{\boldsymbol{\phi}}(\boldsymbol{q}, \boldsymbol{a})^T},$$

where $T > 0$ controls the strength of the tilt. The base model $p_{\boldsymbol{\theta}^*}$ remains fixed, and only the assistant $f_{\boldsymbol{\phi}}$ is trained. In ULD, $f_{\boldsymbol{\phi}}$ is implemented as a rank-$r$ LoRA (Hu et al., 2022) adaptation of the first $k$ layers of $p_{\boldsymbol{\theta}^*}$, with the remaining layers discarded. For retain and forget loss weights $\alpha_r, \alpha_f \geq 0$, the assistant is trained using the ULD objective

$$\mathcal{L}_{\text{ULD}}(\boldsymbol{\phi}\,;\,\boldsymbol{z}_r, \boldsymbol{z}_f) = -\alpha_f \ln f_{\boldsymbol{\phi}}(\boldsymbol{q}_f, \boldsymbol{a}_f) + \alpha_r \text{KL}\left(\text{Unif}(\mathcal{V}) \,\|\, f_{\boldsymbol{\phi}}(\boldsymbol{q}_r, \cdot)\right).$$

**I Don't Know Direct Preference Optimization (IdkDPO)** (Maini et al., 2024) applies the DPO (Rafailov et al., 2023) objective to the forget set samples. It predefines a set of responses like "I don't know" or "I can't answer that" which are used as the preferred label $\tilde{\boldsymbol{a}}_f$, while the original forget set ground truth $\boldsymbol{a}_f$ is used as the dispreferred label $\boldsymbol{a}_f$. The original model before unlearning $p_{\boldsymbol{\theta}^*}$ is used as the reference model. Additionally, it minimizes the NLL loss on the retain set. For a regularization coefficient $\beta \geq 0$ and loss weights $\alpha_r, \alpha_f \geq 0$, the IdkDPO objective is then

$$\mathcal{L}_{\text{IdkDPO}}(\boldsymbol{\theta}) = -\alpha_f \frac{2}{\beta} \ln \sigma \left(\beta \left(\ln \frac{p_{\boldsymbol{\theta}}(\tilde{\boldsymbol{a}}_f \mid \boldsymbol{q}_f)}{p_{\boldsymbol{\theta}^*}(\tilde{\boldsymbol{a}}_f \mid \boldsymbol{q}_f)} - \ln \frac{p_{\boldsymbol{\theta}}(\boldsymbol{a}_f \mid \boldsymbol{q}_f)}{p_{\boldsymbol{\theta}^*}(\boldsymbol{a}_f \mid \boldsymbol{q}_f)}\right)\right) - \alpha_r \ln p_{\boldsymbol{\theta}}(\boldsymbol{a}_r \mid \boldsymbol{q}_r).$$

**Negative Preference Optimization (NPO)** (Zhang et al., 2024) applies a DPO-like objective but without a preferred response. For a regularization coefficient $\beta \geq 0$ and loss weights $\alpha_r, \alpha_f \geq 0$, the NPO loss is defined as

$$\mathcal{L}_{\text{NPO}}(\boldsymbol{\theta}) = -\alpha_f \frac{2}{\beta} \ln \sigma \left( -\beta \ln \frac{p_{\boldsymbol{\theta}}(\boldsymbol{a}_f \mid \boldsymbol{q}_f)}{p_{\boldsymbol{\theta}^*}(\boldsymbol{a}_f \mid \boldsymbol{q}_f)} \right) - \alpha_r \ln p_{\boldsymbol{\theta}}(\boldsymbol{a}_r \mid \boldsymbol{q}_r).$$

**Simple NPO (SimNPO)** (Fan et al., 2025) modifies the NPO objective by applying length normalization and removing the reliance on the reference model $p_{\boldsymbol{\theta}^*}$. For a regularization coefficient $\beta \geq 0$, reward margin parameter $\Delta \geq 0$, and loss weights $\alpha_r, \alpha_f \geq 0$, the SimNPO objective is defined as

$$\mathcal{L}_{\text{SimNPO}}(\boldsymbol{\theta}) = -\alpha_f \frac{2}{\beta} \ln \sigma \left( -\frac{\beta}{|\boldsymbol{a}_f|} \ln p_{\boldsymbol{\theta}}(\boldsymbol{a}_f \mid \boldsymbol{q}_f) - \Delta \right) - \alpha_r \ln p_{\boldsymbol{\theta}}(\boldsymbol{a}_r \mid \boldsymbol{q}_r).$$

### F.2. TOFU Metrics

Recall that for a model $p_{\boldsymbol{\theta}}$ and a question-answer pair $(\boldsymbol{q}, \boldsymbol{a})$, the Truth Ratio, denoted $R_{\text{truth}}(p_{\boldsymbol{\theta}}, \boldsymbol{q})$, measures the ratio of the model's average probability on a set of incorrect "perturbed" answers $\mathcal{A}_{\text{pert}}$ to the paraphrased true answer $\boldsymbol{a}^\dagger$:

$$R_{\text{truth}}(p_{\boldsymbol{\theta}}, \boldsymbol{q}) = \frac{\frac{1}{|\mathcal{A}_{\text{pert}}|} \sum_{\tilde{\boldsymbol{a}} \in \mathcal{A}_{\text{pert}}} p_{\boldsymbol{\theta}}(\tilde{\boldsymbol{a}} \mid \boldsymbol{q})^{1/|\tilde{\boldsymbol{a}}|}}{p_{\boldsymbol{\theta}}(\boldsymbol{a}^\dagger \mid \boldsymbol{q})^{1/|\boldsymbol{a}^\dagger|}}.$$

Using the Truth Ratio, TOFU (Maini et al., 2024) defines two summary metrics: *Forget Quality* and *Model Utility*. We define these metrics in detail.

**Forget Quality.** Forget Quality quantifies whether the unlearned model's behavior on the forget set is statistically indistinguishable from that of a gold-standard model retrained exclusively on retained data. Specifically, it is defined as the p-value of a two-sample Kolmogorov-Smirnov (KS) test comparing the distributions of Truth Ratios generated by the unlearned model versus the retrained model over the forget set questions.

Specifically, let $\boldsymbol{\theta}_r^*$ denote the ground truth retained model parameters. We then define two empirical distributions:

$$\hat{R}_{\text{Truth}} = R_{\text{truth}}(p_{\boldsymbol{\theta}}, \boldsymbol{q}), \quad \boldsymbol{q} \sim \mathcal{D}_f$$
$$R_{\text{Truth}}^* = R_{\text{truth}}(p_{\boldsymbol{\theta}_r^*}, \boldsymbol{q}), \quad \boldsymbol{q} \sim \mathcal{D}_f,$$

The forget quality is computed as the p-value of the two-sample Kolmogorov–Smirnov test between $\hat{R}_{\text{Truth}}$ and $R_{\text{Truth}}^*$. The sufficient statistic of the KS-Test is the supremum absolute error between the CDFs of the two distributions $\hat{R}_{\text{Truth}}$ and $R_{\text{Truth}}^*$. Define the two respective CDFs as $\hat{F}$ and $F_r^*$. Then we compute $D_{\text{KS}}$ as

$$D_{\text{KS}} = \sup_t \left| \hat{F}(t) - F_r^*(t) \right|$$

Then using an approximation for the p-value, which is exactly expressed as a series, we have that the Forget Quality is approximately expressed as $2 \exp \left( -|\mathcal{D}_f| \cdot D_{\text{KS}}^2 \right)$.

A high p-value indicates that the unlearned model successfully mimics the output statistics of a model trained without $\mathcal{D}_f$.

**Model Utility.** Model Utility is calculated as the harmonic mean of the *Probability*, *ROUGE*, and *TR+* metrics across the retain, Real Author (RA), and World Facts (WF) datasets. For a model $p_{\boldsymbol{\theta}}$ and a question-answer pair $(\boldsymbol{q}, \boldsymbol{a})$, we define these metrics in detail.

- **Probability** computes the probability assigned to the answer given the question $p_{\boldsymbol{\theta}}(\boldsymbol{a} \mid \boldsymbol{q})^{1/|\boldsymbol{a}|}$, normalized by answer length. For the retain set it is computed as
$$p_{\boldsymbol{\theta}}(\boldsymbol{a} \mid \boldsymbol{q})^{1/|\boldsymbol{a}|}.$$

  For the multiple choice RA and WF datasets, the Probability metric is additionally normalized by sum of the length-normalized probabilities assigned to a set of incorrect answers $\mathcal{A}_{\text{pert}}$, computed as

$$\frac{p_{\boldsymbol{\theta}}(\boldsymbol{a} \mid \boldsymbol{q})^{1/|\boldsymbol{a}|}}{p_{\boldsymbol{\theta}}(\boldsymbol{a} \mid \boldsymbol{q})^{1/|\boldsymbol{a}|} + \sum_{\tilde{\boldsymbol{a}} \in \mathcal{A}_{\text{pert}}} p_{\boldsymbol{\theta}}(\tilde{\boldsymbol{a}} \mid \boldsymbol{q})^{1/|\tilde{\boldsymbol{a}}|} + \epsilon},$$

  where $\epsilon = 10^{-10}$ is added for numerical stability.

- **ROUGE** computes the ROUGE-L recall (Lin, 2004) which measures the overlap between the model's generated response $\hat{a} \sim p_\theta(\cdot \mid q)$ and the ground truth answer $a$. Let $\text{LCS}(\hat{a}, a)$ denote the longest common subsequence between the two sequences after word stemming. ROUGE-L recall is then computed as the length of the longest common subsequence divided by the reference text length:

$$\frac{|\text{LCS}(\hat{a}, a)|}{|a|}.$$

  In our experiments, responses are generated using greedy decoding. As a result, $\hat{a}$ should be interpreted as a deterministic decode from $p_\theta(\cdot \mid q)$ rather than a formal sample from the model distribution.

- **TR+** computes the inverted, clipped Truth Ratio, where higher values indicate larger confidence in the correct answer:

$$\text{TR+}(p_\theta, q) = \max\{0, \, 1 - R_{\text{truth}}(p_\theta, q)\}.$$

### F.3. TOFU Experiments

We explain our experimental setup in detail. To evaluate each unlearning method, we perform a grid search over hyperparameter configurations using random seeds 1 and 2, selecting the configuration that achieves the best performance. Since we aim to maximize multiple objectives, we select the parameter configuration that maximizes the Forget Quality, since a method optimized for a model utility metric could simply return the original model without any unlearning.

We used the OpenUnlearning (Dorna et al., 2025) implementation of TOFU and all baselines except for ULD which does not have an official OpenUnlearning implementation. For consistency, we implemented ULD within the OpenUnlearning codebase. We used the Llama 3.1 8B model.

All experiments were conducted on a single NVIDIA GH200, using the AdamW optimizer (Loshchilov & Hutter, 2019) and a linear learning rate scheduler with both warmup and decay for all methods. For the parameter-efficient methods ULD and T3-Unlearning, we use a batch size of 32 with no gradient accumulation and ZeRO stage 0 (Rajbhandari et al., 2020). For all other methods, we use a batch size of 8 with gradient accumulation over 4 steps (effective batch size 32) and ZeRO stage 3.

#### F.3.1. Hyperparameter Search

We report the hyperparameter grids on chosen parameters used for each method on each of the 5% and 10% forget set splits of TOFU. We denote the learning rate as $\eta$, number of epochs $N$, number of warmup epochs $N_{\text{warmup}}$, weight decay as $\lambda$, and the retain and forget loss weights as $\alpha_r$ and $\alpha_f$, respectively.

**GradAscent**

*Search space:* $\eta \in [10^{-7}, 10^{-5}]$, $N \in [10, 20]$, $N_{\text{warmup}} = 1$, $\lambda = 10^{-2}$.

*Chosen hyperparameters:*

- 5%: $\eta = 10^{-6}$, $N = 10$
- 10%: $\eta = 10^{-7}$, $N = 10$

**GradDiff**

*Search space:* $\eta \in [10^{-5}, 10^{-4}]$, $N \in [10, 20]$, $\alpha_r \in [0.2, 1.0]$, $\alpha_f \in [0.0, 0.8]$, $N_{\text{warmup}} = 1$, $\lambda = 10^{-2}$.

*Chosen hyperparameters:*

- 5%: $\eta = 10^{-4}$, $N = 20$, $\alpha_r = 1.0$, $\alpha_f = 0.0$
- 10%: $\eta = 10^{-4}$, $N = 10$, $\alpha_r = 0.8$, $\alpha_f = 0.2$

**WGA**

*Search space:* $\eta \in [10^{-5}, 10^{-4}]$, $N \in [5, 10]$, $N_{\text{warmup}} = 1$, $\alpha_r \in [0.1, 3.0]$, $\alpha_f = 1$, $\beta \in [0.1, 5.0]$, $\lambda = 10^{-2}$.

*Chosen hyperparameters:*

- 5%: $\eta = 10^{-4}$, $N = 10$, $\beta = 5.0$, $\alpha_r = 3.0$
- 10%: $\eta = 10^{-5}$, $N = 5$, $\beta = 5.0$, $\alpha_r = 0.1$

**SatImp**

*Search space:* $\eta \in [10^{-5}, 10^{-4}]$, $N \in [5, 10]$, $N_{\mathrm{warmup}} = 1$, $\alpha_r \in [0.1, 3.0]$, $\alpha_f = 1$, $\beta_1, \beta_2 \in [0.1, 3.0]$, $\lambda = 10^{-2}$.

*Chosen hyperparameters:*

- 5%: $\eta = 10^{-4}$, $N = 10$, $\alpha_r = 1.0$, $\beta_1 = 1.0$, $\beta_2 = 1.0$
- 10%: $\eta = 10^{-4}$, $N = 10$, $\alpha_r = 1.0$, $\beta_1 = 1.0$, $\beta_2 = 1.0$

**UnDIAL**

*Search space:* $\eta \in [10^{-5}, 10^{-4}]$, $N \in [5, 10]$, $N_{\mathrm{warmup}} = 1$, $\alpha_r \in [0.1, 3.0]$, $\alpha_f = 1$, $\beta \in [2.0, 16.0]$, $\lambda = 10^{-2}$.

*Chosen hyperparameters:*

- 5%: $\eta = 10^{-4}$, $N = 5$, $\alpha_r = 1.0$, $\beta = 16.0$
- 10%: $\eta = 10^{-5}$, $N = 10$, $\alpha_r = 3.0$, $\beta = 16.0$

**RMU**

*Search space:* $\eta \in [10^{-5}, 10^{-4}]$, $N \in [5, 10]$, $N_{\mathrm{warmup}} = 1$, $\alpha_r \in [0.1, 3.0]$, $\alpha_f = 1$, control vector norm $c \in [1.0, 10.0]$, number of layers $k \in [8, 32]$ (note: Llama 3.1 8B has 32 layers), $\lambda = 10^{-2}$.

*Chosen hyperparameters:*

- 5%: $\eta = 10^{-4}$, $N = 10$, $\alpha_r = 1.0$, $c = 1.0$, $k = 32$
- 10%: $\eta = 10^{-5}$, $N = 10$, $\alpha_r = 0.1$, $c = 1.0$, $k = 32$

**ULD**

*Search space:* $\eta \in [10^{-3}, 10^{-2}]$, $N \in [10, 20]$, $N_{\mathrm{warmup}} = 1$, $\alpha_r \in [0.1, 5.0]$, $\alpha_f = 1$, LoRA rank $r \in [16, 32]$, applied to first $k \in [8, 32]$ layers, tilting strength $T \in [0.5, 2.0]$, $\lambda = 10^{-4}$.

We perform LoRA without bias vectors, setting the dropout to $0.05$ and the "alpha" scaling equal to the rank $r$. We additionally swept the logit filter proportion in $[0.01, 0.5]$ (masking low-probability tokens under the base distribution), but observed that applying no filter yielded the best performance.

*Chosen hyperparameters:*

- 5%: $\eta = 10^{-3}$, $N = 20$, $\alpha_r = 0.1$, $r = 32$, $k = 16$, $T = 1.0$
- 10%: $\eta = 10^{-3}$, $N = 20$, $\alpha_r = 0.1$, $r = 32$, $k = 16$, $T = 1.0$

**IdkDPO**

*Search space:* $\eta \in [10^{-5}, 10^{-4}]$, $N \in [5, 10]$, $N_{\mathrm{warmup}} = 1$, $\alpha_r \in [0.1, 3.0]$, $\alpha_f = 1$, $\beta \in [0.1, 3.0]$, $\lambda = 10^{-2}$.

*Chosen hyperparameters:*

- 5%: $\eta = 10^{-4}$, $N = 10$, $\alpha_r = 3.0$, $\beta = 0.1$
- 10%: $\eta = 10^{-4}$, $N = 10$, $\alpha_r = 0.1$, $\beta = 0.1$

**NPO**

*Search space:* $\eta \in [10^{-5}, 10^{-4}]$, $N \in [10, 20]$, $N_{\mathrm{warmup}} = 1$, $\alpha_r = 1$, $\alpha_f \in [0.5, 1.5]$, $\beta \in [0.05, 0.2]$, $\lambda = 10^{-2}$.

*Chosen hyperparameters:*

- 5%: $\eta = 10^{-5}$, $N = 20$, $\alpha_f = 1.5$, $\beta = 0.1$
- 10%: $\eta = 10^{-4}$, $N = 10$, $\alpha_f = 1.5$, $\beta = 0.1$

**SimNPO**

*Search space:* $\eta = 10^{-5}$, $N = 10$, $N_{\mathrm{warmup}} = 1$, $\alpha_r \in [0.05, 0.25]$, $\alpha_f \in [0.5, 1.5]$, regularization $\beta \in [1.5, 5.5]$, reward margin $\Delta \in [0, 2.0]$, $\lambda = 10^{-2}$.

*Chosen hyperparameters:*

- 5%: $\alpha_r = 0.15$, $\alpha_f = 0.5$, $\beta = 3.5$, $\Delta = 1.0$
- 10%: $\alpha_r = 0.25$, $\alpha_f = 1.0$, $\beta = 5.5$, $\Delta = 1.0$

*Table 4.* Forget Quality, MU-ROUGE, and Model Utility (MU) metrics across random seeds for each unlearning method on the TOFU benchmark using Llama 3.1 8B. Larger values indicate better performance for all metrics. The bottom shaded row corresponds to our method T3-Unlearning.

| Split | Method | Forget Quality | | | | | MU-ROUGE | | | | | Model Utility | | | | |
|---|---|---|---|---|---|---|---|---|---|---|---|---|---|---|---|---|
| | | Seed 1 | Seed 2 | Seed 3 | Seed 4 | Seed 5 | Seed 1 | Seed 2 | Seed 3 | Seed 4 | Seed 5 | Seed 1 | Seed 2 | Seed 3 | Seed 4 | Seed 5 |
| 5% | GradAscent | 0.000 | 0.000 | 0.000 | 0.000 | 0.000 | 0.881 | 0.874 | 0.877 | 0.873 | 0.873 | 0.580 | 0.573 | 0.575 | 0.575 | 0.575 |
| | GradDiff | 0.004 | 0.001 | 0.006 | 0.003 | 0.000 | 0.187 | 0.336 | 0.275 | 0.277 | 0.328 | 0.300 | 0.401 | 0.356 | 0.367 | 0.393 |
| | WGA | 0.178 | 0.221 | 0.142 | 0.394 | 0.328 | 0.396 | 0.465 | 0.373 | 0.376 | 0.509 | 0.411 | 0.428 | 0.389 | 0.402 | 0.462 |
| | SatImp | 0.394 | 0.545 | 0.394 | 0.178 | 0.713 | 0.449 | 0.374 | 0.404 | 0.425 | 0.444 | 0.418 | 0.396 | 0.408 | 0.423 | 0.416 |
| | UnDIAL | 0.040 | 0.004 | 0.030 | 0.002 | 0.004 | 0.638 | 0.678 | 0.639 | 0.645 | 0.668 | 0.566 | 0.574 | 0.551 | 0.561 | 0.581 |
| | RMU | 0.545 | 0.628 | 0.545 | 0.545 | 0.394 | 0.000 | 0.000 | 0.000 | 0.000 | 0.000 | 0.000 | 0.000 | 0.000 | 0.000 | 0.000 |
| | ULD | 0.016 | 0.221 | 0.270 | 0.068 | 0.270 | 0.948 | 0.944 | 0.939 | 0.946 | 0.944 | 0.648 | 0.642 | 0.643 | 0.644 | 0.642 |
| | IdkDPO | 0.713 | 0.270 | 0.545 | 0.221 | 0.924 | 0.693 | 0.660 | 0.670 | 0.635 | 0.685 | 0.552 | 0.552 | 0.548 | 0.532 | 0.542 |
| | NPO | 0.713 | 0.866 | 0.793 | 0.965 | 0.628 | 0.699 | 0.762 | 0.738 | 0.692 | 0.663 | 0.626 | 0.672 | 0.663 | 0.587 | 0.606 |
| | SimNPO | 0.628 | 0.793 | 0.394 | 0.988 | 0.924 | 0.778 | 0.818 | 0.793 | 0.742 | 0.798 | 0.649 | 0.650 | 0.689 | 0.617 | 0.662 |
| | T3-Unlearning | 0.924 | 0.924 | 0.965 | 0.793 | 0.965 | 0.890 | 0.907 | 0.896 | 0.903 | 0.903 | 0.612 | 0.611 | 0.606 | 0.618 | 0.614 |
| 10% | GradAscent | 0.000 | 0.000 | 0.000 | 0.000 | 0.000 | 0.951 | 0.948 | 0.951 | 0.951 | 0.951 | 0.637 | 0.638 | 0.638 | 0.638 | 0.638 |
| | GradDiff | 0.322 | 0.000 | 0.000 | 0.000 | 0.002 | 0.000 | 0.000 | 0.000 | 0.000 | 0.000 | 0.000 | 0.000 | 0.000 | 0.000 | 0.000 |
| | WGA | 0.045 | 0.054 | 0.000 | 0.000 | 0.000 | 0.839 | 0.828 | 0.825 | 0.840 | 0.842 | 0.656 | 0.618 | 0.624 | 0.625 | 0.623 |
| | SatImp | 0.367 | 0.281 | 0.281 | 0.037 | 0.131 | 0.317 | 0.282 | 0.373 | 0.223 | 0.312 | 0.391 | 0.363 | 0.417 | 0.324 | 0.381 |
| | UnDIAL | 0.000 | 0.000 | 0.000 | 0.000 | 0.000 | 0.919 | 0.929 | 0.934 | 0.928 | 0.927 | 0.691 | 0.703 | 0.699 | 0.695 | 0.696 |
| | RMU | 0.000 | 0.001 | 0.000 | 0.004 | 0.002 | 0.000 | 0.000 | 0.000 | 0.000 | 0.000 | 0.000 | 0.000 | 0.000 | 0.000 | 0.000 |
| | ULD | 0.004 | 0.001 | 0.000 | 0.000 | 0.054 | 0.938 | 0.942 | 0.951 | 0.948 | 0.939 | 0.642 | 0.641 | 0.642 | 0.642 | 0.641 |
| | IdkDPO | 0.005 | 0.030 | 0.024 | 0.024 | 0.016 | 0.433 | 0.493 | 0.383 | 0.349 | 0.406 | 0.464 | 0.501 | 0.450 | 0.446 | 0.463 |
| | NPO | 0.094 | 0.367 | 0.322 | 0.078 | 0.065 | 0.550 | 0.482 | 0.535 | 0.500 | 0.527 | 0.499 | 0.488 | 0.506 | 0.489 | 0.486 |
| | SimNPO | 0.967 | 0.003 | 0.641 | 0.700 | 0.367 | 0.856 | 0.830 | 0.846 | 0.822 | 0.847 | 0.671 | 0.643 | 0.662 | 0.668 | 0.668 |
| | T3-Unlearning | 0.758 | 0.758 | 0.758 | 0.322 | 0.758 | 0.890 | 0.900 | 0.897 | 0.896 | 0.912 | 0.612 | 0.608 | 0.605 | 0.620 | 0.616 |

## T3-Unlearning

*Search space:* $\eta \in [10^{-4}, 10^{-3}]$, $N = 100$, $N_{\text{warmup}} = 25$, classifier hidden dimension $h \in [15, 500]$, base model temperature $T \in [1, 2.75]$, $\lambda \in [10^{-4}, 2 \cdot 10^{-2}]$.

*Chosen hyperparameters:*

- 5%: $\eta = 5 \cdot 10^{-4}$, $h = 20$, $T = 2.5$, $\lambda = 10^{-3}$
- 10%: $\eta = 5 \cdot 10^{-4}$, $h = 20$, $T = 2.5$, $\lambda = 10^{-3}$

### F.3.2. DETAILED RESULTS

We report results for each method across random seeds in Table 4, where Table 1 shows the corresponding averages. We observe that T3-Unlearning has much more stable performance than other baselines, leading to better cumulative performance.

*Table 5.* T3-Unlearning performance on the TOFU dataset across the 5% and 10% forget set splits for different temperatures $T$. Values are averaged over five seeds, where larger is better for all metrics. The $T = 2.5$ rows denote the temperature achieving the best Forget Quality and correspond to the results reported in Table 1.

| Split | Temperature $T$ | Forget Quality | MU-ROUGE | Model Utility |
|---|---|---|---|---|
| 5% | 1.0 | 0.000 | 0.936 | 0.641 |
| | 1.5 | 0.008 | 0.934 | 0.626 |
| | 2.0 | 0.586 | 0.905 | 0.644 |
| | 2.5 | 0.914 | 0.900 | 0.612 |
| | 3.0 | 0.458 | 0.902 | 0.457 |
| 10% | 1.0 | 0.000 | 0.942 | 0.642 |
| | 1.5 | 0.001 | 0.931 | 0.624 |
| | 2.0 | 0.191 | 0.893 | 0.646 |
| | 2.5 | 0.671 | 0.899 | 0.612 |
| | 3.0 | 0.576 | 0.910 | 0.468 |

### F.3.3. TEMPERATURE SENSITIVITY

We present an ablation study examining the effect of the base model temperature $T$ on the performance of T3-Unlearning on the TOFU benchmark. For each temperature, we perform a grid search to select hyperparameters using seeds 1 and 2, and report mean results over five total seeds in Table 5. Without tempering, corresponding to $T = 1$, T3-Unlearning fails to achieve non-zero Forget Quality, while excessively large temperatures also degrade overall performance. These results highlight the central role of tempering in both the theoretical analysis and empirical behavior of our method.

