# OpenReview forum: "Temper-Then-Tilt: Principled Unlearning for Generative Models through Tempering and Classifier Guidance"
_ICML.cc/2026/Conference — ICML 2026 regular_

### Official Review · Reviewer_3yfZ · 2026-03-08

**Soundness:** 4
**Presentation:** 4
**Significance:** 4
**Originality:** 4
**Overall Recommendation:** 5
**Confidence:** 3

**Summary:**

The authors propose a method to unlearn forget samples by first tempering the base distribution to smoothen sharp concentrated modes and then tilting the distribution according to a classifier with small number of parameters. The work includes proof of finite sample unlearning of forget set and outperforms baselines on with small overhead.

**Compliance With Llm Reviewing Policy:**

Affirmed.

**Key Questions For Authors:**

1 - Are the authors considering tempering the data used for classification e.g with reweighting samples from $p$ according to $p^{1/T}$ to train the classifier? This is not obvious from the text. If not, is there a reason not to do this?

2 - As a result of tempering is the model becoming less confident about the generated sequences? Can the authors explain the comparison in Table 1 regarding probability, recall and confidence with other baselines and highlight the main contributors to the order? would choosing a smaller $T$ improve the results?

**Limitations:**

yes

**Strengths And Weaknesses:**

Strength: The work studies the information leakage of forget set due to high concentration of the set due to excess classifier risk. And proposes tempering to remedy this problem which is backed by theory and empirical results. They provide detailed analysis of tempering related bias and forgetting the sharply distributed samples.

Presentation: The paper is well-written and structured and easy to follow. The notations are defined and the theorems are explained in simple language.

Significance: The problem of unlearning a forget set is a critical challenge in generative modeling and influential for data privacy.

Novelty: This work formulates the unlearning problem in terms of correction of the generative distribution, which leads to significant increase in efficiency of the proposed method for large generative models like LLMs. This method also increases flexibility of the desired forget set, not impacting the training phase of the generative modeling. And provides theoretical analysis on the leakage from the forget set.

---

> ### Author Rebuttal · Authors · 2026-03-31
>
> **(Q1) Are the authors considering…**
>
> T3-Unlearning does not apply reweighting during classification. The classifier is trained by minimizing the regularized cross-entropy loss in Eq. 3 to estimate the optimal classifier $f^\ast(z)=\mathbb{P}(s=1 \mid z)$. Our analysis is also agnostic to the training procedure for $\hat{f}$ and is expressed in terms of its excess risk $L(\hat{f}) - L(f^\ast)$ under the original data distribution $p$.
>
> The main reason we do not reweight samples according to a tempered marginal is that our analysis is formulated with respect to the original mixture distribution $p$. The excess risk is defined by $p$, and is then related to the unlearning metrics of interest: Retain Error (an expectation under $p_r$) and Forget Error (an expectation under $p_f$). Tempering is used at inference to modify the base distribution and flatten sharp modes in $p_f$, rather than to alter the classifier objective.
>
> Conceptually, reweighting samples by a function of $z$ during classifier training would change the regions of the space that are emphasized when fitting the classifier, but would not change the target $f^\ast$. It would instead yield guarantees with respect to a different risk under the tempered distribution $p^{1/T}$, which would require different analysis to relate back to the Retain and Forget Errors.
>
> We will update the paper to state: "The classifier is trained on samples from the original mixture distribution $p$, while the temperature $T$ is applied only at inference when forming the tilted estimator $\hat{p}_r^{(T)}$."
>
> **(Q2) As a result of tempering…**
>
> Yes, in general tempering can reduce confidence on generated sequences. However, we observe in Table 1 that while tempering can decrease absolute probabilities on the retain set sequences, this does not necessarily imply worse generation quality; on TOFU, MU-ROUGE remains strong at the selected temperature $T=2.5$. Further, the tempered model maintains strong probability on out of sample sequences representing knowledge the model should preserve after unlearning.
>
> The TOFU benchmark evaluates model confidence across three datasets: the Retain set (question-answer pairs about the fictitious authors used during training), a World Facts (WF) dataset capturing general knowledge, and a Real Author (RA) dataset covering knowledge about real-world authors. In Table 1, T3-Unlearning achieves among the highest probability scores on WF and RA, indicating strong preservation of knowledge outside the unlearning data. The T3-Unlearning Retain Probability is lower relative to some baselines like SimNPO which optimize a retain-set loss during unlearning. While such methods can maintain high confidence, this does not necessarily translate to better retention in generated outputs; in contrast, T3-Unlearning achieves the highest ROUGE on the Retain set and the best overall MU-ROUGE among methods with competitive Forget Quality.
>
> To better understand the role of temperature $T$, we report additional results across values of $T$ for the TOFU 5\% and 10\% splits (Table 1 corresponds to $T=2.5$).
>
> | Split | Temperature $T$ | Retain Prob. | WF Prob. | RA Prob.|
> |-------|------------------|--------|----|----|
> | 5%    | 1.0              | 0.991  | 0.481 | 0.410 |
> |       | 1.5              | 0.962  | 0.487 | 0.419 |
> |       | 2.0              | 0.799  | 0.520 | 0.472 |
> |       | 2.5              | 0.415  | 0.545 | 0.511 |
> |       | 3.0              | 0.140  | 0.522 | 0.506 |
> |-------|------------------|--------|----|----|
> | 10%   | 1.0              | 0.991  | 0.487 | 0.409 |
> |       | 1.5              | 0.962  | 0.491 | 0.415 |
> |       | 2.0              | 0.800  | 0.526 | 0.471 |
> |       | 2.5              | 0.423  | 0.543 | 0.506 |
> |       | 3.0              | 0.149  | 0.520 | 0.506 |
>
> Tempering lowers absolute confidence on retain samples, as when $T$ increases, the Retain probability decreases. In contrast, the WF and RA probabilities increase up to $T=2.5$ and then slightly decrease at $T=3.0$, indicating that moderate tempering improves generalization to out of distribution data while excessive tempering introduces additional bias.
>
> While choosing a smaller $T$ (closer to $T=1$) would increase Retain Probability, it significantly degrades Forget Quality (Table 5) and leads to worse performance on the WF and RA datasets. This behavior is consistent with our theoretical characterization of the tradeoff between Retain and Forget Error: tempering increases Retain Error (which corresponds to lower Retain Probability), but can improve Forget Error by reducing the mismatch between the estimated and ground truth retain set distribution over forget set samples, which is captured by the Forget Quality metric in TOFU.
>
> We will incorporate this additional discussion into the revised version of the paper.

---

> > ### Author Rebuttal · Reviewer_3yfZ · 2026-04-02
> >
> > The authors provided detailed answers to my concerns and questions about the method. I believe the novelty of forget set leakage analysis and use of tempering for this application has merit. The authors propose a more appropriate metric to evaluate the forget method and provide sufficient analysis and empirical results to support their claims. The paper well-written and strong in theory and fits the scope of the conference.

---

### Official Review · Reviewer_BHLz · 2026-03-13

**Soundness:** 3
**Presentation:** 3
**Significance:** 3
**Originality:** 3
**Overall Recommendation:** 4
**Confidence:** 3

**Summary:**

The paper studies the problem of unlearning -- given a pretrained generative model, we need to change it in a way that it stops generating  forget-set content, but keeps its performance outside of the forgetting set. They try to do it without fine-tuning the big model: the standard approach is to find a weight function that reweight the base distribution toward the retain data. The weighting function is constructed from the classifier of forget and retain data.

The authors demonstrate that this approach could potentially suffer if the forgetting set is a sharp, concentrated spike in a base model (formally, ||p_f||_infty is large) which can result in a leakage of forget examples during the inference. The paper proposes the inference-time T3-unlearning approach: the initial distribution needs to be tempered first (p_T ~ p_0^(1/T)) to reduce sharpness of peaks and tilt only after that. The authors demonstrate theoretically that this reduces the forget error regime in the worst-case bounds. They support their claim empirically by testing the approach on TOFU benchmark.

**Compliance With Llm Reviewing Policy:**

Affirmed.

**Final Justification:**

The authors' rebuttal addressed my main questions and clarified the scope of the paper. This made the contribution and empirical claims more precise, but didn't significantly change my overall assessment. Based on my reading I continue to view this paper as technically solid, clearly written and particularly relevant to the development of modern LLMs. At the same time, some limitations remain. In particular, the theoretical analysis is mainly worst-case, and the significance of the results depends on a specific threat model. There is also still no clear practical rule for choosing the temperature. The rebuttal reinforces my prior recommendation to weakly accept the paper.

**Key Questions For Authors:**

None

**Limitations:**

yes

**Strengths And Weaknesses:**

Soundness. The method is clearly defined and easy to reproduce: frozen base model + lightweight classifier + tempered guidance at inference. The main theoretical claims are derived using standard tools in the Appendix, and the presence of Theorem 3.5 with the lower-bound construction makes the "worst-case bound" argument more credible, showing that ||p_f||_infty dependence is not a proof artifact. Experiments on TOFU benchmark are appropriate and support the claimed forget-utility tradeoff empirically. However, the theory is largely a worst-case analysis parametrized by classifier risk, and the exact choice of the temperature T is ambiguous: it is unclear how T should be selected without tuning on validation subset.

Presentation. The paper is generally well-written and easy to follow. The abstract could be hard to follow for a broad ML audience, but the introduction contains the clear motivation and all the definitions. My main concern is that the main part includes heavy mathematical bounds that can obscure the intuition. I think some of them could be moved to the appendix, replacing them with the higher-level explanation in the main body (e.g. in Theorem 3.8).

Significance. The work is practically relevant. It proposes the method that makes the model "forget" the data at inference time with a simple approach that avoids heavy model fine-tuning. The paper also has an interesting idea and a justification on why the spikiness of the forget-set could lead to the failure of the standard classifier-based approach, motivating tempering. The impact could be limited because the method doesn't delete the forget-set from the base model's weights and the leakage is still possible; also, the empirical validation is restricted to the TOFU benchmark, and broader evaluations could strengthen the claims further.

Originality. Density-ratio tilting, classifier guidance and temperature scaling are very standard; the main novelty of the paper is the analysis + method: the identification of  a finite-sample failure mode when the forget distribution is highly concentrated, and the idea + theoretical support that tempering can mitigate this problem.

---

> ### Author Rebuttal · Authors · 2026-03-31
>
> **(W1) The exact choice of the temperature T…**
>
> From a theoretical perspective, our analysis provides guidance on how $T$ mediates the tradeoff between Retain and Forget Errors (Eqs. 6 and 7). In particular, Theorems 3.7 and 3.8 characterize how these errors vary with $T$ through upper bounds on the tempered estimator $\hat{p}_r^{(T)}$. These results yield insights on exactly when tempering the original model is useful and how to select $T$ to balance the two error metrics.
>
> However, these bounds depend on properties of the underlying data distribution, such as the sharpness $\|p_f\|_{\infty}$, entropy $H(p_r)$, and the classifier excess risk $\delta$, which are generally not accessible in practice. As a result, directly optimizing these bounds to select $T$ could be intractable.
>
> In practice, this leads to a setting similar to existing LLM unlearning approaches, where $T$ and related hyperparameters (e.g., learning rate, regularization strength $\lambda$, and classifier rank $h$) are selected via empirical tuning. Importantly, $T$ is applied purely at inference time and is decoupled from the classifier training procedure. Consequently, in a deployed setting, our method does not require committing to a fixed temperature, and instead enables post-hoc or online tuning. In particular, $T$ can be adjusted on the fly without requiring retraining. Empirically, on the TOFU benchmark, we observe that $T = 2.5$ performs best, suggesting that in practice a small discrete search over values within the range $T \in [1, 2.5]$ is sufficient to identify a strong candidate, as the effective search space is limited. However, we acknowledge that this is an empirical observation and not theoretically justified.
>
> In general, developing methods for principled hyperparameter selection without access to retrained models or underlying data statistics remains an important open problem, and while our work does not fully resolve this challenge, we hope that the theoretical characterization we provide offers a useful starting point for understanding and guiding this tradeoff.
>
> **(W2) My main concern...**
>
> We will update the revised version of the paper to improve the readability and intuitive understanding of the key takeaways from the theoretical analysis.
>
> **(W3) The impact could be limited…**
>
> As you mention, our method T3-Unlearning leaves the base model frozen and instead learns to tilt its output distribution to match the distribution over the retain set samples $p_r$. As such, **we adopt a distributional perspective, where a potential attacker interacts with the model only through its outputs, including logits, but cannot observe its weights**. This is the predominant use case for current language models at scale, where users send prompts and can only see the model's generated text, and can at most see the model's logits. Accordingly, several recent works on LLM unlearning [1], model security [2], and standard unlearning benchmarks like TOFU and MUSE apply the same logit-access threat model and evaluate unlearning solely through model outputs.
>
> Under this threat model, the goal is to assess whether the unlearned model’s output distribution aligns with that of a model trained solely on the retain set. To formalize this objective, we define the Retain and Forget Errors (Eqs. 6 and 7), which capture two complementary notions of error when using an estimated distribution $\hat{p}_r$ to approximate the ground-truth $p_r$. Our theoretical analysis establishes finite-sample bounds on both quantities for the distribution induced by our method.
>
> In the setting where the attacker only has logit-access, our method outperforms even unlearning methods where the base model weights are updated. Specifically, on the TOFU benchmark, our method achieves the strongest Forget Quality, the primary TOFU metric which measures statistical indistinguishability between the unlearned model outputs and those of a retrained reference. Thus, although T3-Unlearning operates on the frozen base model outputs rather than modifying its weights as in prior methods, it more effectively limits an attacker’s ability to extract information about the forget set from the model’s outputs.
>
> We will update the revised paper to clarify these considerations and our adoption of a logit-access threat model.
>
> [1] "Reversing the forget-retain objectives: An efficient LLM unlearning framework from logit difference." (Ji et al, 2024)
>
> [2] "Stealing Part of a Production Language Model." (Carlini et al, 2024)

---

> > ### Author Rebuttal · Reviewer_BHLz · 2026-04-04
> >
> > I have read the rebuttal, and it addresses my questions. In particular, it clarifies the scope of the method and the underlying threat model, which helped make the paper's claims more precise.

---

### Official Review · Reviewer_GFTS · 2026-03-13

**Soundness:** 2
**Presentation:** 3
**Significance:** 3
**Originality:** 3
**Overall Recommendation:** 3
**Confidence:** 4

**Summary:**

This work has an interesting veiw on unlearning, that the original model distribution is a mixture of the model decomposed based on the latent membership variable. The key tool is 1) the likelihood ratio trick 2) the tilting idea to mitigate the potential failure of unlearning.

**Compliance With Llm Reviewing Policy:**

Affirmed.

**Key Questions For Authors:**

1. In Section 2.1, it is unclear if $p_r(z)\propto p(z) P(s=1|z)$ simplifies to $p_r(y|x)\propto p(y|x) P(s=1|x,y)$, since it would make sense with an additional assumption that $p_r(x)=p(x)$, but is it the case? Likewise, in Section 2.2, it is unclear if $\hat{p}_r^T(z)\propto p(z)^{1/T}\hat{f}(z)$ transfers to $\hat{p}_r^{(T)}(y\mid x)\propto p_t(y\mid z)^{1/T}\hat{f}(x,y)$, where $p_t$ denotes p\_theta. t would need to assume that $\hat{p}_r^{(T)}(x) =  p_t(x)^{1/T}$. However, is this the case? I doubt.

2. When the classifier is trained with the regularizer, can we say that it can truly learn the intended likelihood ratio? I guess not. Also, because of the added regularizer, there is a discrepancy in theory. The core assumption in their theory is that $L(\hat{f})-L(f^*)\leq \delta$, while $L(f) = \mathbb{E}_{z,s}[\ell(f(z),s)]$ itself does not contain the regularizer term. How can we say that the assumption would make sense?

3. Why is it important to forget the error defined in (7)? For the point that is to be forgotten, one would not mind what performance of the reteind model would have on it. What is important would be that $\hat{p}_r(z)$ is very different from $p(z)$ on the forgotten points, which does not necessarily mean $p_r(z)$ is very similar to $\hat{p}_r(z)$.

**Limitations:**

No. I don't see any that they discuss.

**Strengths And Weaknesses:**

I think the topic of unlearning seems important and interesting, and the application of the likelihood ratio trick is exciting.

However, I wonder if their way of doing it is the right way of doing it. If we simply tilt the original model, would there be any way that the attacker could get the original model's information? It is different from completely swapping the original model for another unlearn-ed one. I think this is the biggest concern.

I think the justification of the proposed method seems weak. Their justification is that if we don't "tilt", then there are cases where the "forget error" can be lower-bounded. But I am unsure how relevant this metric "forget error" is. And thus, I don't see the motivation of increasing the upper bound of the retention error (Theorem 3.8) in order to "tilt".

---

> ### Author Rebuttal · Authors · 2026-03-31
>
> **(W1) If we simply tilt...**
>
> Our method freezes the base model and tilts its distribution to match the retain set distribution $p_r$. We assume **an attacker only observes model outputs, including logits, but not model weights**. This matches the main LLM setting, where users observe only generated text (and possibly logits).
>
> Unlearning benchmarks (TOFU, MUSE) adopt this logit-access model and test unlearning via outputs only.
> On TOFU, our method outperforms methods that update base model weights and achieves the best Forget Quality score. Thus, T3-Unlearning better limits an attacker’s ability to extract forget set information from model outputs.
>
> We will update the paper to clarify this threat model.
>
> **(W2) I think the justification...in order to “tilt”'.**
>
> To clarify: “tilting” refers to reweighting $p(z)$ using $\hat{f}(z)$: without tilting, $p$ is unchanged and no unlearning occurs.
>
> We define two errors to measure how closely the tilted distribution matches the target: Retain Error, the discrepancy on retain set samples, and Forget Error, the discrepancy on forget set samples, both being equally important. Ideally both are zero, implying perfect unlearning.
>
> Theorem 3.5 shows a failure mode when only tilting, as it lower bounds the Forget Error when $p_f$ is sharp. To control both errors, we temper the base distribution before tilting. Theorem 3.7 shows that Temper-then-Tilt controls Forget Error even for sharp $p_f$, while Theorem 3.8 quantifies the added Retain Error from tempering.
>
> We will update the paper to clarify this motivation and tradeoff.
>
> **(Q1) In Section 2.1,...**
>
> We emphasize that we do not assume that $p_r(x) = p(x)$, or that  $\hat p_r^{(T)}(x) = p_{\theta^\ast}(x)^{1/T}$.
>
> We derive that $p_r(y \mid x) \propto p(y \mid x) \mathbb{P}(s=1 \mid x,y)$, and will update the paper to clarify our formulation. Define random variables $X$ as the token prefix, $Y$ as the next token, and $S$ as the membership variable where $S=1$ implies that $Z = (X,Y)$ was a retain set sample. $\mathbb{P}$ denotes the probability measure. By Bayes' Rule:
>
> $\mathbb{P}(Z=z \mid S=1) = \mathbb{P}(S=1 \mid Z=z) \cdot \mathbb{P}(Z=z) / \mathbb{P}(S=1)$.
>
> We defined $p_r(z) = \mathbb{P}(Z=z \mid S=1)$ and $p(z) = \mathbb{P}(Z=z)$, giving $p_r(z) \propto p(z) \cdot \mathbb{P}(s=1 \mid z)$, where $\mathbb{P}(s=1 \mid z)$ is shorthand for $\mathbb{P}(S=1 \mid Z=z)$.
>
> In the conditional setting, Bayes' Rule gives:
>
> $\mathbb{P}(Y=y \mid X=x, S=1) = \mathbb{P}(S=1 \mid X=x,Y=y) \cdot \mathbb{P}(Y=y \mid X=x) / \mathbb{P}(S=1 \mid X=x)$.
>
> We defined $p_r(y \mid x) = \mathbb{P}(Y=y \mid X=x, S=1)$ and $p(y \mid x) = \mathbb{P}(Y=y \mid X=x)$. This yields: $p_r(y \mid x) \propto p(y \mid x) \mathbb{P}(s=1 \mid x,y)$ where $\mathbb{P}(s=1 \mid x,y)$ is shorthand for $\mathbb{P}(S=1 \mid X=x,Y=y)$.
>
> Our tempered estimator is then $\hat p_r^{(T)}(y \mid x) \propto p_{\theta^\ast}(y \mid x)^{1/T} \hat{f}(x,y)$,
>
> where $p_{\theta^\ast}$ denotes the base model and $\hat{f}$ is the classifier.
>
> **(Q2) When the classifier is trained…**
>
> In the finite-sample setting, we train $\hat{f}$ on the regularized empirical risk. This does not contradict assuming $L(\hat{f})-L(f^\ast)\le\delta$; rather, regularization is what enables bounding this gap.
>
> The population risk $L(f_{\phi})=\mathbb{E}\_{z,s}[\ell(f_{\phi}(z),s)]$ for a classifier parameterized by $\phi$ is defined without regularization, but we minimize the regularized finite-sample loss $\frac{1}{n} \sum\_{i=1}^n \ell(f\_{\phi}(z_i),s_i) + \lambda \| \phi \|_2^2$ (Eq. 3) to improve generalization performance.
>
> Appendix C provides analysis using standard tools [1] for logistic regression showing that if $\hat \phi\_{\lambda}$ minimizes the $n$-sample regularized loss then $L(f_{\hat \phi\_{\lambda}}) - L(f^\ast) = \mathcal{O}(1/\sqrt{n})$ when $\lambda$ is properly chosen and the optimal classifier is realizable. Thus, minimizing the regularized finite-sample problem bounds the excess risk $L(f_{\hat \phi\_{\lambda}}) - L(f^\ast)$.
>
> We will update the paper to clarify this.
>
> [1] "Understanding Machine Learning: From Theory to Algorithms." (Shalev-Shwartz and Ben-David)
>
> **(Q3) Why is it important…**
>
> Minimizing the Forget Error (Eq. 7) ensures that the tilted distribution does not leak information about the forget set after unlearning.
>
> While some unlearning methods, mainly for supervised tasks like classification, aim to degrade performance on the forget set so that the updated model's predictions diverge from the original [1], we study exact unlearning which aims to recover the model that would result from training on only the retain set, i.e., requiring $\hat{p}_r(z) = p_r(z)$. This follows prior work [2] and ensures no information from the forget set is leaked by $\hat{p}_r(z)$.
>
> We will update the paper to clarify this.
>
> [1] “Towards Unbounded Machine Unlearning.” (Kurmanji et al, 2023)
>
> [2] “Certified Data Removal from Machine Learning Models.” (Guo et al, 2020)

---

> > ### Author Rebuttal · Reviewer_GFTS · 2026-04-04
> >
> > Thank you for your answers. I will keep my scores.

---

### Official Review · Reviewer_Jfih · 2026-03-13

**Soundness:** 3
**Presentation:** 3
**Significance:** 3
**Originality:** 3
**Overall Recommendation:** 4
**Confidence:** 2

**Summary:**

This paper studies machine forgetting in large generative models, focusing on addressing the information leakage problem guided by the classifier when the forgetting set is highly concentrated. The authors propose T3-Unlearning, a two-step method that first smooths the distribution and then corrects the lightweight classifier, directly achieving forgetting through distribution density ratio estimation rather than fine-tuning the model. Both theoretical and TOFU benchmark experiments demonstrate that this method achieves an order-of-magnitude improvement in computational efficiency while maintaining forgetting quality and generation utility.

**Compliance With Llm Reviewing Policy:**

Affirmed.

**Key Questions For Authors:**

1. To what extent does the assumption of a mixed distribution for data (Equation 1) limit the theoretical guidance of the T3 method on unstructured real data, given that the forget set is often a complex subset of the original training data?

2. In real unlearning scenarios lacking retraining models (where Forget Quality cannot be calculated), does the author have an insightful method or theoretical guidance for dynamically selecting the optimal T value?

3. For the high-dimensional complex space generated by LLMs, is a simple linear classifier sufficient to capture the complex manifold structure of the "forget set"? Under what circumstances would this classifier fail?

**Limitations:**

Yes

**Strengths And Weaknesses:**

## Strengths:
1. Through a rigorous theoretical framework and finite sample analysis, the paper transforms the forgetting problem into a distribution correction problem and proves the necessity of tempering. Experimentally, the effectiveness of the method is verified on the TOFU benchmark.

2. The paper is well-structured, logically sound, and clearly illustrated, with precisely defined metrics (e.g., MU-ROUGE) to evaluate generation quality.

3. The work addresses the critical need for efficient machine unlearning in LLMs under privacy regulations and introduces a novel paradigm of treating unlearning as distribution correction rather than parameter updates.

## Weaknesses:
1. Theoretical analysis relies on idealized assumptions (e.g., perfect data distribution learning, disjoint forget/retain sets) that may not hold in complex real-world LLM scenarios.

2. The synthetic experiments rely solely on Gaussian distributions, which fail to capture the complexity of high-dimensional text spaces.

3. The theoretical sections use dense notation, which may be difficult for non-theory readers; intuitive explanations for key symbols in the main text are recommended.

---

> ### Author Rebuttal · Authors · 2026-03-31
>
> **(W1) Analysis relies on assumptions…**
>
> Our theory assumes the pretrained model has exactly learned the original distribution, meaning $p_{\theta^\ast}=p$. If we instead assume $p_{\theta^\ast}\neq p$, then our bounds gain an additional term caused by this mismatch.
>
> Theorem 3.3 shows for $\hat p_r^{(1)} \propto p \cdot \hat{f}$ that $\mathrm{KL}(p_r || \hat p_r^{(1)}) \le \frac{\delta}{1-\gamma}$. If instead $\hat p_r^{(1)} \propto p_{\theta^\ast} \cdot \hat{f}$ with $p_{\theta^\ast} \neq p$, we then obtain:
>
> $\mathrm{KL}(p_r || \hat p_r^{(1)}) \le \frac{\delta + \mathrm{KL}(p \| p_{\theta^\ast})}{1-\gamma}$,
>
> so even with $\delta = 0$ we incur the irreducible error $\mathrm{KL}(p || p_{\theta^\ast})$. We expect similar irreducible error to appear in all bounds, as tilting cannot overcome this mismatch $p_{\theta^\ast} \neq p$.
>
> Regarding the assumption that $\mathcal{D}\_r$ and $\mathcal{D}\_f$ are disjoint, during unlearning we are given the finite-sample forget set $\mathcal{D}\_f$, and then define the retain set $\mathcal{D}\_r = \mathcal{D} \setminus \mathcal{D}\_f$. Thus, $\mathcal{D}\_r$ and $\mathcal{D}\_f$ are disjoint by definition. However, we assume no structure on the population distributions $p_r$ and $p_f$. For example, $\mathcal{D}_r$ and $\mathcal{D}_f$ could contain overlapping phrases of text even though their exact contents are disjoint. Our framework naturally accommodates this setting.
>
> We will update the paper to clarify these assumptions.
>
> **(W2) The synthetic...**
>
> We use Gaussian synthetic data as an interpretable setting to validate our theory. This is complementary to our main results on TOFU which show our method outperforms existing baselines for unlearning with respect to complex distributions over text.
>
> **(W3) The theoretical sections...**
>
> We will update the paper to improve readability and add intuitive explanations.
>
> **(Q1) To what extent...**
>
> We assume $p$ can be decomposed as a mixture of $p_r$, representing the retain-only distribution, and $p_f$, the forget-only distribution.
>
> This model holds exactly for the finite-sample distributions over the training data, as the full data $\mathcal{D}$ is partitioned into the retain and forget sets $\mathcal{D}_r$ and $\mathcal{D}_f$. Eq. 1 extends this to the population level by taking the infinite-sample limit.
>
> This assumption is very general and doesn't impose any constraints on $p_r$ or $p_f$. In particular, if there exists a (possibly stochastic) rule $s(z) \in \\{0,1\\}$ indicating whether a sample $z$ belongs to the retain set (i.e., $s(z)=1$) or the forget set (i.e., $s(z)=0$), then $p$ is mixture of $p(z \mid s(z)=1)$ and $p(z \mid s(z)=0)$, corresponding exactly to $p_r$ and $p_f$, respectively.
>
> We will update the text clarify this interpretation.
>
> **(Q2) In real…**
>
> In theory, our analysis provides guidance on how the temperature $T$ mediates the tradeoff between Retain and Forget Errors (Eqs. 6 and 7), characterizing when tempering the base model is beneficial. However, these bounds depend on properties of the underlying data distribution, such as the sharpness $\|p_f\|_{\infty}$, entropy $H(p_r)$, and the classifier excess risk $\delta$, which are inaccessible in practice, making direct optimization of these bounds intractable.
>
> Similar to existing LLM unlearning methods, $T$ and other hyperparameters (e.g., learning rate, $\lambda$, classifier rank $h$) are found via empirical tuning. Since $T$ is applied only at inference and decoupled from training, we note that it can be adjusted post hoc. Principled hyperparameter selection without retraining or validation remains an open challenge which our work does not fully resolve.
>
> **(Q3) For the high-dimensional…**
>
> Our low-rank linear classifier operating on pooled hidden states of the base LLM offers several practical advantages. Its effectiveness is supported by the expressiveness of the base LLM, as the base model acts as a complex feature map whose output is fed into the trainable classifier. The pooled hidden states encode rich context about the input sequence for the classifier to operate effectively.
>
> The low-rank linear parameterization also acts as a useful regularizer, limiting overfitting. We find that small ranks perform best, as our results on the TOFU benchmark used a classifier rank $h=20$ for the Llama 3.1 8B model which maps hidden states of dimension 4096 to logits over the vocabulary of size 128,256
>
> However, more complex settings where more data is available may require higher classifier capacity. If the classification task is non-linear in the hidden states, or pooling destroys key token-wise information, more expressive architectures such as MLPs or additional transformer layers could be beneficial. Scaling classifier complexity provides a natural way to trade off expressiveness and efficiency depending on the difficulty and sample size of the unlearning task.

---

> > ### Author Rebuttal · Reviewer_Jfih · 2026-04-06
> >
> > I will keep the score.

---

### Decision · Program_Chairs · 2026-04-30

**Decision:**

Accept (regular)

**Comment:**

The paper frames machine unlearning in large generative models as a density ratio estimation problem, avoiding fine-tuning. The authors propose T3-Unlearning, show the necessity of the tempering step when the forget set is concentrated, and show good empirical performance on the TOFU benchmark. Reviewers found the paper well-structured, logically sound, and practically relevant. Overall the idea is interesting and well-executed, though broader evaluation and threat model would be helpful.